# The Role of Transposable Elements of the Human Genome in Neuronal Function and Pathology

**DOI:** 10.3390/ijms23105847

**Published:** 2022-05-23

**Authors:** Ekaterina Chesnokova, Alexander Beletskiy, Peter Kolosov

**Affiliations:** Laboratory of Cellular Neurobiology of Learning, Institute of Higher Nervous Activity and Neurophysiology of the Russian Academy of Sciences, 117485 Moscow, Russia; apbeletskiy@gmail.com (A.B.); kolosov@ihna.ru (P.K.)

**Keywords:** transposons, transposon silencing, transposon exaptation, somatic mosaicism in neurons, neurogenesis, synaptic plasticity, human brain evolution, neuropathology

## Abstract

Transposable elements (TEs) have been extensively studied for decades. In recent years, the introduction of whole-genome and whole-transcriptome approaches, as well as single-cell resolution techniques, provided a breakthrough that uncovered TE involvement in host gene expression regulation underlying multiple normal and pathological processes. Of particular interest is increased TE activity in neuronal tissue, and specifically in the hippocampus, that was repeatedly demonstrated in multiple experiments. On the other hand, numerous neuropathologies are associated with TE dysregulation. Here, we provide a comprehensive review of literature about the role of TEs in neurons published over the last three decades. The first chapter of the present review describes known mechanisms of TE interaction with host genomes in general, with the focus on mammalian and human TEs; the second chapter provides examples of TE exaptation in normal neuronal tissue, including TE involvement in neuronal differentiation and plasticity; and the last chapter lists TE-related neuropathologies. We sought to provide specific molecular mechanisms of TE involvement in neuron-specific processes whenever possible; however, in many cases, only phenomenological reports were available. This underscores the importance of further studies in this area.

## 1. Mobile Genetic Elements in the Human Genome and Their Regulation

### 1.1. Transposon Classification and Types of Transposable Elements in Our Genome

Transposable elements (TEs), also known as transposons, mobile genetic elements, or “jumping genes”, constitute a large fraction of most eukaryotic genomes. Since their discovery by Barbara McClintock [1], transposons in genomes of multiple species have been thoroughly studied and classified, but their interaction with host genes is so complex and diverse that it is still a subject of many ongoing research projects.

TE mobilization is a powerful source of mutagenesis. Besides insertion mutations, repetitive elements provide a basis for nonallelic homologous recombination, leading to the deletion or duplication of genomic fragments [2,3]. An impressive example of TE-mediated adaptive mutation is the famous industrial melanism of the peppered moth [4]. TE-induced mutagenesis may also affect epigenetic regulation, gene expression rate and mRNA processing (see [3,5] for a review).

TEs differ by their mechanisms of mobilization, structure and size. Two main classes of TEs are retrotransposons (class 1) and DNA transposons (class 2). DNA transposons are more ancient; they do not use RNA intermediates for transposition and most of them do not replicate, but use “cut and paste” mechanisms. Retrotransposons replicate using RNA intermediates and reverse transcription (RT) [3,6]. Although the origin of retrotransposons can be traced to near the split of prokaryotes from eukaryotes, these elements only started to thrive around the beginning of mammalian evolution [7]. Retrotransposons include LTR and non-LTR elements, distinguished by the presence or absence of long terminal repeats. Non-LTR elements include two major groups: long interspersed nucleotide elements (LINEs), which are a few kilobases long, and short interspersed nucleotide elements (SINEs), which are usually under 600 bp [6,8]. Another essential difference is that LINEs (and LTRs) are transcribed by RNA polymerase II (pol II), while most SINEs are transcribed by RNA polymerase III (pol III) [8,9]. All active SINEs in mammalian genomes are derived from pol III-transcribed genes [10]. Quite obviously, SINEs inserted within pol II-transcribed genes are also transcribed by pol II as a part of a corresponding transcript [11]. SINEs are nonauthonomous, meaning they depend on the machinery of LINEs in order to retrotranspose. Nevertheless, SINE transcription is controlled by their own regulatory elements [12]. Most mammalian SINEs have a 3′ tail homologous to a “partner” LINE; this LINE-like region allows SINEs to be recognized by LINE retrotransposition machinery [13]. More complex TE classification systems are used to accommodate specific minor TE groups across different taxa [14,15].

The initial sequencing of the human genome estimated that about 45% of our genome is derived from different TEs [16]. More recent TE annotation demonstrated that this fraction is actually higher [17]. By some estimates, TEs and products of their activity may even constitute up to 69% of the human genome [18]—TE insertions that experience no purifying selection inevitably accumulate mutations with time, so the most ancient TE copies may be unrecognizable by homology-based methods [13]. By the current estimation, the human genome contains traces of 10 clades of LINEs, 3 types of SINEs, 1 composite retrotransposon family (SVA), 5 classes of LTR retrotransposons, and 12 superfamilies of DNA transposons [19]. Among these, only three groups of non-LTR retrotransposons (Figure 1): L1 (LINE), Alu (SINE) and SVA, are currently capable of mobilization [3,20]. These three groups of TEs together comprise about a third of the human genome [5,7], with L1 taking up the largest fraction of the genome [5] but Alu being the most numerous [11]. The current estimated rates of germline insertion are about 1 in 20 births for Alu, 1 in 270 births for L1, and 1 in 916 births for SVA [7]. These three TE groups are discussed in detail in Section 1.3, Section 1.4, Section 1.5 and Section 1.6.

Mechanisms of TE interaction with host cells are summarized in Table 1 and described in detail in Section 1.7 and Section 1.8.

Interestingly, TEs are able to horizontally transfer between genomes of diverse eukaryotic species. This happens more often with DNA transposons and LTRs, but horizontal transposon transfer was also reported for some non-LTR retrotransposons [21,22,23].

### 1.2. Inactive TEs in the Human Genome

Our genome harbors multiple copies of TEs that are no longer able to transpose but may still be transcribed and/or used by the host cell as regulatory elements.

Human LTR elements are represented by endogenous retroviruses (ERVs) or their remnants (solitary LTRs), which together account for about 8% of our genome [24]. The genomic organization of intact ERVs is similar to the provirus of their exogenous ancestors [3]. A typical replication-competent provirus is 7–11 kb in size. Its 3′ and 5′ ends have identical LTRs containing a pol II promoter and binding sites for cellular transcription factors (TFs). The canonical ERV genome in its simplest form contains four genes (more accurately, four ORFs): *gag*, *pro*, *pol*, and *env* (Figure 1). More complex ERVs have additional ORFs. In the ERV genome, *gag* encodes structural components of the virion capsid, *pro* encodes protease, *pol* encodes reverse transcriptase/integrase, and *env* encodes a glycoprotein that binds with the receptor on the host cell surface, allowing the virus to enter the cell (*pro* and *pol* sequences are often considered as parts of one gene). Very few ERVs still retain the ability to produce infectious extracellular particles [9,25,26]. In the case of human ERVs (HERVs), such particles were found in pathologic tissues and cancer cell lines [27,28,29]. HERVs may still be transcribed in both healthy and pathologic tissues [9,30,31,32], but the ability of mobilization in HERVs is either nonexistent or very limited [33,34,35]. Out of at least 31 HERV subfamilies, only HERV-K has some indications of recent retrotransposition activity [3]. About 90% of human LTR elements are solitary LTRs [3,24]—products of homologous recombination between two LTRs that excludes viral genes and leaves one LTR at the locus [3,9,25,35]. DNA transposons are apparently extinct in mammals, with very few exceptions [22,36]; in the human genome, they constitute about 3% but are unable to mobilize [3,5]. DNA transposons have a single ORF encoding a transposase enzyme (Figure 1); this ORF is usually rendered inactive by nonsense or frameshift mutations. However, some of these are still transcribed and present a potential threat to the genome [37].

There are also multiple non-LTR retrotransposon families that are currently unable to mobilize but have many copies in the human genome [19]. A prominent example is mammalian-wide interspersed repeats (MIRs, also called CORE-SINEs [38]) that were actively propagating prior to the radiation of mammals and still represent the second most numerous SINE subfamily in humans [9].

### 1.3. L1 Elements: Their Structure and Mechanism of Mobilization

L1 elements (also known as LINE-1) are a subset of long interspersed nucleotide elements. L1s are the only autonomous human retrotransposons because they encode their own reverse transcriptase. They constitute about 17% of our genome [5], with roughly 500,000 L1 copies per haploid genome [33] (by some estimations, there are ~900,000 copies of L1-derived sequences, including numerous fragmented elements [39]). About 79% of human genes contain at least one segment of L1 in their transcription unit, mostly in introns [40]. However, the overwhelming majority of L1 copies are degenerated—they are incapable of mobilization because of 5′ truncations, internal rearrangements, and other mutations. The number of L1 copies currently capable of retrotransposition is less than 100, and just a few of these were shown to be highly active, or “hot” [41]. Nevertheless, more recent analysis of newly inserted L1 elements from geographically diverse individuals demonstrated that “hot” L1s must be more abundant in the human population than was previously estimated; Beck et al. identified dozens of polymorphic active L1s with low allele frequencies [42].

A full-size L1 element is about 6000 bp long [34]; however, L1 truncation during retrotransposition is in fact so common that, in the human genome, an average size of all L1 copies is only 900 bp [16]. The L1 sequence includes an internal pol II promoter, which allows every full-size L1 copy to be transcribed independently of its location in the genome, and two open reading frames, ORF1 and ORF2, which encode proteins necessary for retrotransposition [34,43]. The L1 sequence also contains some elements that seem to be functional, but their role in L1 mobilization is not obvious. Two additional promoters reading “outwards” into the flanking genomic sequence were identified within L1: an antisense promoter in the 5′-UTR [44,45], and a sense promoter in the 3′-UTR [46,47]. These promoters were found in both human and mouse L1 elements [46,48]. There is also an antisense open reading frame, ORF0, that is only found in primate-specific L1s [49].

The strength of the main L1 promoter strongly depends on upstream flanking genome sequences that may either repress or enhance the promoter activity [50]. ORF1 encodes an RNA-binding chaperone ORF1p, and ORF2 encodes ORF2p, an enzyme with endonuclease and reverse transcriptase activities [5,34]. These proteins form a complex with L1 RNA (L1 ribonucleoprotein particle, or L1 RNP) in cytoplasm. L1 RNP includes one L1 mRNA, many ORF1p molecules, and one or two ORF2p molecules, so ORF1p expression level is much higher than that of ORF2p, and ORF1p is much easier to detect in experiments [51]. After being assembled in the cytoplasm, L1 RNP must enter the nucleus, and the exact mechanism of this is still not well understood. However, a noncanonical nuclear localization signal (NLS) was identified within ORF1p [52]. Host proteins that supposedly participate in L1 RNP nuclear import are transportin 1 (TNPO1) [53] and hnRNPA1, which associates with poly(A)+ RNAs [54]. The mechanism of L1 retrotransposition is called target-site primed reverse transcription (TPRT): ORF2p generates a single-strand nick in genomic DNA to expose a 3′-OH. The free DNA 3′ end is then used as a primer for ORF2p to initiate cDNA synthesis using the L1 mRNA as a template [55]. However, the processes of second-strand target site DNA cleavage and second-strand L1 cDNA synthesis are not well-understood yet [56]. Host DNA repair mechanisms are then necessary to restore dsDNA. DNA repair proteins may facilitate or prevent L1 integration, and experiments studying specific proteins produced inconsistent results in different models [57,58,59,60]. L1 endonuclease has relatively weak target-site specificity, preferentially cleaving a loosely defined motif 5′-TTTT/A-3′ (where “/”denotes the cleavage site) [61]. TPRT typically results in L1 insertions with recognizable features: insertion at an L1 endonuclease motif, target site duplications and a poly(A) tail. The resulting L1 insertion is very often truncated at the 5′ end. The exact reason of truncation is not clear, although there are some data suggesting that host DNA repair systems may intervene in the RT before it is complete [58,59]. In addition, during the insertion, a 5’ segment of an L1 may become inverted with respect to its 3’ end [62]. The supposed mechanism of such inversion is “twin priming”, when one overhang of the nicked host-DNA strand anneals to the poly(A) tail of the L1 RNA, while the other one folds and anneals somewhere in the middle of the L1 RNA [63]. A minor fraction of L1s insertions happens not by TPRT but rather by an endonuclease-independent mechanism. These insertions likely occur at pre-existing DNA lesions and lack target site duplications [61,64]. There is also evidence that L1 endonuclease activity may cause DNA double-strand breaks (DSBs) without subsequent insertion of the retrotransposed sequence [57].

L1s are the major source of insertional mutagenesis in the human genome [65]. Each individual has L1 copies not present in the reference genome, which contribute to as much as 20% of all structural variants in humans [66]. Moreover, in some cases, L1 insertions may cause significant deletions of genomic sequence at the target site [67]. Even chromosome rearrangements may potentially result from L1 activity [68]. L1 insertions are interspersed throughout the genome [36]. However, the analysis of de-novo L1 insertions and of recently integrated elements in the human genome revealed the presence of insertional hotspots [69]. Recent studies provide evidence that L1 integration events do not target expressed genes, open chromatin or transcribed regions but instead associate with DNA replication [66]. Another aspect of L1 endonuclease specificity is that this enzyme preferentially attacks pre-existing (germline) L1 elements, which makes these regions particularly prone to DSBs [70].

Nearly all of the current L1 activity in the human genome is due to one particular subfamily, Ta1 within the L1HS family. This family is also human-specific, hence the name [42,71,72].

### 1.4. Trans RNA Targets Mobilized by L1 Elements and Gene Retrocopying

L1 elements are not just able to copy themselves, but their enzymatic machinery may also be used to mobilize other DNA sequences if the latter are transcribed. Non-autonomous retroelements, such as Alu and SVA, “hijack” L1s to be retrotransposed. Similar hijacking is possible for some host genes: both non-coding and messenger RNA transcripts may be used as templates for TPRT. This causes formation of retrocopies with a set of specific features—loss of introns and promoter, acquisition of a poly(A) on the 3′ end and target-site duplications of varying lengths [34,65]. It must be noted that there is some confusion about the usage of “pseudogene”, “processed gene”, “retrogene” and similar terms [73], especially in some early works. These terms are used in different papers, sometimes interchangeably, to describe a gene retrocopy generated by L1 reverse transcriptase from a host mRNA; for a long time, all retrocopies were routinely annotated as pseudogenes [74]. Nevertheless, the term “pseudogene” means any genomic sequence that is similar to a specific gene but is somehow defective, and while most pseudogenes are indeed formed by retrocopying, there are also TE-independent mechanisms of pseudogene formation [75,76] and retrocopies that are functional genes [77]. Most pseudogenes originate from genes highly expressed in the germline, such as housekeeping genes and ribosomal protein genes [73,75]. In the human genome, 3391 fully functional “parent” genes were associated with pseudogenes, and it was found that almost two thirds of parent genes have only given rise to a single pseudogene, while a small fraction of parent genes produced dozens of pseudogenes each [75].

L1 retrotransposition machinery has a strong *cis* preference, meaning that ORF1p and ORF2p preferentially associate with their own mRNA, so L1-mediated retrocopy formation is quite rare. Experiments with retrotransposition assay in cultured cells showed that mobilization of other cellular RNAs happens in less than 0.05% of all L1 transposition events [78]. Nevertheless, it is suggested that *trans* retrotransposition events may account for at least an additional 10% of human DNA [79]. The very low efficiency of *trans* retrocopy formation observed by Wei et al. in cell cultures [78] may reflect only the retrotransposition of two specific *trans* targets used in this experiment. Indeed, there are reports that some *trans* sequences are retrocopied by the L1 machinery more effectively. Based on the hallmarks of L1-mediated retrotransposition in the human genome, most retrotransposed *trans* RNAs (that are not Alu and SVA elements) are pol III transcripts associated with the ribosome and nucleolus. L1 preferential targets include some highly structured small RNAs and some mRNAs encoding housekeeping proteins. The suggestion that L1 machinery has a preference for some specific *trans* RNA targets based on the the high copy number of these sequences in the genome was confirmed in vitro in cells with ectopically expressed ORF2p: analysis of RNAs associated with L1 RNPs showed the presence of Alu, SVA, U snRNAs and hYRNAs. Surprisingly, it was shown that RPLP1, GAPDH and β-actin mRNAs are not only present in the L1 RNPs, but are more abundant there than the L1 transcript itself [80]. Similarly, a recent study showed that, among all RNA species bound by ORF1p in prostate cancer cells, only a small percentage is represented by L1 RNA [81]. It is therefore possible that the *cis* preference of L1 machinery should be explained not by preferential RNA binding but by some other factors. However, the secondary structure of RNA seems to be important for succesful retrotransposition by L1 [82]. L1 RNPs binding *trans* RNA targets is not the only mechanism of retrocopy formation: there is some evidence that ORF2p may switch from L1 RNA to other RNA templates after TPRT has already started. Such template switching was confirmed to mobilize some small non-coding RNAs [83]. The total number of retrocopies in the human genome is estimated to be from 8000 to 17,000 [84]. Ewing et al., by comparing individual whole-genome sequence data with corresponding reference genome assemblies, estimated the rate of novel germline retrocopy insertions in humans as about 1 insertion per 6000 individuals [73]. Very few retrocopies are expressed because they generally lack a functional promoter [34]. As in other cases of gene duplication, retrocopies have a functional counterpart elsewhere in the genome, so they are free from selective pressure and thus can accumulate mutations and acquire novel functions [2,74]. Alternatively, retrocopies may substitute for their parental genes. In mammals, many housekeeping X chromosome genes have given rise to functional retrocopies located on autosomes. Autosomal retrocopies are specifically expressed in testis during and after the meiotic stages of spermatogenesis, when all X chromosome genes are transcriptionally silenced due to male meiotic sex chromosome inactivation [74,77]. Lastly, functional retrocopies may simply provide more resources for producing similar proteins, an example of this being genes encoding APOBEC3 antiviral proteins [85].

Exogenous RNA species may also be retrotransposed. cDNA copies of nonretroviral RNA virus sequences have been detected in genomes of many vertebrate species, and some of these elements have signs of integration assisted by LINE retrotransposons. A very recent study showed that SARS-CoV-2 sequences can integrate into genomes of cultured human cells by an L1-mediated mechanism [86].

The ability of L1 machinery to process *trans* RNA templates was used to create a reporter assay that is now a common method for estimating L1 retrotransposition rate. Ostertag et al. designed an L1-EGFP retrotransposition reporter cassette that is capable to express EGFP only after being retrotransposed. The cassette containing the EGFP coding sequence and regulatory elements (a promoter and a poly(A) signal) was inserted into the L1 3′-UTR in the antisense orientation. Importantly, the EGFP gene within this cassette is disrupted by an intron (containing splice donor and splice acceptor sites) in a sense orientation. The L1 element tagged with the EGFP cassette was then cloned into a pCEP-based mammalian expression vector. Thus, cells transfected with this construct may only express EGFP after an L1 transcript containing the antisense EGFP marker has undergone splicing, RT and integration into chromosomal DNA [87]. This method was used in many of the articles cited below.

### 1.5. Alu Elements

Alu elements constitute about 11% of the human genome [5,88], with about one million copies of Alu per haploid genome [33], which makes them the most successful retrotransposons in humans [11]. Three quarters of human genes have Alu insertions [88].

The name of these elements originally comes from AluI, an *Arthrobacter luteus* endonuclease, because most of these TEs bear an AluI restriction site [89,90]. Alus are about 280 bp long [88,91]. Alu elements are derived from the 7SL RNA gene [90,92,93]. The RNA encoded by this gene is a component of the protein signal recognition complex that directs translating ribosomes to the endoplasmic reticulum membrane [94]. Alu elements contain two monomeric sequences derived from 7SL RNA and end in an A-rich tail. The left monomer contains an internal pol III promoter and is separated from the right monomer by an adenosine-rich sequence [93]. Left and right Alu monomers are imperfect repeats of each other. A number of independent left and right Alu monomers have also been identified, the most important of them is a part of the human BC200 RNA gene [95] (discussed in the Section 2.7.3). The internal pol III promoter is too weak to support efficient transcription by itself; genes of pol III transcripts require additional *cis*-acting elements to stimulate their transcription. In the case of Alus, their ability to be transcribed depends on their flanking sequences, so each of one million Alu members might be, to some extent, uniquely regulated [93,96].

Alus are not autonomous and may only be mobilized by hijacking L1 molecular machinery [90]. However, Alus do not have a specific region homologous to L1 or any other “partner” LINE; L1 ORF2p binds tightly to the A-rich tail and the adjacent RNA sequence is not very important for this [97,98]. Still, L1 machinery somehow discriminates between Alu RNAs and mRNAs that also have a poly-A tail. Grechishnikova and Poptsova identified a highly conserved 3′-UTR stem-loop structure in human L1 and Alu RNAs and suggested that this secondary structure is necessary for binding ORF2p [82]. L1 ORF2p also has relatively weak target-site specificity as an endonuclease, and this explains why more recent Alu and L1 insertions exhibit similar interspersed integration patterns. However, it was repeatedly demonstrated that evolutionarily older L1 and Alu insertions show distinct genomic distributions: older L1s are found mostly in gene-poor AT-rich sequences, whereas older Alus are common in gene-rich GC-rich regions of the genome. This difference likely results from post-integration selection pressure. Some researchers suggest that Alus may possibly have some useful function in gene-rich regions of the genome; the other version is that L1 insertions into genic regions are more deleterious compared with Alu insertions [36]. Indeed, even while human Alus are enriched in genes compared with intergenic regions, they are rarely found in exons [88], suggesting that exon disruption by Alus is harmful.

Alu elements are primate-specific [91], even though there are 7SL RNA-related SINEs in other species. Rodent B1 SINEs are the most well-known example. The B1 sequence is very similar to the left Alu monomer [99]. It must be noted that, in RepeatMasker, a commonly used bioinformatics tool, rodent B1 elements are classified as “Alu family repeats” [100], which sometimes leads to confusion. Rodents also have B2 SINEs that bear no structural homology with Alus, being tRNA-related, but may be regarded as Alu counterparts by their functions. Rodent B2s, just like primate Alus, participate in RNA editing and cell stress response (discussed in Section 1.8.7 and Section 1.7.7, respectively) [11]. Alu-insertion polymorphisms are very useful for studying human phylogeny and population genetics [90]. The most active Alu subfamilies are AluYa5 and AluYb8 [72].

### 1.6. SVA Elements

SVA elements only comprise about 0.13% of the human genome, with about 2700 copies. This is the smallest of the known retrotransposon families [20]. Notably, ~1000 of SVA repeats reside within introns of genes [101].

SVAs stand out because they are composite elements. The acronym “SVA” means “SINE-R, VNTR, and Alu” since these elements contain a SINE-R sequence, a variable-number-of-tandem-repeats (VNTR) locus and an Alu-like sequence. The SINE-R sequence had been historically described as a SINE transposon; however, it is related to the endogenous retrovirus HERV-K10, sharing high homology with its 3′ part that includes *env* gene and the 3′-LTR [20,102]. A “canonical” SVA sequence is about 2000 bp long, but different SVA insertions may have sizes from 700 to 4000 bp [103]. This length variation is due to differing numbers of hexamer repeats in the SVA 5′ region and tandem repeats in the VNTR part [104]. Just like Alus, for their mobilization, SVAs need L1 molecular machinery. Endogenous SVAs are very likely transcribed by pol II [105]. However, SVAs do not have any obvious promoters at their 5′ end. Despite the lack of promoters and small copy number, SVAs are retrotranspositionally quite active; a few pathogenic mutations caused by SVA insertions have been described [106]. It was shown that SVA elements can use external heterologous promoters for their own transcription [107]. SVA sequnce contains an internal enhancer element which was suggested to recruit TFs to such promoters [103]. GC content in SVAs is generally around 60%, with >70% within the VNTR region; this means that SVA insertions may be considered CpG islands and may thus function as transcriptional regulators, binding TFs, altering the local chromatin structure or forming alternative DNA structures that affect transcription. SVAs make up nearly 2% of genome regions predicted to form G-quadruplex structures [108]. Experiments with reporter gene expression confirmed the ability of SVA elements to act as transcriptional regulators in vitro and in vivo; the effect was different in different models. Notably, different parts of the SVA sequence displayed distinct regulatory properties [20,109].

SVA elements are hominid-specific and represent the youngest family of hominid non-LTR retrotransposons [7,104,107]. SVA insertions are valuable markers for human phylogenetic and population genetic studies. The most active SVA subfamilies are SVA E, F and F1 [72].

**Figure 1 ijms-23-05847-f001:**
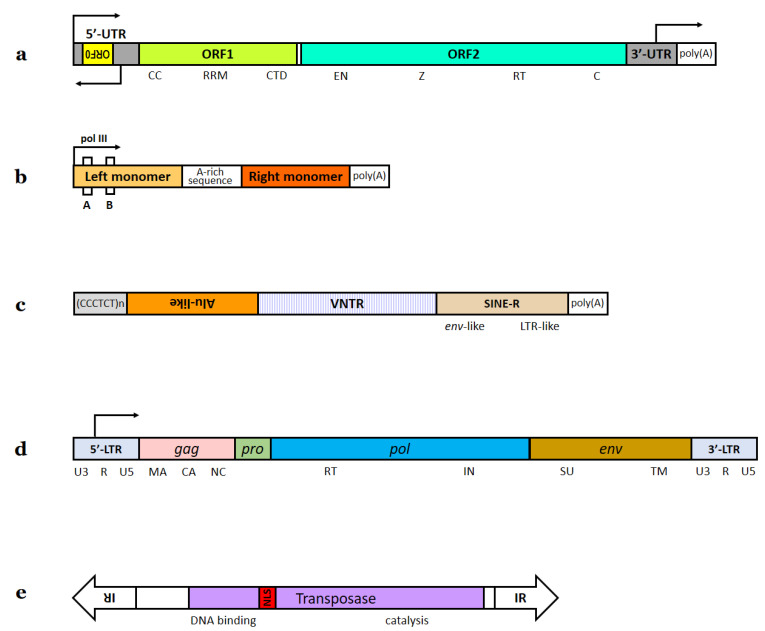
Types of human transposons. (**a**). L1 element. Two ORFs of human L1 are separated by a non-coding spacer region containing a stop codon [48]. The L1 5′-UTR has promoter activity in both sense and antisense directions. A small primate-specific antisense ORF0 was identified within L1 5′-UTR downstream of the antisense promoter [49]. An additional sense promoter was identified in 3′-UTR [46]. ORF1 encodes a protein that contains a coiled–coil domain (CC), a non-canonical RNA recognition motif domain (RRM) and a basic *C*-terminal domain (CTD). ORF2 encodes a protein that has endonuclease (EN) and reverse transcriptase (RT) activities. Functions of the Z domain and the cysteine-rich motif (C) are unknown [101]. (**b**). Alu element. These elements contain two monomeric sequences derived from 7SL RNA and end in an A-rich tail. The left monomer contains an internal pol III promoter and is separated from the right monomer by an adenosine-rich sequence [34]. Alu pol III promoter elements (Box A and Box B) may only provide efficient transcription if an upstream pol III enhancer is present at the site of integration [93]. (**c**). SVA element. These elements include CCCTCT hexameric repeats, the Alu-like domain consisting of two antisense Alu fragments separated by a unique sequence, a GC-rich VNTR domain consisting of tandem repeats, and the SINE-R domain sharing homology to the 3′ end of the HERV-K10 element (a fragment of its *env* gene and right LTR) [103]. (**d**). ERV. The order of the structural genes and the arrangement of their major cleavage products are completely conserved among all retroviruses and are necessary for correct virion assembly. Domains within Gag protein: matrix (MA), capsid (CA), nucleocapsid (NC). Domains within Pol protein: reverse transcriptase (RT), integrase (IN). Domains within Env protein: surface protein (SU), transmembrane protein (TM). Env is translated from a spliced subgenomic RNA and is then cleaved to generate SU and TM subunits. Each of the LTRs is composed of the unique U3 and U5 regions separated by a segment (R) repeated at each end of the viral RNA. U3 contains TF binding sites. Transcription from the provirus starts at the left U3-R junction, and the right R-U5 junction provides the site of 3′-polyadenylation. The left U3 and the right U5 are restored during dsDNA formation in the virus replication cycle [25]. (**e**). DNA transposon (Tc1/*mariner* family). The terminal inverted repeats (IR) contain binding sites for the transposase. The *N*-terminal part of the transposase contains a DNA binding domain. The *C*-terminal part of the protein performs target DNA site cleavage and transposon integration. NLS—nuclear localization signal [110]. The elements in this figure are not drawn to scale.

### 1.7. Molecular Mechanisms Suppressing TE Activity and Specific Cases of TE Unsilencing

TE mobilization or expression can have detrimental consequences for the cell, including genome instability, activation of DNA damage stress response, or toxic effects from accumulation of TE-derived RNAs and proteins. For example, ectopic expression of full-length L1 in cultured cells was shown to cause high levels of DSBs, cell cycle arrest, induction of a senescence-like cellular state and apoptosis; moreover, somatic L1 insertions within genes are associated with cancer [111]. Due to this, host organisms have evolved multiple elaborate mechanisms to control transposition events [112]. Systems that suppress TE activity overlap with mechanisms regulating host chromatin structure and DNA repair [36]. TE repression includes epigenetic silencing, RNA interference and some other mechanisms discussed in Section 1.7.1, Section 1.7.2, Section 1.7.3, Section 1.7.4 and Section 1.7.5.

There are specific cases when TE activity is permitted. Notably, TEs play an integral role in early mammalian embryonic development. Regulation of TEs is stage-specific, and on some stages TE repression is temporarily lifted [113,114,115]. Strikingly, L1 RNA in embryonic stem cells (ESCs) is critical in recruiting necessary machinery to direct a transcriptional program specific to the two-cell stage [115]. L1 epigenetic repression is usually established during early gastrulation and maintained thereafter [116]. Other cases of TE derepression include cell stress response and neuronal differentiation; we discuss these processes below in Section 1.7.7 and Section 2.4, respectively.

It must be noted that some regions in vertebrate genomes seem to be absolutely intolerant of TE insertions. Human and mouse genomes each contain almost 1000 transposon-free regions (TFRs) longer than 10 kb. Human, mouse, opossum, *Xenopus tropicalis* and zebrafish genomes have many orthologous TFRs; however, the sequence of most TFRs is not highly conserved. Over 90% of the TFR sequence is noncoding, and most of these regions do not show unusual nucleotide composition. Apparently, these regions are kept TE-free because most of them are significantly associated with genes encoding vertebrate developmental regulators, such as members of the *HOX*, *SOX*, *FOX* and *TBX* gene families [117,118].

#### 1.7.1. DNA Methylation

Most TEs are normally silenced by DNA methylation. The majority of cytosine methylation in mammals occurs within repetitive elements [36]. TE methylation in the human genome is in fact so thorough that, considering the large portion of the genome occupied by TEs, measuring overall TE methylation level may be used as a simplified approach to estimate global DNA methylation level [119]. This is a potentially heritable epigenetic modification, so it may suppress deleterious TE insertions in the germline [120]. In mammalian cells, DNA methyltransferases DNMT3A and DNMT3B perform de-novo methylation, while DNMT1 shows preference towards hemimethylated substrates and maintains the methylation landscape. Recently, it was shown that DNMT1 also has de-novo methylation activity targeted specifically at retrotransposons [121]. Lack of TE methylation strongly reactivates their transcription: knock out (KO) of *Dnmt3L* (a testis-specific DNA methylathion regulator) in mice prevents de-novo methylation of L1 and LTR elements in male germ cells, leading to TE derepression, meiotic catastrophe and cell death [122]. DNA hypomethylation and transcriptional reactivation of replication-competent L1 copies happens in some tumors, causing increased rate of transposition [24,37]. Hypomethylation of different human TEs (usually HERVs or L1s) was also reported in some autoimmune diseases [123] and after exposure to various toxic substances or radiation [124,125]. Overall methylation of human Tes, especially Alus, decreases with age. Different TEs have different age-dependent demethylation rates, with L1 methylation being largely unaffected by aging; L1 demethylation occurring in cancer likely has different mechanisms [126,127]. However, it seems like highly specific TE demethylation happens in normal tissues and is important for their function. Examination of genome-wide TE DNA methylation status in 11 human cell types revealed tissue-specific hypomethylation signatures. Many genes proximal to hypomethylated TEs were important for tissue functions, and expression of these genes correlated strongly with TE hypomethylation. Moreover, some of these TEs contained tissue-specific TF binding motifs. For 26 of such TEs, enhancer activity was confirmed in a reporter gene assay [128]. Similar results were obtained by Philippe et al., in a study where transcriptional and epigenetic signatures of specific L1 copies were identified in 12 human somatic cell lines. The authors found that the bulk of L1 expression happens in a very restricted subset of L1 loci, and these loci are differentially expressed between distinct cell lines. Strikingly, even in cancer cell lines exhibiting high L1 expression, only selected L1 copies were unsilenced [129]. Using different strategies, Muotri et al. have shown that L1 retrotransposition can be modulated by MeCP2, a protein involved in global DNA methylation. They demonstrated that knock down (KD) of MeCP2 increases L1 promoter activity; that removal of DNA methylation with 5-azacytidine reduces MeCP2 association to L1 promoter and increases L1 expression; that MeCP2 KO in mice causes higher retrotransposition in the brain; and that MeCP2 KO neuroepithelial cells have more L1 copies in their genome than WT cells [130].

#### 1.7.2. Histone Methylation

Modifications of histone *N*-terminal tails alter the binding of proteins with DNA sequences. Nucleosomes associated with TEs are enriched for methylation of histone H3 at lysine 9 (H3K9), marking transcriptionally repressive and inactive chromatin [131]. There are at least two pathways of H3K9me3-mediated TE silencing in human cells: by the KZNF/KAP1 system (described in the next section), or by the human silencing hub (HUSH) complex partnering with the ATPase MORC2. HUSH preferentially binds young, full-length L1s within euchromatic regions, indicating that this is a targeted mechanism of host defense with the capacity to respond to novel insertions [132]. Another important histone modification involved in (less stringent) TE silencing is H3K27 tri-methylation (H3K27me3) deposited by the Polycomb repressive complex 2 (PRC2). H3K27me3 was previously considered to be specific for the silencing of protein-coding genes (facultative heterochromatin), but there is increasing evidence that this modification also targets TEs in diverse eukaryotic species [133]. Disruption of TE silencing by histone methylation may be seen in various cancer types overexpressing histone demethylase KDM4B able to demethylate H3K9me3. This results in altered gene expression and genome instability. Ectopic overexpression of KDM4B in cultured cell lines enhanced L1 retrotransposition efficacy, copy number, and associated DNA damage [134]. Histone modification is especially important for silencing SINE elements transcribed by pol III. ChIP assays in cultured cells showed that the DNA sequence of human (Alu) and mouse (B1 and B2) SINEs is heavily methylated and occupied by methyl-CpG-binding proteins. Despite this, pol III was also detected at these SINEs, as were the pol III-specific TFs TFIIIB and TFIIIC. Global genomic demethylation (achieved by treating cells with 5-azacytidine or *Dnmt1* KO) had little effect on the accessibility of SINEs to pol III transcription machinery or their expression levels. In contrast, pol III loading and expression of SINEs increased significantly in cells treated with chaetocin, an inhibitor of SUV39H1 methyltransferase that deposits H3K9me3 mark [135].

#### 1.7.3. Krüppel-Associated Box Domain Zinc Finger Proteins

Krüppel-associated box domain zinc finger proteins (called KRAB-ZFPs in mice and KZNFs in humans) constitute the largest family of TFs in vertebrates and are one of the most rapidly evolving families of proteins. KRAB-ZFPs have an *N*-terminal KRAB domain and a variable number of tandem C2H2 zinc fingers at the *C*-terminus. Zinc fingers bind with DNA, and different KRAB-ZFPs recognize different sequences. The KRAB domain is a strong transcriptional repressor; it works by tethering the KRAB-associated protein 1 (KAP1, also known as TRIM28) to the DNA sequence recognized by the zinc fingers. KAP1 acts as a cofactor and scaffold protein for the recruitment of multiple proteins participating in chromatin silencing, the most important of them is SETDB1, a methyltransferase that deposits H3K9me2/3 [136,137]. Several lines of evidence demonstrate that TEs are the primary targets of most KRAB-ZFPs, and it is now considered that the rapid expansion of KRAB-ZFP genes is being driven primarily by the coevolution of retrotransposons. The number of C2H2 zinc finger genes correlates with the number of ERVs in different species of tetrapods. In human ESCs (hESCs), about three quarters of KAP1-binding sites reside within retroelements [136]. KAP1 deletion leads to a marked upregulation of a range of ERVs in mouse ESCs and early embryos [138]. ChIP-seq experiments revealed that, in hESCs, KAP1 predominantly associates with active primate-specific retrotransposons. A reporter gene system in mouse cells used to screen multiple primate-specific KZNF genes allowed to identify two genes, *ZNF91* and *ZNF93*, encoding TFs that recognize and silence SVA and L1PA elements, respectively. Both genes emerged in the last common ancestor of humans and Old World monkeys and have rapidly undergone dramatic structural changes that parallel the evolution of their TE targets [139].

#### 1.7.4. RNA Interference

RNA interference is a mechanism of silencing transcripts when small silencing RNAs form a complex with an Argonaute family protein and then hybridize with their target RNAs, which eventually leads to the target being degraded or translationally repressed by specific proteins.There are three classes of small silencing RNAs: miRNAs (that mostly target cell mRNAs), siRNAs (targeting mostly viral RNAs) and piRNAs (targeting mostly TE RNAs). These classes have different lengths, and distinct mechanisms of formation and function (see [140,141,142,143] for review). Both piRNAs and siRNAs participate in TE silencing; earlier it was considered that piRNAs silence TEs exclusively in germline cells, while siRNAs do the same in somatic cells [144,145], but this distinction seems oversimplified in the light of more recent findings [146,147,148,149,150].

piRNAs interact with PIWI proteins. The *piwi* gene was originally discovered as an essential factor for germline stem cell self-renewal in *Drosophila*, and it was demonstrated that mutations in this gene cause hyperactivation of retrotransposons [145]. Both *Drosophila* and mouse have miltiple PIWI proteins, and mutations in these proteins lead to defects in germ cell development [151]. The PIWI–piRNA pathway maintaining germline genomic integrity is conserved in animals. However, piRNAs lack conserved structural motifs and sequence homology across species, being the largest and most heterogeneous class of small non-coding RNAs [152]. piRNAs are mainly derived from intergenic regions containing TEs, TE remnants, and other repetitive elements [145]. TE silencing by piRNAs and associated proteins happens co-transcriptionally or post-transcriptionally. Co-transcriptional silencing involves heterochromatinization and deposition of repressive chromatin marks at target loci. Post-transcriptionally, TE RNAs are cleaved by PIWI [152,153,154,155]. Until recently, piRNAs were thought to be restricted to germline cells, but a few years ago they were also detected in neurons [156] (this is discussed in more detail in the Section 2.7.3).

The siRNA pathway (involving Ago2 protein) was initially discovered to function using exogenous short RNAs in *Drosophila* cells [157,158]. Mutant flies with dysfunctional siRNA pathway components are hypersensitive to viral infections [159], and it was shown that Ago2 binds small RNAs derived from viral RNAs [160]. Endogenous Ago2 partners, endosiRNAs, were identified later. These RNAs originate from retrotransposons or other repetitive sequences in *Drosophila* genome. Some endosiRNAs overlap with piRNA sequences; however, the transcripts have different sizes and bind different proteins [144,157]. Endogenous siRNA derived from retrotransposons were also found in mammalian cells, and KO of *Dicer1* (encoding a key endonuclease generating siRNAs) caused TE derepression in mouse oocytes [149,150,161]. Young and Kazazian demonstrated that the L1 antisense promoter can be used to form L1-specific siRNAs [161]. Berrens et al. performed an acute deletion of *Dnmt1* in mouse ESCs to cause global DNA demethylation and showed that endosiRNAs are involved in an immediate response to TE derepression. Small RNA-seq detected a substantial increase in small RNAs mapping to some TEs. Ago2 immunoprecipitation confirmed that a subset of Ago2-bound endosiRNAs mapped to TEs. KD of Dicer or Ago2 in *Dnmt1*-deficient cells caused strong upregulation of transcription of some TEs. Long-term TE silencing by histone methylation in *Dnmt1*-deficient cells was observed at a later time point [162].

miRNA silencing was confirmed for L1 elements. It was shown that miR-128 directly targets L1 RNA for degradation. The same research group found additional mechanisms of miR-128 indirectly inhibiting L1 transposition: miR-128 binds mRNAs encoding both TNPO1 and hnRNPA1 proteins that are necessary for nuclear import of L1 RNPs [53,54].

#### 1.7.5. Other Mechanisms

Some other proteins interacting with nucleic acids, such as DNA repair factors and proteins binding virus-derived molecules, may also contribute to suppressing TE mobility.

It is well-known that proteins that limit the replication of retroviruses may also inhibit retrotransposons [163]. Proteins of the APOBEC3 family are an important example. These proteins repress retroviruses and retrotransposons by deaminating cytidines (C-U editing) in their RNA [164] or cDNA [164,165]. It was also shown that TE repression by some of the APOBEC3 proteins happens by a deamination-independent mechanism which is not well understood [165,166,167]. Antiviral proteins involved in transposon silencing also include TREX1, a DNase that breaks down ssDNA and dsDNA in the cytoplasm [168,169], and SAMHD1, which depletes dNTP levels and also silences L1 by a specific mechanism [170].

ERCC2 is a DNA helicase participating in nucleotide excision repair. Deficiency of ERCC2 was reported to increase retrocopying. The mechanism of this is unclear, but it is suggested that ERCC2 prevents TE-mediated insertions [171]. Mov10 is an RNA helicase that participates in miRNA-mediated mRNA silencing. However, it was also shown that, in murine neurons, Mov10 suppresses L1 retrotransposition in the nucleus by directly inhibiting cDNA synthesis. Apparently, the *N*-terminal region of Mov10 binds the reverse transcriptase domain of ORF2p, and the *C*-terminal part of Mov10 possessing the helicase activity unwinds the L1 RNA [172]. Another important RNA-binding protein that is involved in TE silencing is TDP-43 [112]; this protein is involved in pathogenesis of some neurodegenerative disorders, and so this mechanism is discussed at length in the Section 3.2.6.

#### 1.7.6. “Self-Restriction” Inherent for Many TEs Is Absent in L1 Elements

TE insertions disrupting host genes jeopardize TE survival in the evolutionary perspective. Thus, many TEs have developed their own highly specific mechanisms that direct their integration to intergenic regions or introns. Even TEs that preferably integrate into gene-rich regions use targeting mechanisms that prevent disruption of ORFs [36,97]. However, L1s do not have evident mechanisms preventing their retrotransposition into genes. Experiments with HeLa cells confirmed that new L1 insertions broadly target all regions of the human genome, being insensitive to transcriptional activity, although a bias for early-replicating genomic domains was noted [66]. Conversely, germline L1 copies are enriched in intergenic regions, which likely reflects negative selection [2,36]. An interesting consequence of purifying selection against L1 insertions is that the density of L1s on the human X chromosome is almost twice as much as on the autosomes [62]; in autosomes, L1 insertions may be cleared by recombination, while sex chromosomes have restricted recombination capacity [173].

Whole-genome profiling of somatic L1 insertions in a few brain regions and the myocardium [174] showed that the rate of such insertions within genes and promoters is, in fact, even higher than predicted for random distribution in all studied samples. No explanation of this observation was proposed in this paper, but it suggests that a specific mechanism directing L1 insertions into genes might exist. The in-vitro results obtained by Sultana et al. [66] suggest that L1 integration does not depend on transcription. Nevertheless, Baillie at al., studying somatic L1 insertions in the brain, found that protein-coding loci are disproportionately affected, and that many genes with intronic insertions are differentially overexpressed in the brain [175]. This leaves the question of L1 insertions targeting transcriptionally active regions open.

#### 1.7.7. TE Derepression during Cell Stress and Its Possible Explanations

TE activation (increased transcription or mobilization) during cell stress was registered in multiple eukaryote species [36]. In experiments in cell cultures and in vivo, TE activation was shown to be triggered by various stressors, including oxidative stress, heat shock, exposure to toxic substances or viral infections [11,33,36,96,176,177,178,179]. Stress-related TE activation may be a part of an adaptive response; examples of this are known in yeast, plants and insects [36,180,181,182]. The role of mammalian TEs in cell stress response is not absolutely clear.

Overexpression of L1 in tumors [36,111] may indicate that L1 activation in stressed cells is primarily a side effect of disrupted chromatin maintenance. In works where L1 activation in tumors and/or cells exposed to genotoxic agents was studied, it is considered either a consequence [183] or even a driver [184,185,186] of genomic instability. The only suggested mechanism by which L1 activation in stress conditions may benefit the host is by increasing genetic diversity [176], which is just a very general idea. Still, Okudaira et al. showed that the cellular cascades involved in inducing L1 retrotransposition after exposure to methamphetamine or cocaine are different from those that trigger L1 activity after DNA damage [187]; therefore, presumably there are multiple mechanisms involved in L1 activation during cell stress.

On the other hand, upregulation of SINEs during cell stress most likely has adaptive functions. It was repeatedly demonstrated that SINE expression in mammalian cells significantly and transiently increases after stress-inducing treatments such as heat shock, viral infection, ethanol application, and UV exposure [11]. This effect was observed in different mammalian species and even in silkworm [177,188]. SINE activation after heat shock was also demonstrated in vivo in experiments with hyperthermia in mice [179]. Notably, other mammalian pol III transcripts are unaffected after heat shock [188] or exposure to DNA-damaging agents [189]; therefore, SINE transcription upregulation must be specific. Mariner et al. showed that, in heat-shock conditions, human Alu RNA acts as a *trans*-acting transcriptional repressor by directly binding pol II [190]. A similar mechanism was shown for murine B2 RNA [191]. Another suggested role for Alu in cell stress is the regulation of stress-induced kinase PKR. PKR normally reacts to viral infection and is activated by viral dsRNA. However, some highly structured RNA species may block PKR activation. Chu et al. demonstrated that Alu RNA forms stable complexes with stress-induced kinase PKR and prevents its activation. The authors studied the consequences of heat-shock recovery, viral infection or cycloheximide treatment in cell cultures. In all three cases, an increase in full-length Alu RNA level was observed. PKR activity decreased in a dose-dependent manner with the level of Alu overexpression [192]. Still, some authors [189] suggest that SINE overexpression after a damaging influence may be detrimental if it causes SINE mobilization and associated mutagenesis.

TEs are not always activated by stress; there are also examples of TE repression under stress conditions in different species [193]. A mechanism of stress-related L1 silencing has been described in human cell cultures. Goodier et al. reported colocalization of ORF1p foci with markers of stress granules and polyadenylated mRNA. The amount of mRNA present in these foci was low under normal conditions and increased as a result of stress induction [194]. It was later shown that artificial inhibition of stress granule formation increases L1 retrotransposition, and overexpression of the viral restriction factor SAMHD1 promotes the accumulation of L1 RNP in stress granules [195]. An example of TE silencing by histone methylation in rats after stress exposure in vivo was reported by Hunter et al.; interestingly, this silencing apparently does not apply to L1 elements [196].

### 1.8. Mechanisms and the Most Well-Known Examples of TE Exaptation

Exaptation is a shift in the function of a trait during evolution. Mobile genetic elements are essentially parasitic, but the host cell may repurpose TE regulatory sequences or RNA and protein products encoded by TEs. A frequently used metaphor for this is “domestication”. Notably, some TEs exhibit tightly regulated activity in somatic tissues, which is likely used by the host organism because it does not ensure trans-generational TE propagation [24].

TE-derived *cis*-regulatory sites are present in promoter and enhancer elements. TE sequences are found in multiple non-coding regulatory RNAs [24,197]. There are many examples of TE-derived protein-coding genes [198,199,200].

#### 1.8.1. TEs as Modulators of Gene Expression Rate

At least three mechanisms of TEs inhibiting transcription of nearby coding sequences were proposed.

Han et al. demonstrated that insertion of an L1 sequence inside a gene reduces its expression rate. They established that this is caused by poor transcription elongation of the ORF2 sequence. An intragenic L1 insertion may also cause premature polyadenylation, since L1 sequence has cryptic poly(A) signals. The authors then analyzed the human transcriptome data and showed that highly expressed genes had 4–5 times fewer L1 nucleotides within their sequence compared with poorly expressed ones [40].

Host gene expression hindering was also showed for SINEs. Estécio et al. demonstrated that B1 insertions can repress the activity of downstream gene promoters in vitro. The mechanism of repression includes acquisition of DNA methylation and loss of activating histone marks. It was established that not all genes are equally sensitive to such repression [201].

Another supposed mechanism by which TE may decrease expression rate of nearby genes is trapping TFs, since many TEs have TF binding sites [2]. However, these sites may actually be used to promote proximal gene expression (discussed in the next Section 1.8.2 and Section 1.8.3).

#### 1.8.2. TEs as a Source of Regulatory DNA Sequences

TEs rely on the host machinery for their transcription, and so they evolved regulatory sequences that mimic host *cis*-regulatory elements [12,24,197]. Analysis of chromatin accessibility maps for 41 human cell types derived from normal, embryonic, and cancer tissues showed that 44.1% of DNAse I-hypersensitive sites (DHS), representing open chromatin, overlapped with TEs (LTRs, DNA transposons, LINEs and SINEs). DHS-associated TEs were enriched in chromatin states corresponding to promoters, enhancers, and insulators, defined previously by histone marks profiling [202].

Kellner and Makałowski scanned the data from the ENCODE project for active TF binding sites located in TE-originated parts of pol II promoters. More than 35,000 promoters in six tissues were analyzed, and over 26,000 of them harbored TEs [203]. Criscione et al. identified 988 sites within the human genome that may generate chimeric transcripts originating from the L1 antisense promoter, suggesting that this promoter contributes to the transcription of up to 4% of all human genes. The authors tested four chimeric transcripts experimentally and showed that they are detectable in normal human tissues [44]. Promoters for pol III-driven transcription may also be derived from TEs. Multiple Alu repeats were shown to serve as promoters for miRNA transcription in the C19MC miRNA cluster [204].

TEs are an abundant source of tissue-specific promoters. Faulkner et al. estimated that up to 30% of human and mouse capped transcripts initiate within repetitive elements. Transcripts that arise from these sites are generally tissue-specific [46]. Mätlik et al. screened GenBank data and characterized 49 chimeric mRNAs expressed from the L1 antisense promoter and overlapping with mRNAs derived from annotated genes. For three of these, tissue-specific expression in 16 different human tissues was confirmed experimentally [205]. More recently, Roller et al. used ChIP-seq to map histone modifications associated with regulatory activity in different tissues of 10 diverse mammalian species. The authors identified multiple TE-derived promoters and showed that tissue-specific active promoters are enriched with L1 transposons. In contrast, tissue-shared active promoters were enriched with L2 LINE elements [206].

TE-derived promoters may also orchestrate stage-dependent transcription [24,207,208]. For example, mouse endogenous retrovirus type L (MERVL) controls transcription at the two-cell stage of the embryo. More than 50 genes transcribed at this stage generate chimeric transcripts linked to MERVL elements. MERVL is sharply induced at this stage and gets rapidly repressed at the next stage [115,208].

TEs contributed to the formation of new enhancers. Computational methods showed that MIR SINEs are highly enriched within putative enhancers genome-wide, that the chromatin environment of MIRs providing core enhancer sequences is similar to that of canonical enhancers, and that there are strong associations of MIR-derived enhancers with gene expression levels and tissue-specific gene expression [209]. It is also speculated that Alu elements may become new enhancers [210,211,212]. Alus upstream of transcription starting sites are enriched with epigenetic enhancer/enhancer-like marks (H3K4me1) in a tissue-specific manner, and the gain of these marks and TF binding motifs is positively correlated with the evolutionary age of Alus [212]. Canonical SVA sequence includes a known enhancer element [103].

Insulators are regulatory elements necessary for organizing eukaryotic chromatin into functionally distinct domains. Enhancer-blocking insulators prevent interactions between enhancers and promoters, and chromatin barrier insulators (boundary elements) separate active chromatin and heterochromatin. A common method to test the insulator activity of some sequences in vivo is an enhancer-blocking assay, where the tested sequence is cloned in a construct between an enhancer and a promoter that drives expression of a reporter gene. Many TE-derived insulators bind the CCCTC-binding factor (CTCF). Another insulation mechanism attributed to TEs involves binding pol III, or pol III-specific TF TFIIIC [213]. CTCF ChIP-seq data for immune cells from human and mouse showed that 70 different types of TEs have contributed strongly to CTCF binding in both species, constituting ~35% of all CTCF binding sites. TE-derived sites are overrepresented among species-specific CTCF binding sites (regardless of whether the TE insertion occurred before or after speciation) [214]. A complex mechanism of insulation involving transcription of insulating sequence has been shown for rodent B1 SINEs. While basal insulator activity is maintained by pol III transcription, induced insulation involves release of pol III and engagement of pol II transcription on the same strand. CTCF also participates in this mechanism [215]. An insulation mechanism involving TE transcription was also reported for B2 SINE elements. In the murine growth hormone (GH) locus, a B2 repeat serves as a boundary, blocking the influence of repressive chromatin modifications. *Gh* gene is initially silenced in GH-producing cells, and TF PIT-1 activates it at a specific stage of embryonic development. Both pol II- and pol III-driven transcripts of the B2 element are required to restructure the regulated locus to facilitate this activation [216]. MIR SINEs share sequence characteristics with known pol III-dependent insulators. Wang et al. predicted 1178 MIR insulators in the human genome. Only six of them overlap with previously characterized CTCF barrier elements, suggesting that the mechanism of insulation for MIR elements is CTCF-independent. Three of the predicted MIR insulators were tested in an enhancer-blocking assay, and their insulator properties were validated [213]. Another ancient transposon family with insulator activity is MER20. Targeted ChIP assays demonstrated that MER20s bind CTCF and other known insulator proteins. It was shown that insulator-type MER20s are located between differentially expressed genes in human endometrial stromal cells [217].

TE copies distributed across the genome and containing similar regulatory elements are able to establish or rewire gene regulatory networks, coordinating the expression rates of multiple genes [218]. An impressive example of this is gene regulation in mammalian pregnancy [198,217,219,220]. Some TE-derived components of transcription regulation in placenta show signs of convergent evolution in different species [221]. Another example of a transcriptional network shaped by TE-derived enhancers is interferon response; Chuong et al. showed that ERVs have dispersed numerous IFN-inducible enhancers independently in diverse mammalian genomes. Deletion of a subset of these ERV elements in HeLa cells impaired expression of adjacent IFN-induced genes [222].

Many *cis*-regulatory elements in TEs evolved to have similar activity in multiple diverse species, making TEs a promising source for the development of new expression vectors in biotechnology [223].

#### 1.8.3. Transcription Factors Binding Sites within TEs

Despite ubiquitous transcriptional repression of TEs, host TFs have been shown to directly activate expression of specific TE subfamilies, mostly relatively young retrotransposons. TEs appear to extensively contribute to mammalian transcription regulation, providing TF binding sites for a number of tissue-specific TFs [12].

Sun et al. performed a computational analysis of L1 transcriptional regulators in human cells using ENCODE ChIP-seq datasets and showed that 178 TFs are bound to L1 in at least one biological condition, with 138 TFs localized to the promoter [224]. Multiple TFs were experimentally confirmed to bind with human L1 elements and/or regulate transcription of nearby sequence: YY1, RUNX3, SOX factors [111,225], p53 [226], MYC, CTCF [224] and TCF/LEF [227]. Not all TF binding sites in L1 are located within its 5′-UTR; for example, a few overlapping SOX/LEF sites were found in ORF2 coding sequence, and these sites may control transcription of nearby genes [227]. Importantly, most of the TFs listed above are involved in neurogenesis [228]. TF binding sites may facilitate either activation or inhibition of L1 transcription; for some sites, both effects have been shown. The highly conserved YY1 binding site facilitates full-length L1 transcription, and, at the same time, is used by the host cell for L1 repression via DNA methylation. It is possible that YY1 recruits DNA methyltransferases directly [229]. Mutations in this site disrupt the accurate position of the transcription initiation complex [230]. One RUNX binding site in the human L1 5′-UTR has been proven to facilitate L1 transcription and retrotransposition [231]. There are two functional SOX binding sites in the human L1 5′-UTR, and experiments with a reporter gene system showed that different members of the SRY TF family have different effects upon binding these sites. SOX11 and SOX3 increased the reporter gene expression with different efficiencies, while SRY reduced the expression [232]. *Sox2* expression is inversely correlated with L1 expression during neuronal differentiation; a temporary decrease in *Sox2* expression is correlated with chromatin remodelling, allowing increased L1 transcription [225] (see also Section 2.4). There are controversial data about the role of p53 in L1 expression. In 2009, Harris et al. showed that the L1 promoter directly binds p53 and is positively regulated by it [226]. More recent studies brought the opposite results: p53 was shown to repress L1 transcription in various models. Apparently, p53 regulates the deposition of repressive histone marks to the L1 promoter [233,234]. One of L1 p53 binding sites responsible for L1 transcriptional repression is the same sequence that was earlier [226] reported to mediate p53-dependent transcriptional activation, suggesting that some additional factors modify the action of this site in vivo [234]. Kuwabara et al. demonstrated that some SOX binding sites within L1 overlap with TCF/LEF-binding sites involved in the canonical Wnt signaling pathway [235]. The regulation of L1 transcription by Wnt3a was confirmed experimentally [227]. Sun et al. demonstrated the involvement of Myc and CTCF in regulating L1 transcription [224].

Alu elements have dozens of predicted TF binding sites, some of these specific to certain Alu subfamilies. Most of these putative sites correspond to TFs involved in stress response or the development of specific tissues [91,236]. TFs that have been experimentally confirmed to increase transcription of Alu-containing sequences include SP1 [237], LXREα [238], estrogen receptor [239], retinoic acid receptors [240] and NF-κB [241]. As was already mentioned, SINEs are transcribed by pol III. Interestingly, binding sites of pol III-associated TFs in SINEs were shown to participate in pol II transcription upregulation. Alu sequence contains A- and B-boxes recognized by TFIIIC. Ferrari et al., studying cell cycle regulation in breast cancer cells, demonstrated that Alus located nearby pol II promoters may recruit TFIIIC without pol III or other components of the pol III transcription machinery. The authors suggested a complex mechanism where specific Alus pre-marked by the ADNP protein recruit TFIIIC; TFIIIC then acetylates histone 3 lysine-18 (H3K18ac) which increases chromatin accessibility, while also reorganizing chromatin loops by interaction with CTCF [242]. TFIIIC-dependent regulation of pol II transcripts was also shown for murine SINEs in neurons, but in this case TFIIIC is involved in transcription downregulation [243].

SVA elements contain many predicted TF binding sites: hormone response element (HRE) half-sites, a glucocorticoid response element, binding sites for SP1 [103], YY1/2 and OCT4 [244]. The ability of different parts of the SVA sequence to act as transcriptional regulators was confirmed experimentally [20,109]; however, no specific TFs were identified in these studies.

Ito et al., using ChIP-Seq datasets, identified putative TF binding sites in HERV sequences and predicted the binding of 84 TFs. Clustering analysis showed that HERV insertions can be divided into four groups: HERVs bounded to pluripotent TFs, to embryonic endoderm/mesendoderm TFs, to hematopoietic TFs, and to CTCF. It was also predicted that HERV regulatory elements tend to locate nearby and/or interact three-dimensionally with genes involved in immune responses [245]. HERV-K transcription was experimentally confirmed to be mediated by SP1, SP3 [246], YY1 and MITF-M [247], IRF1 and NF-κB [248].

#### 1.8.4. TE Insertions Modifying Coding Regions and Causing Formation of New Genes

In the overwhelming majority of cases, TE insertions affecting protein-coding regions are deleterious, since the most common results of this are frameshift or introduction of a premature stop codon. TE-encoded sequences were found only in 0.1% of functional proteins in the Protein Data Bank [249].

Nevertheless, there is a chance that TE-encoded sequence insertion within a coding region may create a new functional protein. There are so-called “orphan” genes that do not show homology to sequences in other species. The origin of such genes is often unclear, but apparently it often involves TE activity. It was shown that 53% of all orphan primate genes contain TEs. In primates, the most common mechanism of TE-mediated new gene formation is Alu exonization (see the next section), but other transposition events may also contribute to the formation of new genes [250].

Both L1 and SVA elements may bypass their own polyadenylation sites and use a downstream host site, taking some adjacent genomic sequence with them and generating readthrough transcripts that are then retrotransposed. This mechanism is called 3′ transduction. Transduced host regions may reach a few kilobases in size, and they may contain coding sequences [104]. Moran et al., using artificial transposition reporter constructs in HeLa cells, confirmed that L1 retrotransposition potentially can mediate mobilization and duplication (shuffling) of exons [251]. Pickeral et al. screened 129 full-length L1 elements and showed that at least 10% of them have a putative 3′-transduced sequence. The authors extrapolated that the total amount of DNA transduced by L1 may constitute about 1% of the human genome [252]. Notably, because most L1 retrotranspositions are 5′-truncated, it is possible that some transduction events are not accompanied by L1 sequences at all (if a truncation happened during an insertion of a hybrid sequence before its L1 part was processed), and their L1-related origin would likely be unrecognizable [251].

At least 10% of SVA elements were found to have 3′ transductions [104]. SVAs are also capable of 5′ transduction: these elements recruit external upstream promoters to be mobilized, leading to mobilization of an upstream genomic region together with the SVA sequence. SVAs may also mobilize coding sequences by “exon trapping”, if an upstream exon is spliced to cryptic acceptor sites within the SVA sequence [107,253].

#### 1.8.5. Exonization of TEs Leading to the Formation of Novel Transcripts

Exonization is a TE insertion into an intron followed by the recognition of some part of the TE sequence as a new exon. This happens relatively often because many TEs carry potential splice sites [11,254]. An indirect proof that exonization is the most common mechanism of TEs entering coding sequences is the fact that the majority of TEs found in protein-coding regions are fragmented. TE-derived novel exons are usually “cassette exons” that may be skipped as a result of splicing. Most TE-derived exons are included only in minor splice isoforms. In these cases, exonization does not destroy the function of the gene [2,255]. Analysis of recently formed coding cassette exons showed that about 80% of them create frameshifts or introduce premature stop codons [256]. Single-base mutations may change an alternative TE splice site into a constitutive one. A few hereditary diseases (Alport syndrome, mucopolysaccharidosis type VII, CCFDN syndrome) have been associated with constitutive inclusions of TE-derived exons [93]. Nevertheless, a substantial number of recently formed constitutive exons overlap with repetitive elements in the human genome [257].

Exonization mechanism is well-known for Alu elements. Their sequence contains multiple cryptic splicing sites, requiring a few mutations to become functional [2,11,93,256]. However, only two of these sites are usually selected in Alu exonizations. Most exonizations involve the right arm of Alu in antisense orientation [254,258]. Interestingly, Alu insertion into an exon may cause formation of a new intron (intronization) [259]. Using multiple genome comparisons, Zhang and Chasin found that 62% of recently evolved (primate-specific) cassette exons and 28% of recently evolved constitutive exons overlap with Alu elements [257]. Mandal et al. analyzed the complete human transcriptome and showed that 12.7% of all transcripts have Alu exonizations. Of these, 80% of transcripts have Alu exonizations within 3′-UTR [260]. A notable example of a gene harboring an Alu exonization is *ADAR2* encoding an enzyme performing RNA A-I editing (this mechanism is discussed in the Section 1.8.7). In human *ADAR2*, exon 8 is a new Alu-derived exon that is alternatively spliced in high inclusion levels in a tissue-specific manner. The new exon is inserted in the catalytic domain of ADAR2 and alters its catalytic activity. Interestingly, a mouse-specific exonization occurred at the same position in the mouse *Adar2* gene [256].

The L1 sequence contains a few functional splicing sites [261], and L1-related exonization events were reported in mouse [262] and human [263] transcriptomes.

Comparative genomic analysis showed that an ancient LF-SINE element provided sequence for 19 different coding exons in tetrapod genomes (in the human genome, an LF-SINE-derived exon is found within *PCBP2* gene). All affected proteins have different functions and belong to different protein families. A total of 16 out of 19 LF-SINE-related exons are alternatively spliced; 11 of them introduce an early stop codon. For 3 of these exons, a regulatory function was suggested [264].

#### 1.8.6. TEs Introducing Alternative Polyadenylation Sites

Many retroelements have an A-rich tail, and a single point mutation in it may produce the polyadenylation signal, AATAAA. Thus, retroposition downstream of the coding region of a host gene could be a source of an alternative poly(A) signal. However, poly(A) signal is not sufficient to direct strong polyadenylation by itself; the poly(A) site region contains other *cis*-elements involved in the recognition of poly(A) signal by the polyadenylation complex [265]. Such elements may also be possibly introduced by a TE insertion. Over half of all human genes have multiple poly(A) sites. A total of 3188 human poly(A) sites from 2565 genes (corresponding to ~8% of all poly(A) sites and ~16% of all genes surveyed) were found to be associated with TEs. Notably, ~94% of human TE-associated poly(A) sites are nonconserved in mouse. Intragenic poly(A) sites associated with TEs are less frequently utilized compared with conserved sites at 3′ ends of transcripts. Different TE classes are associated with poly(A) sites differently: most DNA transposons and LTRs contain a whole poly(A) site region, while many LINEs and SINEs are located either upstream or downstream of poly(A) sites, suggesting the contribution of *cis* elements [266]. The role of Alu elements in generating new poly(A) sites seems to be especially important. There are 107 Alu-derived poly(A) sites in the human genome; importantly, almost half of them constitute major or even unique poly(A) sites in a gene [267].

#### 1.8.7. RNA A-I Editing

A very important RNA modification mechanism is adenosine-to-inosine RNA editing (A-I editing). ADARs (adenosine deaminases that act on RNA) target double-stranded RNA without any apparent sequence specificity and deaminate adenosines to produce inosines (recognized by most RNA-interacting molecules as guanosines). A-I editing may alter the sequence of a protein translated from an affected mRNA and/or create a new splicing site. Moreover, “hyperedited” mRNAs that contain many inosines are retained in the nucleus, providing a specific mechanism of gene silencing [11,268,269,270]. Alu sequence was identified as the primary target of A-I editing. By recent estimations, Alus account for >99% of A-I editing events found in humans [271]. Two nearby Alu repeats in opposite orientations easily anneal with each other, forming an extended region of dsRNA recognized by ADARs [11]. Additionally, it was found that editing of non-Alu sites is dependent on nearby edited Alu sites in the human transcriptome [272,273]. In mammals that do not have Alus in their genomes, other TEs serve as A-I editing targets (in mice, most A-I editing sites are distributed between B1 and B2 SINEs, L1 LINE, and MaLR LTR elements) [274]. Most mRNAs that undergo A-I editing are expressed in the nervous system [275] and are discussed in detail below, in the Section 2.3. Interestingly, ADAR2 mRNA is also subject to A-I editing, providing an example of auto-regulation [276,277].

#### 1.8.8. Other TE-Related Mechanisms Altering the Fate of Transcripts

Back-splicing (circular splicing) is an unusual splicing mechanism that generates a circular RNA (circRNA) and a linear RNA with exon exclusion. It was recently discovered that circRNAs can bind miRNAs and regulate mRNA splicing and transcription [278,279]. Alu elements may facilitate circularization of pre-mRNA and back-splicing because two Alus in an opposite orientation on different ends of a pre-mRNA may easily form an imperfect duplex with each other [2]. Indeed, bioinformatic analysis revealed that circularized exons usually have long bordering introns containing complementary Alu repeats [280].

*Trans*-splicing happens when the splicing machinery joins splice donor and acceptor sites from two different transcripts. Clayton et al. proposed a potential mechanism of TE sequences facilitating *trans*-splicing events by providing complementary sequences for hybridization between pre-mRNAs from different loci. It has not yet been confirmed experimentally [281].

TE sequences within an mRNA may affect its degradation rate by different mechanisms. The poly(T) sequence in antisense Alu elements is the source of ~40% of identified 3′-UTR AU-rich elements (AREs), which regulate mRNA half-life through the competitive binding of proteins that stabilize or destabilize the transcript. Alu elements can also mediate intermolecular base pairing between two RNA molecules. The resulted dsRNA structures may be subject to Staufen-mediated mRNA decay [2]. Another mechanism by which TEs regulate mRNA degradation rate is miRNA-mediated decay and/or translational repression of mRNAs. The most common miRNA target site coincides with the most evolutionary conserved part of Alu [282].

TE-derived RNAs may compete for regulatory molecules with host mRNAs. At least one TE-rich transcript was confirmed to function as an “miRNA sponge” participating in complex gene network regulation [283].

TE insertions may provide alternative translation starts. For example, the insertion of an LTR element at the 5′ end of the *MTH1* (*NUDT1*) gene generated three additional upstream in-frame start codons [284].

#### 1.8.9. TE-Encoded Proteins and Non-Coding RNAs Used by the Host Cell

Proteins encoded by TEs are able to bind, cut, ligate or degrade nucleic acids, as well as interact with other proteins, and all these functions may be used by the host cell as well [3,34,198]. It was estimated that the human genome includes over 40 genes derived from DNA transposases [285] and at least 85 genes derived from the *gag* gene of LTR elements [198]. We will list some specific TE-related genes below.

A famous example of TE-derived protein-coding genes are the genes encoding the RAG1 and RAG2 recombinases, which catalyse the V(D)J somatic recombination in lymphocytes, generating a highly diverse repertoire of antibodies, immunoglobulins and T-cell receptors in vertebrates. The *RAG1* gene has been derived from a *Transib* DNA transposon [286], and then its recombination activity was enhanced by the capture of a pro-*RAG2* sequence [287,288]. In the context of the present review, it is interesting that RAG1 and RAG2 were shown to be expressed in neurons [289,290,291], even though the exact function of these recombinases in neurons is not clear.

The human *CENPB* gene encoding centromere-associated protein B is related to the *pogo* superfamily of DNA transposons [292,293]. CENP-B recognizes and binds a specific DNA sequence (CENP-B box). CENP-B lost its capacity to act as a transposase, although it apparently retains nickase activity, and this might contribute to the high-order structure of the centromeres by promoting recombination hotspots [294].

Syncytins involved in the development of placenta are encoded by ERV *en*v-derived genes. Human syncytins-1 and -2 are essential for the formation and function of syncytiotrophoblast, the boundary layer between maternal and fetal tissue [295,296]. Syncytiotrophoblast is a fused cell layer, and Env proteins have membrane fusogenic capacity [296]. A conserved *env* immunosuppressive domain in syncytins is important for maternal immune tolerance during pregnancy [297]. Notably, domestication of *env* genes from ERVs of different families generating syncytin-like genes has occurred independently in different mammalian lineages [296,298].

Other TE-derived protein-coding genes expressed in the placenta and necessary for its normal development belong to the *MART* family (mammalian retrotransposon transcripts; another name of this family is *SIRH*, *sushi-ichi*-related retrotransposon-homologs). *MART* genes (at least 11 in placental mammals) are derived from the vertebrate Ty3/*gypsy* LTR family called *Sushi* (more specifically, they show similarity to the *sushi-ichi* element). *MART* genes are incapable of mobilization, having lost some structural features such as LTRs; the *pol* region is also completely or partially deleted in these genes. However, all *MART* genes carry a protein-coding region derived from the *gag* region of their ancestral retrotransposon [198,299,300]. It must be noted that genes of this family have many synonymous names used in different publications, which may be confusing. Some important examples of *MART* genes with placental function are *PEG10* (*RTL2*, *MART2*, *SIRH1*), *RTL1* (*PEG11*, *MART1*, *SIRH2*) [301], and *SIRH7* (*LDOC1*, *MART7*, *RTL7*) [302]. One gene of this family, *RTL4* (*MART4*, *SIRH11*, *ZCCHC16*) is expressed in the brain [303]; its possible function is described below in the Section 2.7.2.

New protein-coding genes can also be generated by a fusion between host and TE coding sequences. An example of this is the *SETMAR* gene encoding a protein involved in nonhomologous end-joining (NHEJ) DNA repair [304]. This chimeric gene is a result of fusion of the pre-existing *SET* gene with the *Hsmar1* DNA transposon [305].

Long non-coding RNAs (lncRNAs) are structurally similar to mRNAs but are not translated, instead participating in gene expression regulation. In striking contrast with mRNAs, most lncRNAs include TE-derived sequences [306,307]. HERV-H is a TE family particularly enriched in long intergenic non-coding RNAs (lincRNAs). A total of 127 HERV-H-lincRNAs exhibit dramatic stem cell-specific expression [306].

TEs have also provided coding sequences for multiple small silencing RNAs. piRNAs and siRNAs derived from TEs have been discussed above in the Section 1.7.4. miRNAs may originate from TEs as well. Piriyapongsa et al. identified 55 experimentally characterized human miRNA genes that are derived from TEs. L2 LINEs, MIR SINEs and DNA transposons were enriched among TE-derived miRNA genes. One notable TE-derived miRNA is hsa-mir-566, which is Alu-related and has 1184 predicted target sites [308].

In a recent work by Percharde et al., an important role for the host cell gene expression regulation during embryonic development was confirmed for L1 RNA. This RNA is highly expressed and located in the nucleus in mouse ESCs. Surprisingly, KD of L1 RNA results in a dramatic decrease in ESC self-renewal, reduction in total RNA levels, and reduction of rRNA synthesis. L1 KD ESCs showed upregulation of many genes normally expressed specifically at the two-cell stage (2C); the most important of these genes is *Dux* encoding a master TF. It was shown that L1 RNA interacts with *Dux* and rDNA loci. Experiments in vivo confirmed the importance of L1 RNA for normal embryonic development: the development of mouse embryos with L1 KD was arrested at the 2C. Thus, L1 expression is necessary for the exit from the 2C by repressing the transcription of genes controlled by Dux and activating rRNA synthesis to support rapid proliferation [115].

Even though ERV interactions with host immunity are commonly associated with pathologies [309,310,311], ERV-expressed products may also be used by host organisms to tune immune response mechanisms. While ERV proteins are presented to developing lymphocytes among other “self” peptides, their low expression in healthy tissues due to silencing leads to incomplete immunological tolerance to ERV-derived antigens. By mimicking viral infection, endogenous retroelements may thus provide “intrinsic adjuvant” for the immune response to poorly immunogenic targets [26]. In specific cases of cancer, overexpressed ERVs may serve as tumor-specific antigens recognized by the immune system, thus helping to regress tumors [295]. In addition, antisense ERV transcripts have been proposed to hybridize homologous RNA sequences of exogenous retroviruses, forming dsRNA molecules recognized by innate immunity machinery [297].

#### 1.8.10. TEs Participating in DNA Repair and Chromosome Maintenance

The interplay between TE activity and DNA-damage repair systems may be very complicated. While a TE insertion is a kind of DNA damage itself, so the host DNA repair systems must actively prevent it, it also seems logical that during the long co-evolution of TEs and their host genomes, mobile elements that are able to insert themselves more or less anywhere in the host DNA should eventually be used by the host cell to “patch” some damaged genome regions.

Noncoding parts of chromosomes that only have a structural function also get damaged, and restoring these using TE machinery would be a much simpler task for TEs than the repair of a coding sequence, since the exact sequence of inserted DNA would not be as important. Indeed, TE machinery is used to maintain telomeres. It remains debatable whether telomerases are domesticated retrotransposons; however, their phylogenetic and functional relation to TEs is clear [312,313,314]. The catalytic core of telomerase consists of the telomerase reverse transcriptase (TERT) and the integral telomerase RNA (TR). TR serves as a template for RT. Therefore, TERT, just like L1 ORF2p, is a reverse transcriptase that carries a specific RNA [315,316,317,318]. The amino-acid sequence of telomerase catalytic subunits is similar to that of viral reverse transcriptases [318]. Curiously, there are reports about telomerase being capable to reverse transcribe telomere RNA to internal chromosomal sites, a process quite similar to retrotransposition. Telomerase was also shown to bind RNAs different from TR [79,319]. The TERT-dependent telomere maintenance mechanism is conserved in most eukaryotes; however, dipteran genomes lack the *TERT* gene [320]. *Drosophila melanogaster* cells rely on three retrotransposons (HetA, TART, and TAHRE) to maintain telomeres. Therefore, in this species, not only TE-derived enzymes but also TE DNA are used for telomere elongation [79,314]. TE function in maintaining telomeres has also been suggested in some other eukaryotes [321,322]; on the other hand, even other species of the genus *Drosophila* have different TERT-independent telomere elongation mechanisms that rely on TE insertions to various degrees [323].

There are multiple reports about TE insertions at DNA break sites in coding regions, and in some cases these insertions contain retrotransposed host mRNA sequence. However, the possibility of TEs providing means for the accurate repair of genes is still unclear. RNA-templated DNA repair is a recently discovered repair mechanism [316,324]. It is apparently coupled with transcription [325,326], but the key process in this mechanism must be RT, and currently it is not well understood how it happens. RT may possibly be performed by DNA-dependent DNA polymerases [327], and the role of such a mechanism in DNA damage repair was proposed [328]. However, mammalian cells express three specialized reverse transcriptases: TERT [315], L1 ORF2p [34] and the enzyme encoded by ERV *pol* gene [3]; all these enzymes have a TE-related origin.

Probably the earliest study of TE-mediated DNA repair was performed in 1996 on yeast transformed with plasmids encoding various TE-encoded reverse transcriptases, including the L1 enzyme [329]. A more recent study showed that an endogenous mechanism of RNA-dependent DNA repair exists in yeast, that it normally happens using a cDNA intermediate, and that the reverse transcriptase of the yeast transposone Ty performs the RT [330]. Ono et al. introduced DSBs into mouse zygotes and NIH-3T3 cells by the CRISPR/Cas system and identified long de-novo insertions at DSB-targeted sites. It was shown that DSBs are repaired by sequences deriving from retrotransposons, genomic DNA, mRNA and sgRNA (single guide RNA used in the CRISPR/Cas method). Some insertions derived from intronless mRNA sequences, indicating the involvement of RT. RT inhibitors prevented mRNA- or retrotransposon-derived insertions into the DSB sites [331].

Canonical homologous recombination (HR) DNA repair mechanism uses a sister chromatid as a matrix for repair, and so HR only works during S or G2 cell cycle phases [332]. Wei et al., using U2OS osteosarcoma cells in G0 or G1 phases, showed that HR factors, including RAD52, are recruited to transcriptionally active damage sites. The authors suggested that RNA may be used as an HR template instead of DNA, and experiments with RNase H treatment confirmed that RNA is required for this mechanism [325]. This suggests that one of the TE-encoded reverse transcriptases may be involved in such repair. The same research group then discovered a similar repair mechanism in neurons [333] (discussed in the Section 2.6).

Human L1s were shown to be able to integrate at endogenous DNA lesions in cells defective in the NHEJ repair pathway; experiments with mutant L1 forms confirmed that such integration is endonuclease-independent (ENi) but needs L1 reverse transcriptase function to be intact. ENi L1 retrotransposition was demonstrated to generate unusual insertions: L1 copies inserted at preexisting DNA lesions were truncated predominantly at their 3′ ends, lacked target site duplications, and two out of nine examined insertions contained host cDNA fragments at their 3′ ends [64]. The same research group later showed that ENi L1 retrotransposition can use dysfunctional telomeres as integration sites [79,317]. To check whether ENi L1 integration is possible in vivo, Sen et al. scanned the human genome and identified 21 young non-classical L1 insertions (NCLI) distinguishable from TPRT-mediated ones. A total of 15 out of 21 NCLI loci involved additions of non-L1 DNA segments (up to 312 bp) to the L1 DNA. At two NCLI loci, fragments of cellular RNAs appeared to have been reverse transcribed along with the L1 RNA [61]. Similar observations were made later for Alus (by the same research group). A total of 23 non-classical Alu insertions (NCAI) that have signs of ENi integration were found. Out of these, 13 NCAI loci included non-Alu DNA sequences (the largest one being about 2 kb), and 2 NCAI loci had signs of cellular RNAs apparently being transcribed along with the Alu fragment. In 5 NCAI loci, capture of another TE RNAs along with the main Alu sequence was detected [334]. It must be noted that the methods used in the two latter papers only allowed to reveal germline non-classical TE insertions. We may speculate that in somatic cells ENi insertions are more common.

## 2. The Role of TEs in the Normal Function of Neuronal Tissue

Both germline and somatic TE insertions were found to have functional roles in neurons and neural precursor cells (NPCs). These mechanisms are summarized in Table 2.

### 2.1. Somatic Mosaicism in Neurons: Its Sources and Possible Functions

Somatic mosaicism is a feature of a normal brain [72,275,335,336,337,338,339]. Neurons of the human brain are extremely diverse. Even compared with the immune system, cellular diversity in the nervous system is estimated to be twice as high. Some classifications suggest that there might be up to 10,000 different types of neurons. Neurons with similar morphologies may differ in gene expression and function. Neuronal diversity is generated by a combination of mechanisms acting on different levels: DNA (mutations and different activity of promoters), RNA (alternative splicing and polyadenylation, RNA editing) and proteins (post-translational modifications). At least some of these mechanisms, such as RNA editing and alternative polyadenylation, seem to be more common in the CNS than in other tissues [275]. In this section, we will discuss neuron-specific genomic mutations.

Neurons in general have the highest quantity of “acquired” DNA compared to other cells. Frontal cortex neurons were found to have about 4% more DNA in their genomes compared to lymphocytes. In the same study, it was shown that neurons in the frontal cortex have more DNA compared to the cerebellum of the same brain [340]. A single neuron may contain in its genome at least 800 single-nucleotide variants (SNVs) [341] and 1–2 mutations caused by random L1 integrations [342]. Single-cell sequencing of human frontal cortex neurons revealed that 13–41% of them have megabase-scale de-novo copy number variations (CNV), with deletions being twice as common as duplications. A subset of neurons have highly aberrant genomes with multiple alterations [343]. A noticeable fraction of neurons in the normal brain even have aneuploidy. When specific chromosomes were labeled in adult brain samples by in-situ hybridization, it was found that about 1% of murine cortical neurons have loss or gain of X or Y chromosomes [344], while, in the human brain, chromosome 21 aneuploid cells (including non-neuronal cells) constitute ~4% of all cells [336]. Importantly, experiments with mice demonstrated that aneuploid neurons are not dying cells, but are actually functionally active and integrated into the normal brain circuitry [339].

Neuronal genomic diversity is caused by multiple factors. Neurons are the most long-living cells in the human body [345]; therefore, during their life, multiple mutations inevitably accumulate in their genomes. However, it is important to specify that many neuron-specific mutations actually happen very early, even before the full differentiation of the cell. NPCs undergo multiple double-strand breaks and genomic recombinations (with some long genes being particularly affected) [346,347]. Spectral karyotype analysis demonstrated that a striking 33% of proliferating neural precursors in the brain of a normal mouse embryo have aneuploidy (most of these cells lacked one chromosome). Apparently, most of these aneuploid cells undergo apoptosis, since the level of aneuploidy in adult mouse cortex was much lower [344]. Indeed, it was shown that during embryonic neurogenesis only 15–40% of post-migratory cells survive, suggesting that some kind of selection takes place [275]. In mature neurons, one of the most important causes of mutations is activity-induced DSBs. DSBs happen in neurons as part of their normal function [348]. Such DSBs are not random; they form at specific locations in the genome. DSBs in the promoters of a subset of immediate early genes are necessary to trigger their fast expression by resolving topological constraints [349].

### 2.2. Transposable Elements Are an Important Source of Neuronal Genetic Mosaicism

The brain is the only known somatic tissue where retroelements are de-repressed throughout the life of a healthy human [350]. There are multiple reports [7,39,47,72,174,225,351,352,353,354,355] about NPCs and neurons being more permissive to retrotransposition than other types of cells in mammals (only embryonal [351] and tumor cells [356] have retrotransposition rates comparable to that in neural tissue). These reports are described at length below.

It must be considered that even if transposition were equally frequent in all cell types, neurons would likely still be the most affected (in the sense of TEs having an effect on cell function). Firstly, neurons express a larger fraction of their genome compared to most other cell types. The initial estimates were that up to 50% of all protein-coding genes in the human genome are expressed in the brain (with only kidney and testes having comparable transcriptome complexity) [357]. However, more recently, it was confirmed that up to 95% of protein-coding genes in the genome are expressed in at least one human brain region during at least one period of development or adulthood [358,359]. Secondly, many long genes are expressed in the brain or involved in neuronal-specific processes [7,346]. Recent studies have demonstrated that transcripts of genes longer than 100 kb are highly enriched in neurons [360,361,362]. Long neuronal genes have an increased susceptibility to random TE insertions simply because of their size [7]. Interestingly, *NF1*, a very rare example of a human gene containing specific hotspots for L1-mediated insertions, is quite long, about 280 kb [363]. While *NF1* is not strictly neuron-specific, it has distinct functions in neurons [364]. TE insertion hotspots within *NF1* were associated with a disease [363] described in the Section 3.2.18.

TE activity (L1 transposition in particular) is increased in neural tissue. Experiments with the L1-EGFP retrotransposition reporter demonstrated that engineered human L1s can retrotranspose in adult rat hippocampus NPCs in vitro and in the mouse brain in vivo [225]. Later, L1 mobilization was similarly confirmed in NPCs isolated from human fetal brain and NPCs derived from hESCs [352]. Macia et al. found that L1 expression and engineered retrotransposition is much higher in NPCs than in mesenchymal (MSCs) and hematopoietic (HSCs) somatic stem cells. They have also demonstrated for the first time that engineered L1s can mobilize in mature, non-dividing neurons [354].

Similar observations were made for endogenous L1 activity in vivo. Coufal et al. detected an increased copy number of endogenous L1s in several regions of adult human brain compared with heart and liver samples [352]. Similar results were obtained by Jacob-Hirsch et al.: whole-genome sequencing of human samples (including different brain regions, peripheral blood, and lymphoblastoid cell lines) showed that the number of retrotranspositions in brain tissues is higher than in non-brain samples. Most identified somatic brain transposition events are L1 insertions. Interestingly, the majority of these insertions were shown to be nested in pre-existing L1 elements. These nested insertions usually had signs of ENi transposition [39]. On the protein level, ORF1p expression in different brain regions was confirmed by immunohistochemistry of postmortem brain samples. Notably, kidney, liver, lung and heart samples showed little to no ORF1p expression in this study [365].

A number of studies using single-cell sequencing approaches have confirmed that among all somatic transposition events in the human brain, L1 insertions and L1-mediated deletions are the most frequent. However, due to methodological variation, the estimated rate of L1 insertions in these studies varies considerably: from one insertion per 25 neurons to 13.7 insertions per a single neuron [70,353,366]. Importantly, experiments that calculated the lower bound [366] have higher rates of validation. Even with the lowest estimate, considering the enormous total number of neurons constituting the human brain, there must be tens of billions somatic retrotransposition events within an individual brain. In addition, taking into account the highly networked structure of the brain, a mutation in a single neuron may potentially alter the function of many other neurons, causing far-reaching effects on neuronal circuitry [7,116]. Indeed, it was found that severe neuropathologies may be caused by some somatic mutations with low levels of mosaicism (mutation detected in 10% of blood cells, or as low as 8% cells in the brain samples). This may be explained by cell migration during the brain development. For example, cerebral cortical progenitors generate cells that migrate to the outer layers of the cortex, so cortical neurons with the same somatic mutation would likely be intermingled with neurons without this mutation. Importantly, glial cells also have complex migration patterns [338]. It must be noted that the reports about endogenous L1 activity in glial cells are somewhat controversial [116]—while some studies showed that glial cells harbor less L1 insertions compared to neurons [225,353], other authors found no significant difference between the number of somatic L1-associated variants (SLAVs) in these cell types. SLAVs include both L1 insertions and retrotransposition-independent L1 endonuclease-mediated mutations; surprisingly, the latter happens about as often as the former [70].

It was found that TE insertions in the brain are not only common, but also prone to target neuron-specific genes. Baillie et al. developed a high-throughput protocol called retrotransposon capture sequencing (RC-Seq) to map individual retrotransposition events. The authors applied this method to identify TE insertions in human hippocampus and caudate nucleus samples. It was demonstrated that TEs mobilize to protein-coding genes that are differentially expressed and active in the brain. Protein-coding loci were disproportionately affected by L1 and Alu insertions compared to random expectation and compared to prior germline frequencies. Genes containing somatic intronic L1 insertions were twice as likely to be differentially overexpressed in the brain compared to random expectation. Loci with such insertions were associated with neurogenesis and synaptic function: among them were tumor suppressor genes deleted in neuroblastoma and glioma, genes encoding dopamine receptors and neurotransmitter transporters. Somatic Alu insertions were not overrepresented in introns but were even more common in exons than somatic L1 insertions [175].

New approaches also allowed researchers to pinpoint specific TE variants in the brain. Sanchez-Luque et al. performed a single-cell genomic analysis of human hippocampal neurons, identified and validated one specific somatically active hot donor L1 element. This particular L1 lacked the YY1 binding site and was highly mobile when tested in vitro [229].

Not only mammalian brains have been shown to contain genetically diverse neurons with relatively high TE activity. Perrat et al. performed cell-type specific gene-expression profiling of *Drosophila* neurons and demonstrated that TE expression is more abundant in mushroom body αβ neurons than in neighboring neurons. Deep sequencing identified over 200 de-novo TE insertions in αβ neurons [367]. However, a more recent study found no evidence for TEs being mobile in the fly brain. Even though increased TE expression in the αβ neurons was confirmed, TE sequences were not found to accumulate in neuronal genomic DNA with age [368], making the subject of TE-related neuronal mosaicism in the fly brain controversial.

### 2.3. RNA A-I Editing in TE Sequences Contributes to Neuronal Somatic Mosaicism on RNA Level

The A-I editing described in the Section 1.8.7 has been confirmed to supply an additional level of functional modulation to neuronal genes. A-I editing introduces codon changes in mRNAs and may generate specific protein isoforms that do not correspond to genomic DNA sequences. Disruption of this mechanism was associated with multiple disorders, primarily neurological or psychiatric [275,369,370,371,372]. Mammals have three A-I editing enzymes; ADAR3 is restricted to the brain, while ADAR1 and ADAR2 are preferentially expressed in the nervous system [373]. A-I editing is subject to complex spatiotemporal regulation within CNS: different regions of the brain show up to three-fold differences in A-I editing levels [11], and A-I editing efficiency for both coding [374] and non-coding RNAs [375] increases during murine brain development. Brain-related genes in the human genome are enriched with intronic Alus and have an increased propensity for mRNA editing [7]. Most targets for A-I editing are found in the nervous system, and these mRNAs encode such crucial neuronal proteins as voltage-dependent potassium channels [370,376], various glutamate receptor subunits [268,275,370,376], GABA receptors [7], and serotonin receptors [275,369,370]. The lack of A-I editing of mRNA encoding the AMPA receptor subunit GluA2 in mice causes drastic phenotype changes, with severe epilepsy and premature death within 3 weeks after birth. Mutated receptors have increased rate of tetramerization and provide increased Ca^2+^ influx [268,370,376]. Remarkably, the defects of the ADAR2 KO animals are completely rescued by replacing both WT GluA2 alleles with ones encoding the edited codon, even though A-I editing of many other sites is not restored [268]. In humans, deficient GluA2 RNA A-I editing was linked to spinal motor neuron degeneration in amyotropic lateral sclerosis [377].

### 2.4. TEs and Cell Differentiation in Neurogenesis

In different works, the L1-EGFP reporter system has been shown to readily mobilize in NSCs, NPCs, and post-mitotic neurons [116]. However, L1 derepression seems to be specifically associated with neuronal differentiation.

Kurnosov et al. applied next-generation sequencing to directly compare the rate of L1 and Alu somatic insertions in neurogenic and non-neurogenic regions of human adult brain. The percentage of somatic L1 insertions from the total number of reads in the corresponding library was approximately equal for all studied regions except the neurogenic dentate gyrus (DG), where it was about 1.5 times higher. The highest percentage of somatic Alu insertions was also observed in the DG. In this region, Alu insertions in genes happened more often than predicted, while Alu insertions in promoters had lower than the expected rate. Another neurogenic region, the subventricular zone (SVZ), was not found to have unusual TE insertion distribution like the DG; however, there is some data that in humans, unlike rodents, the neurogenesis in the adult SVZ is negligible [174].

L1 disinhibition happens during neuronal differentiation in vitro. Erwin et al. performed artificial differentiation of hESC-derived hippocampal NPCs and showed highly upregulated L1 ORF2 expression during initial stages of neuronal differentiation [70]. Muotri et al. transfected rat hippocampus neural stem (HCN) cells with an L1-EGFP retrotransposition indicator construct with a puromycin resistance gene. Selected puromycin-resistant clones did not express EGFP at the onset of the differentiation experiment. The authors selectively stimulated HCN differentiation into neurons, astrocytes or oligodendrocytes. EGFP expression was registered only during neuronal, but not glial differentiation. EGFP could be detected as early as 2 h after the induction of neuronal differentiation, indicating that it might have resulted from the derepression of already integrated but silenced *EGFP* rather than from additional retrotransposition events. The authors then performed a mixed differentiation experiment and discovered that most HCN clones harboring detectable L1 retrotransposition events had a tendency to differentiate into neurons rather than glial cells. L1 activation in HCN cells correlated with decreased expression of *Sox2*. Moreover, recruitment of SOX2 and HDAC1 on the endogenous rat L1 promoter region in undifferentiated HCNs was confirmed by by ChIP-PCR. The authors suggested that SOX2 represses L1 prior to neuronal differentiation, and release of SOX2 leads to restructuring of chromatine, allowing L1 expression. It was also noted that some of the L1 insertions in the selected HCN clones occured within neuron-specific genes [225]. In their consequent work, Muotri et al. showed that the L1 promoter undergoes demethylation during neuronal differentiation. ChIP-qPCR showed high levels of MeCP2 in association with endogenous L1 promoter regions in rat NSCs compared to neurons generated from these cells. Specific CpG sites within the L1 promoter were found to demethylate during neuronal differentiation. MeCP2 was also associated with ORF2 but this association did not change during differentiation [130].

Some data suggest that L1 derepression during neuronal differentiation reflects L1 *cis*-regulatory sites being used to promote transcription of nearby genes. Jönsson et al. performed a KO of *DNMT1* gene in fetal-derived human NPCs to cause global DNA demethylation. This resulted in transcriptional activation of evolutionarily young, hominoid-specific L1 elements. The authors also established that protein-coding genes located within 50 kb of activated L1 elements were upregulated; most of these genes are normally upregulated upon neuronal differentiation of NPCs. Gene Ontology analysis confirmed that these genes were enriched for functions such as synaptic transmission and cell communication. Many of the activated L1 elements gave rise to fusion transcripts, confirming their role as promoters [378]. Kuwabara et al. discovered that L1 elements in rat adult hippocampal NSCs are subject to regulation by the same TFs as NeuroD1 (a proneural TF essential for the CNS development and adult neurogenesis). There are a few SOX2 and TCF/LEF binding sites within the *Neurod1* promoter, including overlapping SOX/LEF-binding sites that may bind both of these factors. SOX2 and SOX/LEF binding sites are necessary for expression downregulation by FGF2, while TCF/LEF and SOX/LEF binding sites are necessary for expression upregulation by Wnt3a. Importantly, the L1 sequence also contains multiple SOX/LEF-binding sites, not only in its 5′-UTR but also in the ORF2 coding region. In reporter gene assays, the ORF2 sequence demonstrated promoter activity in both forward and reverse orientations in cultured NSCs undergoing neuronal differentiation; this activity was the highest 1 d after neuronal induction. L1 promoter activity in differentiating neurons was then confirmed in vivo in an experiment with a lentiviral vector encoding ORF2 fragment fused with *EGFP* gene. This virus was injected into the DG of young adult rats. Notably, EGFP expression observed after the infection was restricted to neurogenic areas, and EGFP-positive cells contained markers of NPCs and newborn neurons. These data indicate that both NeuroD1 and L1 expression are specifically induced during NSCs transition to newborn neurons. The authors scanned human, mouse and rat genomes for L1 elements near genes and identified a few genes expressed in neurons that may possibly be subject to transcription regulation by Sox/LEF sites within proximal L1 sequences [227].

Importantly, some specific TEs need to be downregulated during neuronal differentiation [32]. Primate-specific transcription repressors ZNF417 and ZNF587 silence HERV-K and SVA elements in hESC. Neurons derived from ZNF417/587-depleted iPSC were characterized by the aberrant expression of nonneuronal genes and upregulation of transcripts related to potassium channel activity or to GABAergic neurotransmission. Another important observation in this study is that six of HERV-K elements predicted to encode neurotoxic Env proteins were upregulated in iPSC-derived neurons as a result of ZNF417/587 KD [379]. Accordingly, it was shown that downregulation of one specific *env* gene within an evolutionary young HERV-K subtype HML-2 in iPSC resulted in the dissociation of stem cell colonies and enhanced differentiation along neuronal pathways [380]. The same HERV-K subtype was studied by Padmanabhan Nair et al. They used a CRISPR-based approach to directly upregulate several distinct HML-2 promoters in hESCs. The authors differentiated transgenic and control hESCs into cortical neurons and estimated their properties at day 60. Neurons produced from cells with activated HML-2 transcription exhibited altered morphology and a drastic reduction in levels of MAP2a/b (neuron-specific cytoskeletal proteins). Forebrain organoids generated from transgenic hESCs showed a disrupted layer structure and were smaller in size compared with organoids produced from control hESCs. RNA-seq showed that HML-2 overexpression upregulates neuronal development-related genes in cortical neurons. Interestingly, dopaminergic neurons generated from control or transgenic hESCs had no difference in morphology or MAP2 expression at day 50 [381]. Skariah et al. confirmed the importance of Mov10 RNA helicase in brain development. They found that Mov10 level is elevated in the postnatal mouse brain and that Mov10 KO has an embryonic lethal phenotype, while heterozygous *Mov10*^+/−^ mice have disrupted dendritic arborization, increased anxiety and hyperactivity. Sequencing experiments showed that Mov10 binds multiple RNAs involved in neuronal development and neurite outgrowth. However, Mov10 is also a strong suppressor of L1 mobilization. In brain samples from *Mov10*^+/−^ mice (postnatal day 2), ORF2 genomic copy number was two times higher than in WT samples [172].

Even though most studies of TE derepression in neuronal tissue cited above either focused specifically on NPCs or did not discriminate between neurons and NPCs, a few recent works showed that postmitotic neurons permit L1 expression and/or mobilization. Macia et al. reported mobilization of engineered L1s in neurons [354]. Salvador-Palomeque et al. measured L1 promoter methylation at multiple points of neurodifferentiation in cells generated from a cultured hiPSC line. Surprisingly, a reduction in methylation was observed in day 112 neurons compared to day 72 neurons. (However, the methylation level was increased again in day 156 neurons [355].) Erwin et al. differentiated hESCs to hippocampal neurons and measured endogenous ORF2 expression at different stages. ORF2 level was low in NPCs, then it increased at days 1–2 of in vitro differentiation, diminished by day 5 and peaked in post-mitotic neurons (day 28) [70].

### 2.5. Specific TE Regulation in the Hippocampus: Is There a Role for TEs in Learning and Memory?

The hippocampus is the key brain region participating in learning and memory that also contains a site of adult neurogenesis [382]. This region also plays an important role in brain reaction to stress and early-life experiences [383,384]. The hippocampus seems to stand out when somatic retrotransposition rate is considered. While L1 retrotranspositions are more frequent in the mouse brain compared with non-neuronal tissues, in the hippocampus L1 activity is the highest among brain regions [385]. Single-cell RC-Seq showed that genes preferentially expressed in the hippocampus are significantly enriched for intragenic somatic L1 insertions in both hippocampal neurons and hippocampal glia. Curiously, no similar enrichment was found for cortical neurons. In the hippocampus samples, transcribed enhancers active in NSCs were also enriched for L1 insertions [353]. These observations suggest some functional significance of TE insertions specifically in the hippocampus.

Importantly, in many studies examining the hippocampus in the context of learning retrotransposition may be overlooked, since de-novo DNA synthesis registered in this structure is usually attributed solely to replication necessary for neurogenesis [386]. There is currently only a few published works in which the role of retrotransposition in learning and memory was specifically addressed, and while some correlations between TE activity and memory formation were confirmed, molecular mechanisms underlying these correlations are still poorly understood. Changes in TE activity specifically in the hippocampus were also reported to take place after stress or voluntary exercise.

A link between learning and reverse transcriptase activity in the hippocampus was first proposed quite a long time ago, in 1983, even though the methods used in this work were not accurate enough by modern-day criteria [387]. In the study performed by Kokaeva et al., a mixture of two antisense probes, one binding with the reverse-transcriptase-encoding region of L1 and the other binding with a unique L1 region, was used. Intrahippocampal injection of this mixture to rats was performed 6 h before training in a Morris water maze. Short-term memory (STM) and long-term memory (LTM) formation were assessed 30–40 min and 48 h after the training, respectively. While no difference in performance between the groups was registered during the training session or 30–40 min later, animals treated with anti-L1 oligos showed impaired results in the memory retention test 48 h after the training [388]. Bachiller el al. obtained similar results in mice using a different learning paradigm. Mice were systemically injected with lamivudine (RT inhibitor) just after passive avoidance (PA) training. Lamivudine did not affect the STM formation estimated 1 h after the training; however, at 72 h after the training, lamivudine-treated mice showed impaired LTM formation. The functional role of L1 expression in LTM formation was confirmed by intrahippocampal injection of antisense probes binding with ORF1 or ORF2; both probes impaired LTM in the PA paradigm and in the novel object recognition test. In another series of experiments, these authors demonstrated a neuronal-activity-dependent activation of L1 elements in murine hippocampus after exposure to a novel environment. A transient increase in both ORF1 and ORF2 mRNA levels was observed at 1 h after the novel exploratory session. In the genomic DNA extracted from the same hippocampi, a permanent increase in the number of ORF2 insertions, but not ORF1 insertions, was registered 1 h after the exposure. Systemic administration of a lamivudine (immediately after the exposure) or glutamatergic antagonist cocktail (before the exposure) both blocked de-novo ORF2 insertions. No exact mechanism of L1 participation in memory-formation processes was found in this study, but it was speculated that L1 insertions may contribute to LTM formation by generating neurons with unique genomes and, therefore, long-lasting distinct transcriptomes by altering expression of nearby genes and/or the fate of their transcripts. However, it was shown that lamivudine administration had no effect on the exploratory-dependent expression of immediate early genes *Fos* and *Egr1*; therefore, the genes that are possibly targeted by neuronal activity-dependent L1 insertions are yet to be discovered [389]. Bedrosian et al. reported a correlation between early life experience and L1 retrotransposition in murine hippocampus. It is known that L1 activity increases during neuronal differentiation, and during the first week of life the hippocampus still undergoes extensive cell division and differentiation in mice. Thus, the sensitivity of L1 transposition to experience may be detected at this stage. The authors did not use artificial training paradigms, but compared animals who received high or low maternal care. These two “maternal styles” were identified by monitoring the behavior of animals in their home cage. It was found that pups reared with low maternal care had more L1 copies in their hippocampi. Copy numbers of other murine mobile elements did not correlate with maternal care. There was no such correlation for L1 copy number in the frontal cortex or heart. Cross-fostering experiments showed that L1 3′-UTR copy number in the hippocampi of rats at the age of 21 days correlated more with the maternal care level of their foster mother than that of their biological mother. This work shows that there is neuronal plasticity at the level of the DNA sequence in response to environmental changes at early age; however, physiological or pathological mechanisms in which such plasticity may be involved are not exactly clear [385].

Low maternal care may be considered a chronic stress exposure. In that context, the increase in hippocampal TE activity reported in [385] is in line with the general tendency of TE activation as a result of stress on cellular or organismal levels [11,33,36,176,177,178,179,390]. However, another research revealed a retrotransposon silencing response in the hippocampus following a stress exposure (specifically, acute restraint stress). Hunter at al. first observed an overall increase in the repressive histone H3 lysine 9 trimethylation (H3K9me3) level in the hippocampus of rats subjected to restraint. Notably, this effect was hippocampus-specific; there were no changes in H3K9me3 level in cerebellum, frontal cortex, heart, liver, and skeletal muscle. More detailed screening with ChIP-Seq to determine the genomic localization of the H3K9me3 response showed that, after the stress exposure, H3 methylation level in repetitive regions increased more prominently compared with its increase in genes or intergenic regions. RT-qPCR was then performed to estimate the expression of different TEs; remarkably, while hippocampal RNA levels of B2 and ERV elements were reduced as a result of stress, L1 RNA level was not; therefore, it seems like TE repression by methylation spared L1 elements. The authors point out that, previously, H3K9me3 was associated with ERV but not LINE elements [196]. In another study, the changes in L1 expression after a stress exposure were only observed in the hippocampus, but not in prefrontal cortex or amygdala [391] (this paper is also discussed in Section 3.1.3).

Muotri et al. generated L1-EGFP transgenic mice with an L1 indicator cassette that activates the expression of the EGFP reporter only after somatic retrotransposition, and provided them unlimited access to running wheels. The authors foung that EGFP expression in granular cells in the DG of the hippocampus is increased in mice exposed to voluntary exercise when compared with sedentary animals (with no access to running wheels). The authors suggested that NPCs in adult brain may support de-novo L1 mobilization upon exposure to a new environment. Interestingly, it was noted that EGFP expression in non-neurogenic brain regions also increased as a result of exercise [392]. This is in agreement with the results reported by Ivashkina et al., who observed increased DNA synthesis in DG and in non-neurogenic brain regions in mice 3 days after PA training. This study did not identify the source of the new DNA, but retrotransposition was one of the supposed mechanisms [386].

### 2.6. Is It Possible That Neurons Use TEs for RNA-Templated DNA Damage Repair?

Neurons and NPCs are especially prone to DSBs compared to other cell types. Two kinds of genes are particularly affected by this: immediate early genes, the expression of which was shown to be regulated by promoter breaks [349,393], and genes with extremely long introns and multiple small exons [346,347]. Neurons have to live for many years, and the integrity of their genome needs to be constantly maintained to make this possible. Accumulation of unrepairable or incorrectly repaired DNA lesions was linked with neuropsychiatric diseases [394] and neurodegeneration [395]. Repair mechanisms such as NHEJ, DNA mismatch repair (MMR), base excision repair (BER), and nucleotide excision repair (NER) (including transcription-coupled repair (TCR, or TC-NER)), were confirmed to be necessary for normal neuronal function [396,397,398]. However, HR, the most accurate mechanism of DSB repair, is considered to be impossible in neurons, since HR may happen only during the S or G2 phases [332]. This still leaves the opportunity of RNA-templated recombination that was discovered not long ago. This mechanism in yeast [329,330] and the U2OS cell line [325] was described above, in Section 1.8.10. The same research group that studied this mechanism in U2OS cells then continued investigating DSB repair in neurons. Welty et al. observed that in neurons, just like in U2OS cells, RAD52 is recruited to the sites of DNA damage and that this process is dependent on the presence of nascent mRNA generated during active transcription; both RNase H treatment or transcription inhibition prevented RAD52 recruitment. Electrophoretic mobility shift assay demonstrated that RAD52 binding affinity to ssRNA is even higher than its affinity to ssDNA. Furthermore, RAD52 had a high binding affinity to R-loops, the damage-prone structures at transcriptionally active sites in the genome. The authors also showed that high concentrations of amyloid β (Aβ) inhibit both the expression of RAD52 and its association with DNA damage sites. Importantly, Aβ had a similar effect on RAD52-dependent repair in neurons and U2OS cells, suggesting that the same repair pathway functions in dividing and non-dividing cells [333]. All these results, taken together, provide conclusive evidence that RAD52 participates in RNA-templated DNA repair in neurons. Nevertheless, in this study, the reverse transcriptase responsible for RNA-dependent DNA synthesis was not identified. We may speculate that, since neurons were shown to have increased L1 activity [353,354], L1 ORF2p is a likely candidate. Importantly, experiments with non-neuronal cells demonstrated that L1 retrotransposition happens mainly during the S phase, while cells artificially blocked in the G1 phase show very low levels of retrotransposition [51]. However, mature neurons are naturally in the G0/G1 phase and still show increased L1 activity, and this may be considered an indirect argument for the unusual functions of L1 elements in neurons. Further experiments to investigate these functions are necessary.

### 2.7. Specific Examples of Domesticated Mobile Elements in Neurons

#### 2.7.1. Neuron-Specific Transcription Regulation Provided by Exapted TEs

TEs from different families were exapted to provide regulatory sequences important for mammalian brain development and function. Emera et al. calculated how many neocortex-specific enhancers overlap with repetitive elements in the human genome. They found that neocortical enhancers in general do not exhibit strong evidence of TE exaptation. However, when recently evolved and more ancient enhancers were assessed separately, it was found that less than 6% of bases in mammalian-specific and older neocortical enhancer cores intersect a repeat element, while for eutherian-specific neocortical enhancer cores this figure was about 30% [399].

Sasaki et al. identified a new SINE family in the genomes of *Amniota* and named it AmnSINE1. A total of 32 of AmnSINE1 loci are located near genes involved in brain development. The authors focused on locus AS071 located 178 kb downstream from *Fgf8* (fibroblast growth factor 8) gene. They used *lacZ* transgenic mouse system to test whether AS071 may regulate the expression of adjacent genes. It was found that AS071 is expressed in the lateral diencephalon and hypothalamus of mouse embryos, and that the expression pattern of *Fgf8* in these regions is identical to that of AS071, suggesting that AS071 functions as a *Fgf8* enhancer [400]. Further experiments with constructs encoding AS071 with various deletions showed that AS071 contains at least three distinct sub-elements directing enhancer activity in different diencephalic domains. AS071 includes three putative TF binding sites, but deletions of these sites did not significantly affect *lacZ* expression pattern in enhancer assays [401]. The same research group studied another conserved AmnSINE1 locus, AS021. Tashiro et al. used the *lacZ* reporter transgene to confirm that AS021 has enhancer activity in developing mouse brain. The expression pattern of AS021 was compared with those of genes located within 2 Mb of AS021. AS021 expression recapitulated the expression of the *Satb2* gene (encoding the TF Special AT-rich sequence-binding protein 2) at later embryonic and postnatal stages in the deep-layer cortical neurons that project axons into *corpus callosum*. The conserved region of AS021 contains 27 predicted TF binding sites; modified AS021 sequence with mutations in 15 of these sites had no enhancer activity. However, the exact mechanism of AS021 enhancer function was not established in this work [402].

MER130 (a very ancient transposon family that cannot be easily classified [19]) also displayed enhancer properties in the brain. Notwell et al. used ChIP-seq for p300 (a transcriptional co-activator recruited by active enhancers) to identify putative enhancers. MER130 elements were highly enriched among p300-binding sequences in the mouse dorsal cerebral wall (which gives rise to the neocortex) at embryonic day 14.5, but not in other tissues, including the forebrain at an earlier time point. Six of these MER130s are located next to genes important for telencephalon formation. Almost all MER130 instances contain a conserved core with five putative TF binding sites: a NeuroD/Neurog motif, an NFI dimer and two additional NFI sites. These TFs are implicated in brain development. In luciferase reporter assays in primary neuronal cultures, 22 of 23 MER130s strongly enriched for p300 signal induced more than two-fold expression upregulation. Mutations in most of the TF binding sites caused a significant reduction in reporter activity [403].

Another important example is the hypothalamus-specific enhancer that regulates neuronal expression of the proopiomelanocortin (*POMC*) gene in mammals. Santangelo et al. compared multiple mammalian genomes and demonstrated that the neuronal *POMC* enhancer nPE2 originated from the exaptation of a CORE-SINE (MIR) element in the lineage leading to mammals and remained under purifying selection during the mammalian evolution. Analysis of *POMC* expression in the brains of transgenic mice featuring deletions in different parts of ectopically expressed nPE2 showed that two regions of nPE2, both derived from CORE-SINE, are essential for enhancer activity in POMC hypothalamic neurons [404]. The same research group, using the same in-silico paleogenomics strategy, then showed that another *POMC* enhancer, nPE1, also has a TE-related origin. nPE1 evolved from an LTR of the MaLR family, and it happened after the emergence of nPE2; nPE1 is a placental mammalian novelty. Transgenic mice expressing different fluorescent markers driven by nPE1 or nPE2 demonstrated coexpression of both reporter genes along the entire arcuate nucleus. The onset of reporter gene expression guided by nPE1 and nPE2 was also identical and coincidental with the onset of *Pomc* expression in the presumptive mouse diencephalon [405].

Bejerano et al. identified an ancient TE called LF-SINE and confirmed experimentally that at least one copy of this element, conserved between mammals, chicken and frog, is a functional enhancer. Expression pattern of a representative reporter gene construct driven by the human *ISL1*-proximal LF-SINE in whole-mount mouse embryos closely resembled the expression pattern of the murine *Isl1* gene. *ISL1* encodes a TF necessary for motor neuron differentiation [264].

Crepaldi et al. showed that SINEs binding pol III-specific TF TFIIIC play an important role in pol II-dependent transcription of genes induced by neuronal activation. The authors performed ChIP-seq for H3K9K14ac (used to identify active neuronal genes) in the somatosensory cortex of mice exposed to novel enriched environmental conditions (NEE). Increased H3K9K14ac level in response to NEE was detected in SINEs located within 100 kb of transcription start sites (TSS) of NEE-induced genes. These SINEs contain TFIIIC binding sites. ChIP-PCR confirmed that TFIIIC binds to specific SINEs located near immediate early genes: an RSINE1 downstream of *Fos* and a B1F element downstream of *Gadd45b*. An immuno-DNA FISH experiment used to visualize transcription factories in the nuclei of neurons showed that co-localization of both *Fos* and *Gadd45b* loci with transcription factories increased in response to depolarization. KD of a TFIIIC subunit increased the co-localization of *Fos* and *Gadd45b* with transcription factories in resting conditions, while depolarization has no further effect in these neurons. Thus, TFIIIC apparently functions as a transcriptional “brake” in non-stimulated neurons. Indeed, on the morphological level, neurons with TFIIIC KD resembled neurons subject to chronic depolarization [243]. Later, the same research group showed that pol III also participates in pol II transcription regulation by SINEs. Policarpi et al. identified more than 1000 enhancer SINEs (eSINEs) that are enriched within 100 kb of TSS of depolarization-inducible genes and have H3K27ac and H3K4me1 marks (associated with putative enhancers). eSINEs bind TFIIIC upon depolarization and are transcribed in activated neurons. One specific eSINE, Fos^RSINE1^, had been previously identified as a regulatory element [243]. RT-qPCR performed at different time points after depolarization showed that transcription of the Fos^RSINE1^ enhancer RNA (eRNA) starts earlier than transcription of the corresponding Fos mRNA. Experiments with mouse neuroblastoma cells Neuro-2a showed that Fos^RSINE1^ eRNA is needed for *Fos* relocation to transcription factories, and that a pol III binding site is crucial for the enhancer activity of Fos^RSINE1^. Importantly, RNA immunoprecipitation confirmed that Fos^RSINE1^ eRNA directly binds with pol II. In Neuro-2a cells with Fos^RSINE1^ eRNA KD, recruitment of initiating and elongating forms of pol II at the *Fos* gene was hindered. Study of primary cortical neurons in vitro and in utero electroporation experiments in vivo showed that that Fos^RSINE1^ eRNA regulates migration and neuronal differentiation of NPCs and is required for activity-dependent dendritogenesis in maturing neurons [406].

#### 2.7.2. Neuron-Specific Proteins Encoded by TE-Derived Genes

The most important example of a TE-derived gene participating in normal neuronal function is *ARC*, which encodes Arc (activity-regulated cytoskeleton-associated) protein. The *ARC* gene in tetrapods and its analogs in flies (*darc1*, *darc2*) independently originated from distinct lineages of Ty3/*gypsy* retrotransposons [407]. *ARC* has homology with retroviral *gag* genes. Some aspects of Arc mRNA regulation also resemble those of viral RNAs, as Arc mRNA contains an internal ribosomal entry site (IRES) that allows cap-independent translation [408]. Arc protein is crucial for multiple stages of long-term plasticity in neurons. Arc plays a critical role in AMPA receptor trafficking and has been implicated in actin polymerization at synapses. Most likely, Arc is involved in other plasticity-related processes that are not yet identified [408,409]. Arc-deficient mice exhibit profound deficits in memory consolidation, despite intact STM and learning acquisition. Synapses in the visual cortex of Arc-deficient animals were shown to be insensitive to the effects of both experience and deprivation [408]. Arc function in neurons is surprisingly similar to the behavior of the viral Gag protein. Experiments with Arc solutions prepared by protein expression in bacteria showed that Arc self-assembles into oligomers resembling viral capsids, detectable by electron microscopy, and that these capsids contain RNA and require RNA for successful assembly. Arc protein and Arc mRNA were then detected in extracellular vehicles (EVs) from cultured neurons. Stimulation of neuronal activity increased the release of Arc-containing EVs, and Arc-KO neurons were able to uptake Arc protein and Arc mRNA from EVs purified from WT neuron culture medium. Since Arc capsids did not show specificity in RNA binding in vitro, the authors suggested that Arc transfers abundant mRNAs between neurons, providing a novel mechanism in intercellular signaling [407]. A similar mechanism was shown for dArc1 protein in Drosophila. dArc1 was confirmed to bind its own mRNA and enter EVs transferred from motor neurons to muscle cells. Artifical disruption of this transfer affected both synapse maturation and synaptic plasticity [410].

*POGZ* (*pogo* transposable element with zinc finger domain) encodes a protein containing a CENP-B-like DNA-binding domain. Mutations in this gene were associated with intellectual disability (ID) with comorbid autism spectrum disorders (ASD). In the human brain, *POGZ* is co-expressed with ASD- and ID-associated genes involved in chromatin remodeling and transcription regulation. *POGZ* was suggested to serve as a neuron-specific transcriptional regulator. *POGZ* was also shown to be involved in mitosis [411,412]. White et al. identified five patients with gene-disrupting mutations in *POGZ* and variable neurodevelopmental disorders. All patients had ID, global developmental delay, behavioral abnormalities and vision abnormalities. Furthermore, 3 out of 5 subjects had microcephaly [412]. Stessman et al., after screening sequencing data of thousands of patients, collected a cohort of 24 individuals diagnosed with various neurodevelopmental disorders and possessing likely gene-disrupting de-novo mutations in *POGZ*. Patients with these mutations all had some degree of developmental delay; most of them had mild ID. In a *Drosophila* model, KD of *row* (an ortholog of *POGZ*) in neurons led to deficits in habituation, a form of learning that is highly relevant for both ID and ASD [411].

The *JRK* gene is also related to *pogo* DNA transposons and shows similarity with *CENPB* [293,413]. It was originally named *Jerky* because mice lacking this gene are prone to epileptic seizures. JRK was shown to bind RNA with high affinity. In mice, JRK mRNA is expressed in multiple tissues, but on the protein level JRK showed brain-specific expression. In primary rat hippocampal cultures, JRK was labeled mostly in neurons. It was shown that JRK may participate in mRNA conservation in neurons: in brain extracts, JRK was identified in translationally inactive mRNP particles but did not associate with ribosomes. JRK is also supposed to have a CENP-B-like function in the nucleus based on similarity between these proteins [413].

The *RTL4* gene (also called *MART4, SIRH11* or *ZCCHC16*) encodes a putative protein that has a homology with Gag retroviral protein and includes a conserved RNA-binding domain. It must be noted that human *RTL4* is located in the region of X-chromosome where several X-linked ID genes have been identified. *Rtl4* KO mice exhibit a number of behavioral abnormalities—hyperactivity, poor working memory, enhanced impulsivity and/or lack of attention. Mutant mice also have abnormal regulation of the monoamine metabolism in their prefrontal cortex. A negative correlation between expression levels of *Rtl4* (more precisely, its 3′-UTR, left intact in the *Rtl4-*KO mice) and *Dbh*, the gene encoding dopamine β-monooxigenase, was observed in the brain, suggesting that the activity of these two genes may be regulated by the same environmental factors. Surprisingly, while *Rtl4* expression in various brain regions was confirmed by RT-PCR, numerous experiments provided no direct evidence of Rtl4 mRNA or protein localization in the brain. This leaves the possibility that Rtl4 in neurons functions as a non-coding RNA. However, the conservation of the amino acid sequence of the Rtl4 provides indirect evidence that Rtl4 must be a protein-coding gene [303].

The *NRIF* (*ZFP369*) gene includes the conserved SCAN domain, a protein–protein interaction motif derived from the sequence encoding the *C*-terminal portion of the Gag retroviral protein [199,414]. NRIF (neurotrophin receptor-interacting factor) encoded by this gene is a zinc finger protein that binds the neurotrophin receptor p75NTR and serves as a mediator of neuronal apoptosis. The retinae of mice with a targeted mutation in the *NRIF* gene show reduced cell death, similar to nerve growth factor (NGF)-null mice [414]. Another important function of the NRIF protein is regulation of neuronal cholesterol biosynthesis in the hippocampus. Mice with *NRIF* deletion showed a reduction in cholesterol biosynthesis-related gene expression in their hippocampi [415].

The *PGBD5* gene is related to *piggyBac* DNA transposons. It encodes a neuron-specific domesticated transposase with a yet unknown function. This gene is highly conserved in vertebrates, and it was shown that, in mammals, *PGBD5* is almost exclusively expressed in the brain, with preferential expression in certain granule cell lineages. The putative *PGBD5* promoter has a binding site for NRSF/REST, a factor that works as a scaffold for proteins that repress neural genes in non-neural cells. The PGBD5 protein in mouse brain is detected in both nuclear and cytoplasmic fractions, but it was shown that nuclear PGBD5 is not bound to chromatin, since PGBD5 was not released from the crude nuclear pellet either by complete DNase I digestion or by elevated NaCl concentrations. These results suggested that PGBD5 is probably a structural component of the nucleus [416]. More recently, it was confirmed that ectopically expressed *PGBD5* is capable to induce stereotypical cut-and-paste DNA transposition in HeLa cells [285].

#### 2.7.3. Neuronal Non-Coding RNAs Encoded by TE-Derived Genes

It was estimated that about two thirds of all known lncRNAs in zebrafish, mouse, and human contain at least one TE-derived sequence [417]. In the human genome, about 40% of all identified lncRNAs are specifically expressed in the brain [418]. lncRNAs are abundant in the brain and show spatially limited and time-varying expression. They are suggested to play a key role in the hippocampal neurogenesis [419]. Most lncRNAs expressed in developing mouse brain overlap with genes involved in neurogenesis and are coexpressed with these genes [420].

*BCYRN1* is a gene derived from a monomeric Alu element. *BCYRN1* encodes brain cytoplasmic RNA BC200, a primate-specific non-coding RNA [7,421,422]. Interestingly, a functional BC200 analog in rodents, BC1 RNA, is also related to a family of SINEs (ID elements); apparently *Bc1* gene is not a domesticated ID repeat, but rather a “master gene” that gave rise to a whole subfamily of IDs [423]. Both BC1 RNA and BC200 RNA are expressed in neurons, providing rare examples of tissue-specific pol III transcription [95,424], and have the same function, playing an important role in synaptic plasticity. BC1/BC200 participates in local translation regulation in neuronal dendrites. In the absence of synaptic activation, BC1/BC200 binds translation initiation factors, preventing the “premature” translation of dendritic mRNAs. Once the synapse is stimulated, BC RNA releases these factors and the local translation may start [425,426]. *BCYRN1* kept its transpositional ability: more than 200 retrocopies of this gene were identified [422].

Carrieri et al. identified a lncRNA antisense to mouse *Uchl1* gene (encoding ubiquitin carboxy-terminal hydrolase L1). UCHL1 is a neuron-restricted protein; loss of UCHL1 function has been associated with different neurodegenerative disorders. The authors showed that antisense *Uchl1* transcript (AS Uchl1) increases Uchl1 mRNA translation in the MN9D dopaminergic cell line. AS Uchl1 includes a B2 SINE, which is necessary for the lncRNA activity. In control cells, rapamycin treatment caused increased UCHL1 translation; this did not occur in cells with AS Uchl1 KD or cells overexpressing the mutant form of AS Uchl1 lacking the B2 element [427]. Later, more lncRNAs like this were discovered; this new functional class of lncRNAs was named SINEUPs (SINE element-containing translation up-regulators). Schein et al. screened human brain transcriptome and identified 129 putative SINEUPs. The authors selected two of these transcripts (not neuron-specific) and confirmed their SINEUP properties experimentally [428].

lncRNAs may also perform regulatory functions as “miRNA sponges”, competing with mRNAs for binding of corresponding miRNAs. There is an example of TEs providing multiple miRNA binding sites within one RNA sequence expressed in neurons. Bhattacharya et al. characterized CYP20A1_Alu-LT RNA (a non-coding transcript of the orphan human gene *CYP20A1*) with an unusually long (8.93 kb) 3′-UTR, almost two thirds of which is derived from exonizations of 23 Alus of different subfamilies. CYP20A1_Alu-LT is widely expressed; it is present in published data on the single nucleus RNA-seq of human cortical neurons. CYP20A1_Alu-LT 3′-UTR harbors thousands of predicted miRNA recognition sites. There are 140 specific miRNAs having 10 or more recognition sites each within this sequence. The miRNA sponge function of CYP20A1_Alu-LT was indirectly confirmed in experiments with primary human neurons. Heat shock downregulated CYP20A1_Alu-LT transcription in these cultures, while the application of the HIV1-Tat neurotoxic protein upregulated it. RNA-seq identified 380 genes, expression of which positively correlates with that of CYP20A1_Alu-LT after heat shock or HIV1-Tat treatment. All these genes contain at least one recognition site for neuron-expressed miRNAs predicted to bind to CYP20A1_Alu-LT. These genes and corresponding miRNAs are known to be involved in hemostasis and neuron development [283].

Two other TE-related non-coding RNAs that apparently play an important role in neurons are SLC7A2-IT1A and SLC7A2-IT1B isoforms. These two RNAs have different transcription starts within an intron of *SLC7A2* gene, and both contain the sequence from an L1 insertion nested in an Alu insertion. SLC7A2-IT1A/B expression was confirmed in several human brain structures but the function of these transcripts is not known. They have no homology with any known protein or regulatory motif. It was demonstrated that a mutation increasing their piRNA-mediated silencing leads to lethal encephalopathy [429].

TE-derived small RNA species were also found to be expressed in neurons. piRNAs, many of which have a TE-related origin, used to be considered germline-specific. However, recent studies suggest that this class of small RNAs may also have some functions in neurons in both vertebrates and invertebrates [146,147,148,156,430]. There are also reports about endogenous siRNAs being expressed in neurons [156,430]. Small RNAs of these two classes may arise not only from TEs but from other genomic regions as well. In some papers, the origin of neuron-specific piRNAs is unclear [146] (Rajasethupathy, personal communication), while other works explicitly state that such piRNAs are mapped to unique sites in the mouse genome [147]. However, a large-scale sequencing of small noncoding RNAs in mouse primary hippocampal neurons performed by Nandi et al. discovered numerous TE-related piRNAs and siRNAs. RNA-seq analysis of mouse brain regions revealed the expression of transcripts encoding proteins involved in piRNA and endo-siRNA biogenesis. Mice with KO of the *Mili* (*Piwil2*) gene (encoding a murine analog of PIWI) displayed hyperactivity and reduced anxiety. Analysis of brain samples from these mice allowed the authors to identify specific L1 promoter regions that are putative targets for methylation by a MILI/piRNA-related mechanism in neurons [156].

### 2.8. TEs in the Human Brain Evolution

TE activity is one of the most notable sources of mutagenesis, and it was linked to major genome rearrangements [431] and the formation of new species [432]. Notably, the human genome is particularly enriched for large, highly homologous segmental duplications. Bailey et al. examined sequence features at the junctions of duplications and observed a highly significant enrichment of Alu sequences near or within junctions [433]. Analysis of UTRs of human and mouse mRNAs showed that, in both species, recently expanded gene classes have transcripts enriched in TEs [207]. The key part of brain evolution is changes in the brain development, and there is significant evidence that TE-related mechanisms participated in shaping mammalian embryonic development, including neuronal differentiation [9]. Here, we will focus on primate-specific and human-specific TEs, and, in particular, on neuron-specific genes that could be affected by TE insertions during human evolution.

#### 2.8.1. An overview of TE Evolution in Primate Lineage

During primate evolution, there have been multiple waves of LINE retrotransposition as well as the birth of completely new mobile elements such as Alu and SVA [7]. Comparative genomic analysis between human and chimpanzee genomes allowed the identification of about 11,000 TEs that are differentially present. It was determined that, during the past several million years (My), the overall level of transposition was higher in human than in chimpanzee. L1, Alu, and SVA elements together comprised more than 95% of recent insertions in both species. For all three types of TEs, the number of recent TE insertions in the human genome was at least 1.5 times higher than in the chimpanzee genome (more than three times higher for Alus). Alu insertions constituted 71% of all human-specific TE insertions and 56.1% of chimpanzee-specific insertions [434]. More recent analysis of 83 fully sequenced human, chimpanzee, bonobo, orangutan and gorilla genomes focused on L1 and Alu elements showed that Alu retrotransposition rate underwent the most dramatic changes over the shortest time intervals, with rates of accumulation differing by 15 fold and varying significantly among all branches. Differences in the L1 insertion rate were much more modest and gradual. The human lineage showed the most notable decline in L1 accumulation and increase in Alu accumulation, even though humans, chimpanzees, and bonobos all experienced increased Alu retrotransposition [435]. Although L1 elements were not extensively mobilized, their retrotransposition machinery was used by other elements to propagate. About 40–50 My ago, coincident with the onset of the radiation of the higher primates, ancestral primate genomes underwent intensive amplification of Alus, pseudogenes generated by retrocopying, and specific L1 subfamilies (L1PA6, L1PA7 and L1PA8). This suggests that these L1 subfamilies must have mobilized RNAs in *trans* at accelerated rates [436]. Hominid-specific SVA elements first originated ~25 My ago; this coincided with the increase in the frontal cortex size. The largest increase in *Homo* brain size occurred within the past ~5 My. During this period, L1Hs, AluYa5, AluYb8, and SVA flourished, and these elements are still the most active TEs in the human genome [7].

#### 2.8.2. TE-Mediated Recombinations in Human Brain Evolution

Sen et al. compared human and chimpanzee genomes and identified 492 human-specific deletions attributable to recombinations between Alus. The majority of the deletions happened within known or predicted genes (including three deletions that removed functional exons). Of particular interest is the deletion of the fourth exon in the predicted chimpanzee gene *LOC471177*, which is orthologous to human *CHRNA9* gene encoding neuronal acetylcholine receptor subunit α9, associated with cochlea hair cell development. Mutations in this gene may affect the dynamic range of hearing [437].

*RHOXF2* is a homeobox gene expressed primarily in the testis, but a growing body of evidence suggests that this gene also has some important function in the brain. There are two copies of this gene in the human genome; in striking contrast to non-human primates, humans appear to have homogenized their two *RHOXF2* copies by the ERV-mediated non-allelic recombination mechanism [438].

#### 2.8.3. Multiple New Regulatory Elements Evolved from TEs in the Human Lineage

In mammalian brains, TE-derived sequences have been implicated in multiple layers of gene regulation, from transcriptional and post-transcriptional control to three-dimensional chromatin organization. There are a few examples of convergent evolution of such regulatory elements in primates and rodents (reviewed in [9]).

It is now considered that the speciation and evolution of human lineage happened due to many incremental independent genomic regulatory changes over extended evolutionary periods, not due to a singular phenotype-defining event. Guffanti et al. analyzed human dorsolateral prefrontal cortex transcriptomes and noted that a large set of TEs expressed in this region are highly conserved during 8 My of primate evolution. Although TEs cannot be considered the only mechanisms driving the evolution of the human brain, these conserved elements are likely conveying important primate-specific regulatory functions [439]. As was already mentioned, analysis of chromatin accessibility maps in multiple non-neuronal human cell types showed that almost half of open chromatin sites (DHS) overlapped with TEs; notably, this overlap was higher for primate-specific sequences [202]. In a study by del Rosario et al., anthropoid-specific constrained (ASC) regions were identified in human, orangutan, rhesus macaque, and common marmoset genomes. It was shown that 99.7% of the ASC base pairs are noncoding, suggesting their *cis*-regulatory functions. ASCs were highly enriched in loci associated with fetal brain development, motor coordination, neurotransmission, and vision. Notably, 56% of ASCs overlapped with TEs [440].

Luo et al. compared 3D genomes [441] of human, macaque, and mouse brains and identified hundreds of human-specific chromatin structure changes. The authors showed that human-specific chromatin loops are enriched in enhancer–enhancer interactions, and the regulated genes show human-specific expression in the subplate, a transient zone of the developing brain critical for neural circuit formation and plasticity. Importantly, human-specific topologically associating domains (TAD) boundaries are enriched with TEs. For Alu and L1, it was shown that only evolutionarily younger subfamilies of these elements (AluY and L1PA) are enriched in human-specific TAD boundaries. The conserved TAD boundaries are more enriched in housekeeping genes, while the human-specific boundaries are more enriched in brain-development-related genes. One example is the human gained TAD boundary around *CNTN5*, a gene involved in neuron circuit formation and autism spectrum disorders; this boundary contains AluY elements and is correlated with an increased *CNTN5* expression in human compared with macaque [442].

Nataf et al. screened the EnHERV database to determine whether candidate lists of cognition-/behavior-related genes are enriched in genes harboring promoter-localized HERV LTRs. The authors found that promoter regions of human genes associated with ID (but not with autism or schizophrenia) are uniquely enriched in LTR sequences of primate-specific MER41 elements. MER41 LTRs in these promoters harbor TF binding sites for YY1 and immunity-associated TFs such as STAT1, STAT3 and NFκB1. Human and chimpanzee genomes were found to have different MER41 LTR insertion sites. A survey of the human proteome allowed the authors to map a protein–protein network which links the suggested immune/MER41/cognition pathway to FOXP2, a key TF involved in the emergence of human speech [443].

SVA elements, despite being the youngest and smallest active human TE family, are supposed to play an important role in human evolution, providing new *cis*-regulatory elements. Some genes important for neuronal function have SVA insertions in their upstream flanking sequence. For two specific SVA D elements upstream of *PARK7* (involved in pathogenesis of Parkinson’s disease) and *FUS* (linked to multiple neuropathologies including amyotrophic lateral sclerosis), the ability of SVA sequence to regulate gene expression rate was confirmed experimentally [20,109]. Of particular interest is possible SVA-related regulation of neuropeptide signaling. VNTRs that are a part of the SVA sequence have been demonstrated to be key neuropeptide regulatory domains. The arginine vasopressin 1a receptor gene (*AVPR1A*) contains two upstream VNTRs, and their regulatory role was confirmed experimentally. SVA insertions were also found in the upstream regions of *OXT* (the gene encoding oxytocin), *TACR3* (tachykinin receptor 3 gene), and *TRPV3* (the gene encoding a vanilloid receptor involved in the modulation of neuropeptide release) [108].

#### 2.8.4. Alu Elements Participated in Our Brain Evolution in Diverse Ways

A recent computational analysis suggested that Alu elements have the largest overall regulatory impact compared to other types of TEs in the human genome: Alu elements showed the highest probability of contributing to regulatory regions of any type (defined by chromatin state). Interestingly, this study showed that most TE types are primarily associated with reduced gene expression, while Alus are associated with upregulated gene expression [444]. Considering Alu expansion in the human lineage [434,435], the effect of these elements on our gene regulation is very important.

An important example of regulatory sequences that have been widely distributed across the genome by Alu expansion is retinoic acid response elements [7]. Retinoic acid is a crucial regulator of both embryonal and adult neurogenesis [445,446]. In humans, 90% of DR2 DNA motifs recognized by retinoic acid receptors are present within Alu repeats. Alu-DR2 motifs are located adjacent to numerous known retinoic acid target genes, and some of these elements were confirmed to recruit retinoic acid receptors in cell culture experiments. Alus also contain other retinoic acid response elements, including 50% of consensus DR4 motifs [447].

The Alu sequence is the primary target of A-I editing in human RNAs. The intensity of A-I editing in human transcriptome is much higher compared with other animals [274,369,448]. In mice, for example, A-I editing sites are distributed between TEs of different families, and two elements with a low level of homology are less likely to form a stable dsRNA structure recognized by ADAR enzymes (while two human Alus form such a structure easily) [274]. However, the A-I editing level in human brain was found to be higher even when compared with other primates that have similar Alu repeats in their genomes [449]. The number of Alu RNA editing sites in the human genome is more than a million [450], and by some estimations may even be more than 100 million [271]. A-I editing is necessary for normal neuronal function, since it affects genes encoding key proteins involved in neurotransmission, neurogenesis, gliogenesis, and synaptogenesis [451]. The examples of neuron-specific transcripts that are the most important targets for A-I editing have been already provided above in the Section 2.3. In addition, Mattick and Mehler analyzed the human RNA editing datasets and found many transcripts involved in nervous system development, formation of the neural tube, and NSCs regulation [371]. Importantly, Paz-Yaacov et al. found that new editable species-specific Alu insertions that happened after the human–chimpanzee split are significantly enriched in genes related to neuronal functions and neurological diseases [449]. It must be noted that recent studies using full sequencing of genomes and transcriptomes of diverse species revealed that recoding mRNA A-I editing sites affecting the structure of proteins are actually rare; in most cases, editing happens in noncoding mRNA regions, and in human transcriptome the percentage of recoding A-I editing sites is negligible compared to some invertebrate species [372,452]. However, non-coding regulatory RNAs may also be A-I editing targets [371,453]. A-I editing may also influence splicing and miRNA/mRNA interaction [453]. The transcript of the *NARF* gene provides an interesting example of alternative splicing made possible by a few A-I editing events within Alu elements. The resulting new exon is alternatively spliced in a tissue-dependent manner, with a high level of inclusion in the brain [454].

Alu exonizations in neuron-specific genes seem to have distinct properties compared to Alu exonizations in other genes. While analysis of the whole human transcriptome showed that most Alu exons are present in 3′-UTRs [260], in the cerebellum transcriptome most Alu exons are located in 5′-UTR regions. It was also found that, among the cerebellum-expressed genes, Alu exonization events are strongly enriched in genes encoding zinc finger TFs (ZNF); notably, ZNF genes underwent rapid expansion during human evolution [455]. An important example of a novel Alu exon is noncoding exon 2 of the *GPR56* gene encoding a G-protein-coupled receptor required for cortical development. This gene in humans has at least 17 alternative first exons, while only 5 such exons were identified in the mouse *Gpr56* gene. Interestingly, this gene also has an upstream *cis*-regulatory element showing a homology to a 4/RTE mammalian LINE [456].

Recombinations between Alu copies caused deletions [437] and duplications [433] of genome sequences; the papers cited here have been already discussed above.

There are two examples of specific Alu insertions that generated new genes active in human neurons. A monomeric Alu insertion that appeared after the split of prosimians and anthropoids generated the *BCYRN1* gene encoding the functional non-coding RNA—brain cytoplasmic RNA BC200 [7,421,422]. However, this RNA is not exclusively primate-specific; it has an analog in rodents [424,426]. Its structure and function are described in Section 2.7.3. Parrott et al. identified a small non-coding RNA family named snaR that also evolved from an Alu monomer. snaR RNAs are transcribed by pol III and associate with the nuclear factor 90 (NF90/ILF3) protein; they are suggested to participate in cell growth and translation regulation. *SNAR* genes were only found in African great apes, and some are unique to humans; however, other primates have snaR-related elements. Human *SNAR* genes are expressed in testis and brain, with differential expression of snaR subsets in different regions of the brain, but their function in neurons is not clear yet [457].

A deleterious Alu insertion was traced to happen shortly before the time when brain expansion began in human ancestors. This insertion inactivated the *CMAH* gene encoding the key enzyme for the synthesis of N-glycolylneuraminic acid (Neu5Gc). In mammals with a functional *CMAH* gene, its expression seems to be selectively downregulated in the brain. These facts led the authors to propose an intriguing hypothesis about Neu5Gc somehow restricting brain growth [458]. However, *Cmah* inactivation in mice had no effect on general appearance, sensorimotor reflexes, or learning and memory [459]. On the other hand, brain-specific *Cmah* overexpression in mice resulted in abnormal locomotor activity, impaired object recognition memory, and abnormal axon myelination, confirming that Neu5Gc has negative effects on neural functions, even though mutant mice showed no gross defects in brain size [460].

#### 2.8.5. Human-Specific TE Insertions within Neuron-Associated Genes

In the study by Mills et al. where human and chimpanzee TEs were compared, a third of all new insertions were located within known genes in both species. Further analysis of insertion patterns revealed that they were far from random. Only ~14% of all human genes were affected by TEs, and about a third of these genes contained multiple insertions. Similar results were observed in the chimpanzee genome. It appears that a large fraction of the new transposon insertions in both humans and chimpanzees were somehow targeted preferentially to specific genes [434]. We further analyzed data sets published in this paper using Gene Ontology [461,462,463]. In the human dataset, TEs were significantly overrepresented in 1866 genes; in the chimpanzee dataset, the analysis yielded significant over-representation of TEs in 820 genes. Most interestingly, many of these genes are associated with neuronal function (see GO terms in Figure 2).

## 3. The Role of Neuronal TEs in Pathology

We have found reports about neuronal TE activation in multiple diverse neuropathologies. This information is summarized in Table 3.

### 3.1. TE-Related Mechanisms of Neuropathology

#### 3.1.1. Known Mechanisms of TE-Associated Disorders in General

TEs have long been associated with different pathologies. It was demonstrated that full-size L1 insertions in human genome are subject to negative selection [71]. In 1988, Kazazian et al. were the first to demonstrate that a germline L1 retrotransposition event is a cause of a human disease (hemophilia A) [464]. Since then, more than a hundred de-novo germline TE insertions disrupting normal gene function have been implicated in multiple inherited diseases, most notably various cancer types, hemophilia and neurofibromatosis type I [465]. Somatic mutations caused by transposition or TE-mediated chromosomal rearrangements have been causally linked to several types of cancer. Derepression of a TE may also activate transcription of an adjacent proto-oncogene via the TE regulatory elements [24]. The vast majority of TEs that are transcriptionally activated in pathology are no longer replication competent. However, there are other ways in which their activity may be harmful for the host cell. Some of these transcripts encode toxic proteins or RNAs. Exotic nucleic acid structures derived from activated TEs trigger immune responses [24,33,297,310].

#### 3.1.2. Neuropathologies Associated with TE Activation

In this review, we will focus on diseases that are thought to be caused by either de-novo germline TE insertions in neuron-specific genes, or somatic transposition events in neurons. Most of TE-associated neuropathologies involve neurodegeneration and/or neuroinflammation. To date, multiple authors have provided evidence that TE activity may be involved in pathogenesis of Alzheimer’s disease, autism, schizophrenia, Rett syndrome, ataxia-telangiectasia, amyotrophic lateral sclerosis, macular degeneration and some other neuropathologies [39,56,112,113,350,466]. Physiological conditions such as stress and aging are not formally considered disorders, but since they are associated with some detrimental effects as well as with TE derepression [384,467,468,469], we discuss these conditions as well.

Multiple studies described in the previous chapter confirmed that neuronal tissue is normally more permissive to TE activity compared to other somatic tissues. Even higher TE expression and/or transposition levels were found in a number of neurological disorders. In a recent review, Terry and Devine discussed that mutations in genes that normally play a role in regulating TE activity were only confirmed in some neuropathologies. Other neuropathologies with known overall TE activation are disorders caused by a combination of genetic predisposition and environmental stress factors or disorders associated with aging. It is likely that for some diseases TE activation may be merely a side effect of cell stress [350]. However, examples of specific mechanisms directly linking TEs with detrimental processes in neurons are also known [470,471]. There are also case reports about very specific TE insertions affecting certain genes relevant for neuropathology [363,472,473,474].

Abrusán used flux balance analysis, a mathematical method used for simulation of genome-scale metabolic networks, to estimate the influence of somatic retrotransposition on human brain metabolism. The simulation showed that TE mobilization may influence the biosynthesis of more than 250 molecules, including dopamine, serotonin and glutamate. When metabolic changes associated with various neuropathologies were compared with metabolic changes caused by TE insertions, it was found that the number of metabolites influenced by TEs is significantly higher than expected by chance in the case of Parkinson’s disease, schizophrenia, and autism [475]. Larsen et al. listed 37 neurological and neurodegenerative disorders, including Alzheimer’s disease, Parkinson’s disease, epilepsy and amyotropic lateral sclerosis, in which deleterious Alu activity has been implicated as a contributing factor [451]. Jacob-Hirsch et al. used whole-genome sequencing to characterize the retrotransposition landscape of patients affected by various neurodevelopmental disorders. To estimate the role of somatic retrotranspositions, they studied diseases characterized by extensive variability of clinical features, including intra-family variability. The authors showed that the number of retrotransposition events is higher in diseased brains compared to healthy ones. They also found out that somatic L1 insertions in neurons mostly target genes associated with neural functions and diseases; this was especially prominent for brain samples from patients with subependymal giant cell astrocytoma, Rett syndrome, ataxia-telangiectasia and non-syndromic autism [39].

#### 3.1.3. TEs in Neurons and Stress

“Stress” is a very broad term for any intrinsic or extrinsic stimulus that evokes a biological response. Stress can induce both beneficial and harmful consequences. It has many effects on human CNS, and may even cause structural changes in the brain. Stress exposure activates autonomous nervous system and the hypothalamus–pituitary–adrenal axis and triggers the release of glucocorticosteroids and catecholamines. The hippocampus is the brain region with the highest density of glucocorticosteroid receptors and the highest level of response to stress. Other key brain structures involved in stress response are the amygdala and temporal lobe [476]. Many complex human diseases, including neurological ones, are affected by chronic and/or severe types of stress, and the control of TE activity is partly lost in many of these diseases. Many TFs involved in the stress response (such as the glucocorticoid receptor) regulate expression of epigenetic modifying enzymes (DNA-methyltransferases or histone methyltransferases) that participate in TE silencing [384,467]. There are most likely other mechanisms of TE regulation in stress conditions. Understanding of such mechanisms is hindered by the fact that, during stress, preprogrammed molecular mechanisms work in the background of multiple random, unspecific events that only happen because normal cell homeostatic mechanisms are strained. There are numerous reports of TE activity in the brain or other tissues being affected by stress, and the observed patterns seem to be quite complex.

Hunter et al. reported repressive histone H3 lysine 9 trimethylation of TEs and decrease in SINE and ERV transcription levels in the hippocampus of rats subjected to acute restraint stress. Interestingly, while there were no changes in H3K9me3 level in the cerebellum, RT-qPCR showed L1 and ERV transcription upregulation by acute stress [196]. In the study of mechanisms underlying stress-enhanced fear learning in rats (a model of PTSD) gene expression profiles in the amygdala were examined 3 weeks after the exposure. One surprising finding was a statistically distinct module containing tightly coregulated transcripts that mapped to L1 elements. All these transcripts were highly upregulated in the basolateral amygdala of rats subjected to stress-enhanced fear learning [390]. A study by Cappucci et al. showed that neuronal L1 expression in stress conditions depends on the genetic background. BALB mice subjected to five daily restraint sessions showed increased L1 transcription in the hippocampus, while in the hippocampi of C57 mice L1 expression was downregulated after the same stress experience [391].

There are some data about early-life stress influencing TE activity. The paper by Bedrosian et al. about low maternal care causing L1 mobilization in the hippocampus [385] has been discussed above, in the Section 2.5. Cuarenta et al. used the neonatal predator odor exposure paradigm to model early-life stress in rats. On the last day of exposure (postnatal day 3), the authors measured L1 ORF1 and ORF2 DNA copy number and mRNA expression in the amygdala, hippocampus and prefrontal cortex of the experimental animals by qPCR. It was found that stress causes an increase in ORF1 copy number in the hippocampus, but only in males. The differences in mRNA expression after the stress exposure were also observed only in male animals: ORF1 mRNA level increased in both amygdala and prefrontal cortex, while ORF2 mRNA level increased only in amygdala [477]. The same group then studied delayed effects of early-life stress. L1 mobilization and expression after the neonatal predator odor exposure were assessed like in the previous work, with the difference that the animals were sacrificed at postnatal day 33, a month after the exposure. In this case, the only significant effect of stress on L1 activity was the increase in ORF1 copy number in the amygdala. Behavioral tests showed that ORF1 copy number in the amygdala negatively correlated with juvenile social-play levels. The authors also observed sex differences in L1 copy number and transcription levels in all three studied regions [478]. Nätt et al. showed that high cortisol level in hair samples (a common biomarker for chronic stress) correlates with a genome-wide decrease in DNA methylation (measured in whole blood DNA) in preschool children. The authors found that cortisol-associated differential methylation generally targets SINEs, but not other types of repeats. SINEs were associated with both hypo- and hypermethylated regions in this study, more so with the latter. Hypomethylated, but not hypermethylated differentially methylated regions (DMRs) were strongly associated with genes. Many of these genes are involved in neurodevelopment and calcium transport. More than a half of the DMR-associated genes had previously been linked to disease, mostly with neuropsychiatric disorders, especially those related to aging [479].

#### 3.1.4. TEs in Neurons during Normal Aging

Aging is a result of gradual accumulation of damage in cells and tissues [469]. A phylogenetically conserved feature of aging is induction of stress response pathways that was confirmed in all systems studied, including whole organism analysis of invertebrates and analysis of the brain of different mammalian species [480]. There is also evidence that human aging is associated with mild chronic inflammation in the absence of infection, revealed by elevated levels of inflammatory biomarkers [481]. Old age is a major risk factor for neurodegenerative disorders. Moreover, even normal aging in healthy people includes decrements in learning and memory, attention, decision-making speed, sensory perception and motor coordination, and even reduction in brain size [476]. Many pathological features of neurodegeneration are aggravated hallmarks of normal brain aging (including genomic instability and epigenetic alterations), suggesting that neurodegeneration may be actually seen as accelerated aging [468,469]. One of the most important mechanisms in neuronal aging and age-associated cognitive impairment is oxidative damage [482]. Just like in the case of stress, studying the role of TEs in aging is complicated by preprogrammed mechanisms coinciding with unspecific cell damage consequences.

TEs, and L1 elements in particular, become unsilenced in different organs and cell types during aging. De Cecco et al. discovered that, during replicative senescence in human fibroblasts, chromatin undergoes extensive changes, and expression levels of L1 and Alu elements and L1 genomic copy number are increased [483]. These observations were confirmed in vivo—expression levels of murine LTRs, LINEs and SINEs are elevated in liver and skeletal muscle samples of older mice [484]. SIRT6 (sirtuin 6) is a multifunctional protein participating in cell stress response, transcription regulation and DNA repair, and SIRT6 depletion is strongly associated with premature aging [485]. SIRT6 is involved in silencing of L1 elements. SIRT6-deficient cells show excess genomic instability, DNA damage, accumulation of cytoplasmic L1 cDNA and strong IFN-I response; this is prevented by RT inhibitors or L1 RNA KD. SIRT6 KO mice show a strong activation of L1 in multiple tissues (including the brain) and develop a severe premature aging phenotype; treatment with RT inhibitors improves the health and lifespan of these mice [486].

Elevated TE expression was reported in aging brain. Van Meter et al. examined L1 expression in brain, liver and heart tissue from young (4 months old) and old (24 months old) mice. The brain exhibited the most significant deregulation of L1 expression with age: in old brain tissue L1 expression increased more than seven fold compared to the young brain. In contrast, in the old liver, L1 expression level was 65% higher than in the young liver, and no significant difference in L1 expression between old and young heart was found [487]. Ramirez et al. analyzed TE RNA levels in forebrain lysates from B6C3HF1 mice aged 6, 12 and 20 months from publicly available RNA-seq data. Among all TEs differentially expressed between age groups, 92.7% were upregulated in 20-month-old mice; most of these TEs were ERVs [488]. In *Drosophila* brain, expression of R1, R2 and *gypsy* TEs increases with age. Mutations of *dAgo2* (encoding an Argonaute protein) cause overexpression of TEs in brains of young flies to the levels comparable with those in older flies. This is accompanied by rapid age-dependent memory impairment and shortened life span [489].

#### 3.1.5. TEs, Autoimmunity and Neuroinflammation

It is now well-known that autoimmunity and inflammation are among underlying pathological mechanisms in multiple neurological disorders [56,490,491,492].

The most obvious connection between TEs and inflammation exists for endogenous retroviruses. dsRNA and RNA:DNA hybrids produced during ERV transcription and mobilization are known to trigger various innate immunity cascades in different cell types [297,310]. HERV proteins are also capable to trigger antiviral responses in the absence of exogenous viral infection [32,309]. Experiments with non-neuronal cells, neurons and astrocytes showed that type-I interferon (IFN-I) upregulates transcription of HERV-K via interferon-stimulated response elements in the viral promoter, meaning there is a positive feedback loop increasing inflammatory response [248]. HERVs were implicated in various neuropathologies [56], including multiple sclerosis [493], amyotrophic lateral sclerosis [248], schizophrenia and bipolar disorder [494], and Alzheimer’s disease [495].

L1 elements are also able to cause immune responses. As mentioned in the previous section, there is a link between cellular senescence, increased L1 expression and activation of the IFN-I response demonstrated in various cell types of human and mice [486,496]. Elevated L1 DNA content has been reported in several neurological disorders associated with autoimmunity, including Rett syndrome, ataxia telangiectasia, schizophrenia and Aicardi–Goutières syndrome. The abundance of L1 DNA in these disorders is most likely a result of accumulation of L1 DNA molecules not integrated into the nuclear genome. Accumulation of DNA in the cytoplasm may trigger immune responses mediated by various sensor pathways [116].

#### 3.1.6. TEs and Mitochondrial Dysfunction in the Context of Neurodegeneration

The human brain has the highest energy requests among all organs. Neurons utilize highly oxidative glucose metabolism, making neuronal mitochondrial function especially important and vulnerable. Beyond their function in energy metabolism, mitochondria are also involved in cellular Ca^2+^ homeostasis, and calcium signaling plays a fundamental role in neuronal function. Mutations in mitochondrial DNA are associated with aging-related pathological processes in neurons [482]. Notably, in mammals, mitochondrial DNA only encodes 13 proteins, and more than 1000 other proteins necessary for mitochondrial function are encoded by nuclear DNA (mitonuclear genes) [497]. Thus, TE-related alterations of the nuclear genome can affect mitochondrial function.

Larsen et al. suggested that cell stress at early stages of neurodegeneration may be caused by somatic mutations in mitonuclear genes. These genes were found to be enriched with Alu elements (compared with protein-coding genes not related to mitochondria). Alu-mediated mutations were documented in a number of mitonuclear genes implicated in various neuropathologies, such as *ABCD1* being associated with adrenoleukodystrophy, *OPA1* being linked to autosomal dominant optic atrophy, or *NDUFS2* and *PDHA1* being involved in neurodegeneration [498]. There are also 57 mitonuclear genes with A-I editing occurring within putative Alu exons; in 52 cases, the editing changes protein sequences. Many of these genes encode proteins involved in essential neuronal processes [451].

Baeken et al. confirmed the involvement of L1 activation in mitochondrial dysfunction and related neurodegeneration using in vitro and in vivo models of Parkinson’s disease [499]. This paper is described below in the Section 3.2.2.

TE RNA accumulation may activate innate immunity machinery, which has been discussed in the previous section; it must be noted that these processes also increase ROS production in mitochondria, which contributes to cell death [471].

### 3.2. Specific Neurological Diseases with Reported Changes in TE Activity

Here, we compile some reports about changed TE mobilization and/or expression in various neuropathologies. While some of these studies suggest specific molecular mechanisms of TE involvement in disease, many papers cited below merely provide observations that are yet to be explained.

#### 3.2.1. Alzheimer’s Disease and Tauopathy

Alzheimer’s disease (AD) is the leading cause of dementia. At the tissue level, the main features of AD pathology are extracellular neuritic plaques and intracellular neurofibrillary tangles, comprised of aggregated, misfolded Aβ peptide and Tau protein, respectively. Other abnormalities such as astrogliosis, microglial activation and cerebral amyloid angiopathy are often observed; the downstream consequences include neurodegeneration leading to macroscopic atrophy. The vast majority of AD cases are sporadic, but it is now thought that ~70% of AD risk is attributable to genetic factors. The ε4 allele of the *APOE* gene is the single biggest risk for sporadic AD. Having one or two copies of this allele increases the risk 3- or 12-fold, respectively [500].

Guo et al. analysed cortical transcriptomes of elderly people with and without AD and found evidence of global TE transcriptional activation in AD. Activation of three ERV clades correlated with global cognitive performance in the year proximate to death. A total of 9 individual TEs that were associated with Tau tangles were identified: a single L1, 5 SINEs, and 3 ERVs. The authors used two *Drosophila* genetic models to establish the link between TE expression levels and tauopathy. Several TEs showed transcriptional activation in neurons in one or both Tau transgenic strains. For some elements, activation was further enhanced with aging. Tau pathology was previously linked to chromatin reorganization, and the authors supposed that it may derepress silenced TEs, which, in turn, may cause genome instability and neuroinflammation [501]. The role of chromatin reorganization by Tau was confirmed by Sun et al.: they found that most TEs that were found to be upregulated in Tau transgenic *Drosophila* are similarly upregulated in flies with mutations that cause depletion of constitutive heterochromatin. Tau^R406W^ fly brains also showed downregulation of multiple piRNAs, as well as decreased PIWI protein level. Pan-neuronal *piwi* overexpression reduced neuronal death in Tau^R406W^ flies, suggesting a causal contribution of PIWI-piRNA mechanism dysfunction in Tau-induced neurotoxicity. Accordingly, piRNA system disruption caused by mutations of *flamenco* (a locus encoding a major piRNA cluster) enhanced neuronal death and exacerbated Tau-induced locomotor deficits in Tau^R406W^ flies (but not in flies without the Tau-encoding gene) [502]. More recently, the same research group made similar observations in murine models of tauopathy, rTg4510 and PS19 transgenic mouse strains. L1 and ERV genomic copy numbers were increased in the cortex of 12-month-old rTg4510 mice compared to control animals. TE (predominantly ERV) transcription levels were increased in CNS samples from 9-month-old rTg4510 and PS19 mice. Publicly available RNA-seq data from JNPL3 tauopathy strain (but not from APdE9, an Aβ-based mouse model of AD) also showed TE upregulation. Gag protein level was found to be elevated in frontal cortex of 6- and 12-month-old rTg4510 mice [488]. Interestingly, *MAPT* gene encoding Tau protein incorporates multiple Alu elements and Alu-related A-I editing sites, which may be involved in the formation of pathogenic Tau isoforms [451].

The role of TE demethylation in AD pathogenesis was suggested. Bollati et al. evaluated methylation of repetitive elements in peripheral blood leukocytes from AD patients and matched healthy control subjects. Alu methylation was decreased in AD samples, while L1 methylation level was increased in this group. However, the observed changes were small in size. Methylation level of L1 or Alu did not correlate with age, APOE status, or CSF levels of Aβ and Tau [503].

The role of retrocopying in AD was also demonstrated. Lee et al. studied small populations of nuclei from AD cerebral cortex samples to detect mosaic alterations of *APP* gene encoding Aβ. RT-PCR detected unusual APP mRNA isoforms with loss of central exons, and some of these isoforms were also detected in genomic DNA samples. Sequencing experiments showed that *APP* retrocopies with exon loss and single-nucleotide variations were more prevalent in AD than in control samples. Two *APP* retrocopy variants with exon loss were also identified in neurons of J20 mice (a genetic model of AD); importantly, one of these variants accumulated with age. When the cytotoxicity of three proteins encoded by atypical APP mRNA variants was examined, two of them could induce cell death. Experiments with cell cultures showed that incorporation of *APP* retrocopies into the genome only happens if DNA damage is induced; application of RT inhibitors prevented *APP* retrocopy formation. The reverse transcriptase responsible for *APP* retrocopy formation was not identified, but the authors do not exclude the possibility that it is ORF2p [504].

L1 mobilization is likely not involved in AD pathogenesis, since no significant difference in L1 genomic copy number in brain and blood samples between AD patients and age-matched controls without dementia was detected by qPCR [505].

#### 3.2.2. Parkinson’s Disease

Parkinson’s disease (PD, or parkinsonism) is the second most common neurodegenerative disorder after AD. The main histopathology associated with PD is degeneration of mesencephalic dopaminergic (mDa) neurons in the *pars compacta* of the *substantia nigra* (SNpc). The most prominent symptoms are motor abnormalities—resting tremor, bradykinesia, rigidity and postural reflex impairment. As the disease progresses, decreased cognitive ability may appear as well. The etiology of PD is complex, involving both genetic and environmental factors. Molecular mechanisms of PD pathogenesis are still largely unknown, even though at least 11 genes have been associated with PD. The most important of these genes is *PRKN* (*PARK2*) [506].

Blaudin de Thé et al. studied L1 expression in mDa neurons using diverse models of PD. For in-vitro experiments, the authors used primary cultures of ventral midbrain neurons subjected to oxidative stress by H_2_O_2_ application. For experiments in vivo, either *En1*^+/−^ mutant mice or mice injected with 6-OHDA were used. *En1* encodes Engrailed-1, a homeobox TF specifically expressed in adult mDa neurons; in heterozygous *En1*^+/−^ mice, mDA neurons show a progressive degeneration that starts at 6 weeks of age. Used in this study 6-OHDA is a drug that induces oxidative stress specifically in mDA neurons. The authors observed increased L1 expression in SNpc mDA neurons of 6-week-old *En1*^+/−^ mice compared to WT siblings. In experiments with acute oxidative stress in both cultured midbrain neurons and adult mDA neurons in vivo, DSB formation correlated with enhanced L1 expression. The causal role of L1 activity in oxidative stress in neurons was confirmed in further experiments. The authors inhibited L1 expression in vitro and in vivo by four independent methods: transcriptional repression by Engrailed-2 application, a siRNA targeting L1 transcript, application of RT inhibitor stavudine, and ectopic overexpression of *Miwi* (leading to L1 silencing by RNA interference). In all these models, decreased L1 transcription led to less prominent oxidative stress. These results confirm that L1 overexpression in mDA neurons triggers oxidative stress-induced DNA damage. The suggested role of Engrailed proteins in these neurons is neuroprotection through the repression of L1 transcription [507]. In human dopaminergic LUHMES cells, treatment with MPP+ or rotenone (respiratory chain complex I inhibitors used to model PD) caused an increase in ORF1p expression. Similar ORF1p expression increase was registered in midbrain lysates from mice treated with a metabolic precursor of MPP+ that selectively targets dopaminergic cells. Experiments with cell cultures also showed that respiratory chain complex I inhibitors cause a decrease in overall DNA methylation level, suggesting a potential mechanism of TE mobilization [499]. Pfaff et al. utilized whole-genome sequencing data to investigate the role of retrotransposition-competent L1 elements (RC-L1) in PD. The authors identified 22 reference and 50 non-reference RC-L1 loci that are polymorphic for their presence or absence. A total of 18 of these loci were determined to be highly active. Healthy controls and PD subjects were categorized based on the total number of highly active RC-L1s. In the control group, 45% of individuals had ≥9 highly active RC-L1 in comparison to the 57.2% of PD subjects. PD patients with a higher number of highly active RC-L1s showed an increase in markers of disease progression such as the total levodopa equivalent daily dose or the degree of DA neurons degeneration. The authors hypothesize that the RC-L1s may not trigger neurodegeneration but likely contribute to the process and aggravate the cell stress [508]. In a study where the methylation of the L1 promoter in blood mononuclear cells of subjects recently diagnosed with idiopathic PD was measured, no correlation between this parameter and the disease was found; however, the inverse correlation between PD and ever regularly smoking tobacco was strongest for subjects with the lowest L1 promoter methylation [509].

Whole-genome sequencing of three families with early onset PD revealed five structural variations in the *PRKN* gene; retrotransposon sequences were identified within 1–2 Kb of the deletions’ break points, suggesting that all identified variations might originate through retrotransposition events [472].

#### 3.2.3. Huntington’s Disease

Huntington’s disease (HD) is a dominantly inherited neurodegenerative disorder, characterized by progressive motor impairment, cognitive decline and psychiatric disturbances. HD patients show selective loss of medium-sized spiny neurons of the striatum; at later stages, pathologic alterations happen in other brain regions. HD is caused by mutations in *HTT*, the gene encoding huntingtin. Pathological alleles of *HTT* have an increased number of CAG repeats, and as a result mutant protein forms have an expanded polyglutamine tract (polyQ) and the tendency to aggregate [510]. A link between huntingtin polyQ-expansion and chromosomal instability was also suggested [511].

In a murine model of HD, it was shown that increased L1 mobilization may be involved in HD pathogenesis. L1 DNA copy number in HD mouse brain caudate regions was increased compared to WT control. Expression of ORF1 and OFR2 RNAs and proteins was detected in brain caudate of HD but not WT mice. Pyrosequencing showed that, in HD mice, L1 regions are less methylated. Increased L1 content in HD mouse neurons correlated with increased cell death and dysregulation of cell survival signaling pathways (indicated by decreased mTORC1 activity and AMPKα level). Overexpression of L1 ORF2, but not ORF1, in 293T cells caused similar dysregulation of cell survival signaling and cell death [512]. Conversely, in another study, L1 expression was found to be decreased in HD. Analysis of published RNA-seq data of two HD mouse models showed decreased expression of some specific L1 RNAs in the striatum, but not the cortex, of transgenic mice expressing pathological alleles of human *HTT* (compared with mice expressing its normal allele). L1 downregulation in striatum was more prominent in mice expressing the *HTT* allele with more CAG repeats. In human HD post-mortem brain samples (prefrontal cortex Brodmann area 9 that is involved in HD pathogenesis but suffers less neuronal death than striatum), L1 expression was downregulated compared with normal controls, but the difference was not significant [510]. TE expression and mobilization were found to be increased in a *Drosophila* HD model; moreover, inhibition of TE mobilization with RT inhibitors rescued polyQ-dependent eye neurodegeneration and genome instability and increased fly lifespan [513].

#### 3.2.4. Ataxia Telangiectasia

Ataxia telangiectasia (AT) patients exhibit progressive neurodegeneration and eventual death in the second or third decade of life. The disease is caused by mutations in the *ATM* gene, which plays an important role in DNA damage signaling. Coufal et al. performed KD of ATM in hESCs that were then differentiated to NPCs and transfected with L1-EGFP retrotransposition reporter. L1 retrotransposition level in ATM-deficient NPCs was increased, but these cells were still able to differentiate into functional neurons. L1 expression was not increased in ATM-deficient cells; insertion of L1 copies in unusual sites of the genome was not registered. However, L1 insertions in ATM-deficient cells were found to be longer (with truncation being less frequent). The authors suppose that ATM may recognize intermediates generated during the process of L1 integration as DNA damage and initiate its repair, reducing the length of the resultant retrotransposition events. The correlation between ATM deficiency and elevated retrotransposition level discovered in cultured NPCs was confirmed in vivo using *ATM* KO mice with human L1 and EGFP retrotransposition reporter in their genome. Brain samples of 3-month-old *ATM* KO mice had more EGFP-positive cells compared with WT animals. The most noticeable increase was observed in the hippocampus, and EGFP-positive cells were not apparent in non-neuronal tissues. In the same study, L1 DNA copy number was found to be increased in AT postmortem brain samples compared with matched controls. It is not clear whether L1 retrotransposition contributes to AT pathogenesis [58].

#### 3.2.5. Spinal Muscular Atrophy

Spinal muscular atrophy (SMA) is a devastating childhood motor neuron disease that often leads to death. The major pathological characteristic of SMA is selective degeneration of lower α-motor neurons in the ventral horn of the spinal cord, resulting in progressive muscle denervation, skeletal muscular atrophy and eventual paralysis. SMA is a rare example of a monogenetic neurodegenerative disease: the causative factor of SMA is the loss of SMN, a protein encoded by the *SMN1* gene. The loss of SMN affects not only motor neurons but also some other neurons, muscle cells and some other cells and tissues. SMN is a multifunctional protein, it is expressed ubiquitously, and homozygous deletion of *Smn* in mice is lethal. Humans have two *SMN* genes, and the loss of SMN in SMA is partially compensated by expression from *SMN2*. The *SMN2* gene does not fully substitute for *SMN1* because almost 90% of *SMN2* transcripts lack exon 7, producing a truncated SMNΔ7 protein that is quickly degraded. Disease severity varies according to the number of *SMN2* copies carried by the patient. In the most severe form (one *SMN2* copy), disease onset occurs before 6 months of age with death of respiratory distress usually within 2 years. With multiple *SMN2* copies, the disease is characterized by an early-adulthood clinical onset and typically a normal life expectancy [514,515].

Each *SMN* gene contains more than 60 Alu-like sequences. Some *SMN1* mutations in SMA patients were found to be caused by Alu-mediated recombinations, and it was suggested that more yet-unidentified Alu-related mutations in this gene may lead to SMA. However, SMA is a rare example of a neuropathology in which TE-related mechanisms may not only cause the disease but also compensate the damage. Ottesen et al. identified a novel exon within *SMN* gene sequence, 6B, generated by exonization of an Alu element located within intron 6. Transcripts containing exon 6B are not abundant because they are prone to nonsense-mediated decay and Staufen-mediated mRNA decay. However, these transcripts were found in all tissues. The protein produced from these transcripts, SMN6B, has the same length as normal SMN but has different 16 amino acids on the *C*-terminus. SMN6B is less stable than SMN but more stable than SMNΔ7. SMN6B was shown to have similar intracellular localization as SMN and is able to interact with Gemin2, a protein partner of SMN. It is very likely that SMN6B may substitute for SMN loss better than SMNΔ7. However, levels of SMN6B produced in SMA patients remain unknown [516].

#### 3.2.6. Amyotrophic Lateral Sclerosis and Fronto-Temporal Lobar Degeneration

Two other neurodegenerative disorders—amyotrophic lateral sclerosis (ALS) and fronto-temporal lobar degeneration (FTLD)—were linked to TE activity in the brain and share mechanisms of pathogenesis. Both diseases typicaly start after the age of 50. ALS is a progressive motor neuron degeneration, and increasing muscle weakness leads to respiratory failure and death in 3–5 years. ALS is usually sporadic—only 5–10% of cases are inherited [517]. At least 25 genes have been linked to ALS, the most important of them are *SOD1* (superoxide dismutase) and *C9ORF72* [518]. FTLD is a heterogeneous syndrome characterized by progressive decline in behavior or language. FTLD patients show gross atrophy of the frontal and anterior temporal lobes: loss of pyramidal neurons, cortical gliosis, loss of myelin and axons. FTLD is a common cause of early-onset dementia. Survival of patients with FTLD is shorter, and their cognitive decline is more rapid compared with AD. FTLD patients may suffer from ALS at the same time. Up to 40% of FTLD patients have a history that suggests familial transmission. Most inherited FTLD cases are caused by mutations in the genes encoding Tau protein and progranulin, leading to abnormal protein aggregation in neurons [519]. Frontotemporal dementia (FTD) is a subgroup of FTLD-associated disorders [520].

One of the key proteins involved in both diseases is TDP-43 (trans-activation response DNA-binding protein of 43 kDa), a multifunctional RNA-binding protein [120,519,521]. Accumulation of TDP-43-containing cytoplasmic inclusions is a shared pathological hallmark in a broad spectrum of neurodegenerative disorders [112,521]. TDP-43 has thousands of RNA targets [120] and multiple functions, including alternative splicing, regulation of mRNA stability and miRNA biogenesis [112]. Some data indicate that TDP-43 must be a stress-responsive RNA-associated factor [521]. In mouse models of TDP-43 proteinopathy, transcription and splicing of hundreds of RNAs were misregulated [522,523].

Li et al. re-analyzed the accessible data of deep sequencing of RNA targets co-purifying with immunoprecipitated TDP-43 from a few studies performed on mice [522,523,524], rat [524] and human [525] samples. An extensive binding of TDP-43 to RNA derived from all major classes of TEs (LINE, SINE, LTR and DNA transposons) was revealed. It was found that, in mice with disrupted TDP-43 function, multiple LINE, SINE and LTR transcripts are upregulated. In cortical tissue of FTLD patients, binding between TDP-43 and its TE targets is selectively and dramatically reduced compared to control subjects. When similar analysis was performed for FUS, another RNA-binding protein implicated in neurodegenerative disorders, no preferential interaction of FUS with TE-derived RNAs was found. All these findings suppose that TDP-43 normally plays a protective role by specifically binding and silencing multiple transposon transcripts [112]. Liu et al. used subcellular fractionation and FACS to enrich for diseased neuronal nuclei without TDP-43 from post-mortem FTD-ALS human brain samples. The authors showed that loss of nuclear TDP-43 is associated with chromatin decondensation around LINE elements and increased L1 DNA content. The authors noted some specificity toward decondensation of LINEs compared with other repeat elements, but the exact mechanism of this is unclear [526].

Conversely, some other studies suggest that TDP-43 causes activation of TEs (ERVs in particular) and contributes to neuropathology. Krug et al. expressed human TDP-43 in *Drosophila* neurons or glia and reported that it disrupted siRNA silencing, caused derepression of LINE and LTR transposons and led to age-dependent neurological deterioration and reduced lifespan. In fly glial cells (but not in neurons), TDP-43 expression caused unsilencing of *gypsy* (an ERV that is related to human HERV-K). It was found to causally contribute to degeneration, because inhibiting *gypsy* genetically or pharmacologically was sufficient to rescue the phenotypic effects [527]. Combined ectopic expression of TDP-43 and HERV-K in cultured human cells increased HERV-K transcription and reverse transcriptase activity (as compared with cells cotransfected with HERV-K and a control plasmid); KD of endogenous TDP-43 decreased HERV-K transcription. A total of 5 binding sites for TDP-43 in HERV-K promoter (LTR) were identified. In the same study, it was found that HERV-K is activated in a subpopulation of patients with sporadic ALS. Env protein was detected in some cortical and spinal neurons of these patients. Transgenic mice expressing the *env* gene under a neuron-specific promoter showed an expression pattern of the Env protein similar to that of ALS patient tissues. Mutant mice after 2–3 months of age displayed progressive motor dysfunction, selective atrophy of the motor cortex, abnormalities in neuron shape and function, and DNA damage; however, the exact mechanism of HERV-K Env neurotoxicity is not well understood. The authors suggest that virus particles may be transmitted between neurons, explaining the spread of the disease along anatomical pathways in ALS [30].

Pereira et al. suggested that ORF1p protein aggregation may be involved in ALS pathogenesis. They showed that ALS-associated mutant forms of FUS, TDP-43, and SOD1 proteins, but not WT proteins, strongly colocalize with ORF1p in granules in cell cultures. Experiments with cell culture transfections showed that two ALS-associated mutant forms of TDP-43 fail to inhibit L1 retrotransposition with the same efficiency as the WT TDP-43. A similar observation was made for one ALS-associated mutant form of angiogenin. Conversely, two different ALS-associated mutant forms of FUS reduced L1 retrotransposition, while the WT FUS had no such effect [518].

Analyses of transcriptomes of postmortem cortex samples with machine-learning algorithms showed that ALS is most likely a heterogeneous condition, and TE activation was only observed in one of three distinct molecular subtypes of ALS, comprising 20% of all examined samples. In this subtype, altered expression of genes in multiple pathways linked to TDP-43 was also noted [528].

#### 3.2.7. Fragile X-Associated Tremor/Ataxia Syndrome

Fragile X-associated tremor/ataxia syndrome (FXTAS) is a late-onset neurodegenerative disorder characterized by a progressive action tremor with ataxia. Patients have degeneration in the cerebellum (including Purkinje cell loss and Bergman gliosis) and intranuclear inclusions in neurons and glia throughout the brain. FXTAS is distinct from fragile X syndrome, but is also caused by an increased number of CGG trinucleotide repeats in the 5′-UTR of the *FMR1* gene. Tan et al. studied a *Drosophila* FXTAS model (rCGG-repeat transgenic flies) that shows a distinct neurodegenerative eye phenotype. The authors measured mRNA levels of different TEs in the brain of mutant flies and registered increased expression of *gypsy* and *copia* LTR elements. Derepression of *gypsy* expression achieved by crossing rCGG-repeat transgenic flies with another mutant line was shown to enhance neurodegenerative eye changes. Accordingly, reduction in *gypsy* expression by RNAi rescued neurodegenerative eye changes. The authors suggest that rCGG-repeat RNA and TE RNA share the same binding proteins, and accumulation of rCGG-repeat RNA leads to sequestration of these proteins and TE unsilencing that contributes to neurodegeneration [470].

#### 3.2.8. Multiple Sclerosis

Multiple sclerosis (MS) is a complex multifactorial disease of the CNS. Pathogenesis of MS involves neuroinflammation, death of oligodendrocytes, demyelination, axonal damage and defects in myelin repair, which result in physical and cognitive disabilities [311,529]. In MS, infiltrating lymphocytes and macrophages are observed in the perivascular space of the CNS [530]. Environmental factors, viral infections in particular, are able to trigger MS pathogenesis. Multiple studies suggested a correlation between expression of HERVs (mostly HERV-W) and onset and progression of MS (reviewed in [493]).

Two specific ERVs of the HERV-W family are activated in MS: MSRV (MS-associated retrovirus), and ERVWE1 [531]. Env proteins encoded by these two ERVs are called, respectively, MSRVenv and syncytin-1 (the latter also has a function in placenta formation [296], discussed in the Section 1.8.9). MSRVenv and syncytin-1 are highly homologous (>94% identity at the RNA level) and very difficult to discriminate. A major difference between MSRV and ERVWE1 is that only MSRV is found as an extracellular virus, visualized by electron microscopy, while syncytin-1 is detected only intracellularly or on the plasma membrane [531]. MSRVenv is detected on macrophages or microglial cells in active plaques of MS brains [311]. Similar observations were reported for syncytin [529].

Experiments in cell cultures showed that ectopic expression of syncytin in human fetal astrocytes (HFAs) and monocyte-derived macrophages (MDMs) increased production of proinflammatory cytokine IL-1. Conditioned medium from HFAs (but not MDMs) ectopically expressing syncytin was highly cytotoxic to human or rat oligodendrocytes [529]. Application of MSRVenv stimulated both adhesion of HL-60 cells (promyeoloblasts) to the monolayer of HCMEC/D3 cells (endothelium) and transmigration of HL-60 cells through the endothelial monolayer. MSRVenv induced production of pro-inflammatory cytokines in HCMEC/D3 cells and primary endothelial cultures. In the same study, syncytin had a similar but less prominent effect on primary endothelial cultures. KD of toll-like receptor 4 (TLR4) abolished HCMEC/D3 response to MSRVenv, suggesting that TLR4 is implicated in MSRVenv recognition [530].

In vivo, MSRVenv was shown to induce autoimmunity to myelin proteins and cause experimental allergic encephalomyelitis in mice (an animal model for MS) [311]. Mice infected with viral particles carrying the syncytin-expressing vector developed neuroinflammation, oligodendrocyte and myelin damage. These animals also demonstrated decreased coordination, muscle strength and seeking behavior [529].

It was shown that some exogenous viruses may can cause reactivation of HERV-W copies. *Herpesviridae* are the most common triggers known to upregulate MSRV expression [532].

#### 3.2.9. Aicardi-Goutières Syndrome

Aicardi–Goutières syndrome (AGS) is a very rare hereditary interferonopathy with a neurological phenotype. Neurological symptoms include severe encephalitis, demyelination of motor neurons, and psychomotor retardation. Patients have elevated IFN-I levels in CSF. Many AGS symptoms resemble those caused by congenital viral infection. Mutations in seven genes have been confirmed to produce AGS phenotypes; all these genes are involved in regulating cytosolic nucleic acids [168,169,533]. One of the genes linked to AGS is *TREX1* encoding an anti-viral DNase. Stetson et al. purified cytoplasmic DNA fragments from hearts of WT and *Trex1* KO mice. In mutant mice, this DNA was more abundant and more complex. DNA fragments derived from retrotransposons were enriched in the samples from mutant mice. More specifically, L1-, LTR- and SINE-derived cDNAs were identified. Experiments on cell cultures cotransfected with vectors expressing retroelements and *TREX1* confirmed that TREX1 metabolizes DNA derived from these elements and prevents retrotransposition [168]. Thomas et al. generated TREX1-deficient pluripotent stem cells (two cell lines generated from hESCs with introduced mutations in *TREX1*, and an iPSC line from an AGS patient with a naturally occurring *TREX1* mutation). The authors differentiated these cells into NPCs, neurons, and astrocytes. All these cell types exhibited increased levels of ssDNA, as was shown by immunostaining. Application of RT inhibitors to TREX1-deficient NPCs reduced ssDNA amount to near-control level. Sequencing of extrachromosomal DNA in TREX1-deficient NPCs showed increased level of L1 cDNA, and the relative presence of L1s compared to other transposable elements far exceeded its genomic representation. L1 mRNA levels were elevated in TREX1-deficient astrocytes (but not NPCs and neurons) compared to their WT counterparts. Neurons were found to be more vulnerable to the loss of TREX1 compared to NPCs and astrocytes. TREX1-deficient neurons had downregulation of canonical neuronal markers, altered shape, increased levels of cleaved caspase-3, and excessive DNA breakage. These effects were alleviated by RT inhibitors [169]. Another gene mutated in AGS patients and involved in TE silencing is *SAMHD1* encoding a dNTPase. Zhao et al. demonstrated that mutant SAMHD1 proteins from AGS patients are defective in L1 inhibition. The authors also showed that dNTPase activity may not be critical for SAMHD1-mediated L1 inhibition, since a dNTPase-defective SAMHD1 version maintained the ability to inhibit L1 retrotransposition [170]. Yet another mechanism supposedly linking TE activation with neuroinflammation in AGS is disruption of ADAR1 A-I editing. Rice et al. performed whole-exome sequencing of patients diagnosed with AGS but screened negative for mutations in *TREX1* and a few other genes known to be relevant to the AGS phenotype. The authors found *ADAR1* mutations in some of the studied patients. All tested individuals with mutations in *ADAR1* showed upregulation of interferon-stimulated genes (assessed by RT-qPCR of 15 targets in whole blood samples). The authors speculated that, since most of the known ADAR1 editing substrates are TE-derived RNAs, ADAR1 dysfunction may cause TE dsRNA accumulation which prompts IFN-I response [534].

#### 3.2.10. Glioblastoma

Glioblastoma is the most common of all primary malignant CNS tumors, and one of the most lethal of all human cancers. Glioblastomas display striking cellular heterogeneity and plasticity between different cellular states of the tumor [535].

Quite paradoxically, increased L1 mobilization is not associated with brain cancers, even though one might expect this based on L1 unsilencing in both normal neuronal tissue and various cancer types. A few studies showed very little L1 mobilization in glioblastoma [536,537].

Microarray analysis of cellular and exosomal RNA from a low-passage glioblastoma cell line indicated high transcription levels of retrotransposons. The levels of Alu, L1 and, especially, HERV transcripts were frequently higher in microvesicles than in cells (while RNA from DNA transposons was similar in content in cells and microvesicles). This suggests selective packaging of retrotransposon RNA sequences, especially HERV, in tumor microvesicles. The most abundant HERV species in microvesicles from glioblastoma cells was HERV-H. This finding is consistent with HERV-H being the most active HERV in fetal brain cells; glioblastomas likely derive from dedifferentiated glia cells or neuroprecursor cells [538], even though other versions of glioblastoma origin were suggested [535].

Interestingly, it was also found that expression of L2-derived miRNAs and their targets is disturbed in glioblastoma [539].

#### 3.2.11. Autism Spectrum Disorders

Autism is a set of heterogeneous neurodevelopmental conditions, characterised by early-onset difficulties in social communication, unusually restricted interests and repetitive behaviors. More than 70% of patients have comorbid conditions, about 45% of individuals with autism have ID, and 32% show loss of previously acquired skills. Electrophysiology, neuroimaging and molecular data suggest that patients with autism have atypical neural connectivity. Autism has high heritability, but its genetics is very complex. Multiple loci were linked to this disorder (up to 1000 by some estimations), and many genetic variants contributing to the autism phenotype have a high degree of pleiotropy [540]. The term “autism spectrum disorders” (ASD) includes autism, Asperger’s syndrome and pervasive developmental disorder-not otherwise specified [541]. Heterogeneity in the phenotypic presentation of ASD makes it difficult to identify specific genes involved in pathogenesis of this condition. Hu et al., using multiple clustering algorithms, suggested dividing ASD population into four phenotypic subgroups based on similarity of symptom severity [542].

Shpyleva et al. studied postmortem brain samples of patients diagnosed with autism and matched controls. Four brain regions were considered: three neocortex areas and the cerebellum. L1 expression was found to be increased in the cerebellum but not in the cortex of autism patients. The binding of MeCP2 and histone H3K9me3 to L1 sequences was lower in autism cerebellum. The link between L1 overexpression and oxidative stress was suggested: the increase in L1 RNA level inversely correlated with glutathione redox status, and the expression of FOXO3 (a TF inducible by oxidative stress) was also increased [176]. Tangsuwansri et al. investigated the association between L1 elements and altered gene expression profiles in ASD using published sequencing data. Genes containing L1 insertions were overrepresented among differentially expressed genes (both up- and downregulated) in ASD blood cells. A total of 351 L1-containing genes were differentially expressed in ASD in at least two independent transcriptomic studies. Ingenuity Pathway Analysis showed that 25 of these genes are associated with autism or ID. Neurodevelopmental disorders comorbid with ASD (mental retardation, cognitive impairment, and neuromuscular disease) were also associated with this gene set. It was found that the percentage of overall L1 methylation is decreased in the subgroup of ASD individuals with severe language deficits, but not in other ASD subgroups [543]. The same research group performed similar analysis for Alu elements. It was found that genes containing Alu insertions are differentially expressed (mostly downregulated) in ASD blood cells. A total of 320 Alu-containing genes were shown to be downregulated in more than one study. Ingenuity Pathway Analysis showed association of these genes with neurological diseases, nervous system development and function. A total of 21 genes were associated with autism or ID. There was no significant difference in AluS methylation between ASD and matched control groups when all ASD individuals were combined; however, significant AluS methylation patterns were associated with specific ASD subgroups [544]. Balestrieri et al. studied expression of four HERV families in peripheral blood mononuclear cells (PBMCs) from ASD patients. RT-qPCR showed that HERV-H expression is upregulated, while HERV-W is downregulated in PBMCs from ASD patients. Cultured PMBCs from ASD individuals showed an increased potential to upregulate HERV-H expression upon stimulation with a T-lymphocyte-specific mitogen. It was also found that younger ASD patients have higher levels of HERV-H, and that high HERV-H expression is associated with severe impairment of developmental level in ASD [545].

#### 3.2.12. Rett Syndrome

Rett syndrome (RTT) is a major neurodevelopmental disorder, characterized by mental retardation and autistic behavior and caused by mutations in the *MeCP2* gene. *MeCP2* encodes methyl-CpG-binding protein 2, an epigenetic regulator highly expressed in the nervous system. This gene is located in the X chromosome, and almost all patients with RTT are female (in males with one *MeCP2* copy its mutation is usually lethal). Approximately 99% of RTT cases are sporadic [546,547]). RTT is associated with comorbid autism [176].

Zhao et al. profiled genome-wide somatic L1 insertions in prefrontal cortex samples obtained from RTT patients. Non-brain tissues (heart, eye, and fibroblasts) were also studied. L1 activity in RTT and control subjects was found to be higher in prefrontal cortex compared to other tissues. In cortical neurons of RTT patients, the pattern of insertions was different compared to controls: unexpectedly, genomes of RTT neurons had fewer insertions within exons. L1 depletion in the exons was significant for long (>100 kb) genes (many of which encode important neuronal proteins [360]). The authors suggest that this observation might reflect the negative selection against cells with multiple exonic insertions (especially within important genes). In cells that already have defective MeCP2, such retrotransposition events would presumably compromise the genome integrity so badly that these cells would die [342]. Muotri et al. used fibroblasts taken from a patient with RTT and from a control subject to generate iPSCs. These cells were then artifically differentiated into NPCs. An L1-EGFP reporter construct was used to measure L1 retrotransposition frequency. Retrotransposition was increased about twofold in NPCs generated from cells of the RTT patient. In the same study, qPCR confirmed that L1 genomic copy number in RTT post-mortem brain samples is higher than in control samples [130].

#### 3.2.13. Schizophrenia and Bipolar Disorder

Schizophrenia (SZ) is a devastating chronic illness characterized by psychosis (delusions and hallucinations), alterations in drive and volition, impairment of memory and attention, and affective dysregulation. SZ typically debuts in adolescence and early adulthood. Symptoms and course of the disease vary greatly across individuals [548]. Dorsolateral prefrontal cortex (DLPFC) is one of the brain regions particularly affected in SZ [549,550]. Clinical heterogeneity of SZ is underpinned by considerable genetic heterogeneity. Thousands of common polymorphisms, as well as multiple individually rare CNVs, were suggested to contribute to SZ risk [551]. The inherited component to risk for SZ is estimated at 64%. Genome-wide association studies reported more than 100 genetic loci containing common alleles conveying only minor influences on SZ risk. Rare CNVs increase disease risk substantially, but only in a small fraction of patients diagnosed with SZ [552]. Bipolar disorder (BD) is characterized by mood instability with periods of mania or hypomania and periods of depression. During manic episodes, patients experience psychotic symptoms similar to signs of SZ. Genome-wide association studies suggest genetic overlap between SZ and BD. Many of the risk alleles identified in these studies are implicated in both disorders [553]. Family, twin, and adoption studies show that genetic correlation between SZ and BD is about 60% [554]. Considering this, in the present review, we discuss TE involvement in SZ and BD in the same section. Importantly, a genetic overlap between SZ and neurodevelopmental disorders including autism has also been reported [548,551]. On the other hand, involvement of HERV-W elements and myelin inflammation in pathogenesis of SZ and BD links these diseases with multiple sclerosis [494,555].

Guffanti et al. identified 1689 TEs differentially expressed in the DLPFC of SZ patients using a novel bioinformatics approach. Most of these TEs are located within introns of 1137 protein-coding genes [439].

Bundo et al. reported a high L1 copy number in neurons from the prefrontal cortex of SZ patients. Whole-genome sequencing revealed brain-specific L1 insertions localized preferentially to synapse- and SZ-related genes. In animal models of perinatal environmental risk factors for SZ, brain L1 content was also increased [556]. Doyle et al. identified intragenic L1 insertions absent in the reference genome in postmortem DLPFC samples from SZ patients. At least 23 of 63 genes with such insertions identified in the SZ samples from this and previous [556] studies were known to have an association with SZ. Notably, out of 210 insertions identified by Doyle et al., only one is likely identical by its position to an insertion detected earlier by Bundo et al. It is located within an intron of the *NPAS1* gene [552] encoding a TF that regulates multiple genes involved in SZ and ID [557]. Guffanti et al. performed a deep whole-genome sequencing of blood samples from six members of the same family, four of which were diagnosed with SZ, to identify non-reference L1 insertions. Using two different computational algorythms, the authors registered 110 such insertions in this family (about 40 insertions per subject). A total of 52 non-reference L1 insertion loci mapped to protein-coding or RNA genes. It was found that L1 polymorphic insertions in SZ subjects have an impact on genes previously implicated in SZ, such as *GABRB1*, *FHIT*, and *RYR3* [473]. In a recent study, Zhu et al. analyzed L1 and Alu retrotransposition in sorted cells using deep whole-genome sequencing and a machine-learning method. Two brain-specific L1 insertions detected in the same SZ donor were thoroughly investigated. Both insertions were within introns of genes previously associated with neuropsychiatric disorders (*CNNM2* and *FRMD4A*). *CNNM2* is located within a locus that has been associated with SZ, and the L1 insertion within *CNNM2* affected a putative transcriptional regulatory element. For insertions in *CNNM2* and *FRMD4A*, respectively, mosaicism levels quantified using ddPCR were 0.72% and 1.2% in neurons and 0.54% and 0.53% in glia. Both insertions were present in neurons and glia from different brain regions across both hemispheres, indicating that these two retrotransposition events occurred during early embryogenesis. Cell culture experiments with reporter gene system confirmed that these insertions could reduce *CNNM2* and *FRMD4A* expression in the subject’s brain cells [474].

HERV-W RNA was identified in plasma and CSF from patients with SZ and related pathologies [558]. Frank et al. performed a microarray-based analysis of HERV *pol* transcriptional activity in DLPFC samples from individuals with SZ or BD and matched controls. HERV *pol* expression profiles of patient groups were found to be generally similar to those of the control group. The most active HERV elements were equally represented in all three groups. Among less active elements, HERV-IP, NMWV7, and HERV-L were overrepresented in SZ compared to BD specimens. Seq63 and NMWV7 elements were underrepresented in BD samples compared to control samples. Most notably, HERV-K10 was significantly overrepresented in both BD and SZ samples compared to controls [31]. Perron et al. addressed expression levels and mutations in the MSRV *env* gene in peripheral mononuclear cells collected from patients with SZ and BD. MSRV *env* transcription was elevated in both patient groups compared with healthy controls. MSRV *env* overexpression was more prominent in the BD group. Paradoxically, MSRV *env* DNA copy number was lower in the genomes of patients with BD or SZ compared with the control group. Differences in nucleotide sequence of MSRV *env* were found between all three groups. The authors also noted that, in patients with BD or SZ molecular characteristics of MSRV, *env* are different from those reported previously in patients with MS [494].

#### 3.2.14. Major Depressive Disorder

Major depressive disorder (MDD) is a psychiatric disorder defined by episodes of depressed mood lasting for more than 2 weeks with disturbed sleep and appetite, excessive guilt, reduced concentration, slowed movements and suicidal thoughts. MDD affects approximately 17% of the population. Both genetic and environmental factors are thought to be involved in the etiology of MDD. Liu et al. examined DNA samples from peripheral blood cells and showed that MDD patients have increased L1 genomic copy number and decreased L1 5′-UTR methylation level compared with healthy control subjects. In the same study, mice were subjected to chronic unpredictable mild stress as a model of depression. L1 copy number in peripheral blood cells of stressed mice was increased, just like in MDD patients. However, in the prefrontal cortex of stressed mice, L1 copy number was decreased compared with the control group (while no change was found in other studied brain regions) [559].

#### 3.2.15. Post-Traumatic Stress Disorder

Severe stress or trauma may cause permanent changes in brain circuitry, leading to dysregulation of fear responses and the development of post-traumatic stress disorder (PTSD). PTSD is considered an anxiety disorder. Symptoms include re-experiencing the trauma, avoiding stimuli associated with the trauma, numbing of general responsiveness, and increased arousal. Many PTSD patients show an exaggerated reaction to mild stressors. The amygdala is the key brain structure involved in the consolidation and recall of traumatic memories [390]. PTSD is quite common in combat troops after deployment. Rusiecki et al. examined LINE and Alu methylation in serum DNA from US military soldiers with diagnosed PTSD and matched controls; in all cases, serum samples taken before and after deployment were available. It was found that pre-deployment Alu methylation level was higher in PTSD subjects versus controls; these groups did not differ post-deployment. In the control group, but not in the PTSD group, L1 methylation increased post-deployment [560]. An animal model of PTSD is stress-enhanced fear learning (SEFL) where animals are pre-exposed to a stressor before classic fear conditioning. Ponomarev et al. used the SEFL paradigm with and without exposure to isoflurane in the concentration blocking the behavioral effects of stress and performed gene expression profiling in lateral/basolateral amygdala 3 weeks after the exposure. It was found that L1 transcripts were upregulated by shock and downregulated by isoflurane. The authors suggested that L1 expression in SEFL may simply reflect general chromatin remodeling during unsilencing of multiple genes [390].

#### 3.2.16. Drug Addiction and Alcoholism

Drug addiction is a chronic relapsing disorder characterized by maladaptive learning about drugs and predictive cues that leads to compulsive drug seeking. Genetic factors contribute to drug addiction, but it is, primarily, a behavioral learning disorder that only develops after repeated exposure to a drug of abuse. The dopamine signaling pathway involved in reward learning is dysregulated in drug addiction; the most important brain region in this pathway is *nucleus accumbens* (NAc). Recently, the role of epigenetic changes in drug-induced plasticity was discovered [561,562]. Some drugs of abuse were shown to be able to induce L1 retrotransposition.

In experiments with neuronal cell lines, Okudaira et al. demonstrated that methamphetamine and cocaine in non-cytotoxic concentrations trigger L1 mobilization. Other drugs tested in these experiments, ethanol and sodium barbital, did not have the same effect. It was confirmed that methamphetamine and cocaine do not induce DSBs. Both drugs had no effects on the expression or splicing of L1 RNA. A mutation in the reverse transcriptase domain of ORF2 prevented the drug-induced retrotransposition. It was also shown that methamphetamine or cocaine recruited ORF1p to chromatin in a CREB-dependent manner [187]. A few other studies showed a link between cocaine administration and TE activation in vivo. Voskresenskiy et al. administered cocaine chronically to pregnant rats and registered increased L1 ORF1 expression in blood cells of the mothers and in brain, spinal cord and heart of the offspring at the first day of life [563]. Maze et al. reported that, in mice, both acute and repeated cocaine administration dynamically alters H3K9me3 distribution and heterochromatin formation in NAc (but not in other brain regions implicated in cocaine action). ChIP-Seq experiments showed that repeated cocaine administration significantly changed H3K9me3 level at 16 repetitive element types, and for 11 of these H3K9me3 level was reduced. The most prominent H3K9 demethylation happened in L1 elements, and L1 unsilencing was confirmed by RT-qPCR [564]. Doyle et al. studied postmortem samples of medial prefrontal cortex from individuals with cocaine addiction. The authors used L1-seq, a specialised method to identify specific L1 insertions by their flanking sequences, in sorted neuronal and non-neuronal nuclei. Novel L1 insertions were identified in neurons from both addiction and control brain samples. Notably, in the addiction group, these insertions were enriched in GO terms and pathways previously associated with cocaine addiction. Comparison between different samples suggested that most L1 insertions identified in the addiction group are either germline or occurred de novo at an early developmental stage. This means that these insertions could influence predisposition to the addiction, rather than be a consequence of cocaine intake [565].

Another drug of abuse associated with TE activation is morphine. Trivedi et al. characterized the influence of morphine on DNA methylation in cultured neuronal SH-SY5Y cells. L1 methylation (assessed by the MethylCap-Seq method) was increased after 4 h of morphine treatment and decreased after 24 h of treatment. Correspondingly, L1 RNA expression was decreased after short-term morphine treatment and increased after 24 h of treatment. Changes in L1 methylation were in agreement with changes in global DNA methylation induced by morphine, even though methylation level of individual CpG regions in L1 was not predictive of global methylation level [566]. Sun et al. demonstrated that H3K9me2 repressive mark is downregulated in mouse NAc (but not in dorsal striatum) by repeated morphine administration. ChIP-seq was then used to map the distribution of H3K9me2 binding, and 24 types of repetitive elements were found to have significant morphine regulation, most of them showing decreased H3K9me2 enrichment and de-repression. Among unsilenced TEs were five L1 subfamilies [567].

Ponomarev et al. performed a gene expression profiling in postmortem human brain samples of alcoholics. It was shown that alcohol abuse resulted in upregulation of LTR retrotransposons. Overrepresentation analysis of coexpression modules identified several modules that showed significant enrichment with TEs in all brain regions. Many of these TEs have retained functional promoters, and DNA methylation in the LTR region of these transposons was decreased. The authors suggested that these TE promoters may regulate the expression of adjacent genes [568]. Only one Illumina probe corresponding to a retrotransposition-competent L1HS family element was used in this study, and this element was found to be highly upregulated in alcoholic brain [120] (Ponomarev, personal communication).

#### 3.2.17. Creutzfeldt-Jakob Disease

Creutzfeldt–Jakob disease (CJD) is a rare, fatal neurodegenerative disorder caused by prion proteins. The most common form of CJD is sporadic and is diagnosted in one person per million per year worldwide. The key molecular event in the pathogenesis of CJD is the conformational change of cellular prion protein PrP^C^ to its abnormal isoform PrP^Sc^. Accumulation of PrP^Sc^ multimers induces vacuolation of gray matter by microglial activation and neuronal loss, leading to progressive neurodegeneration and astrogliosis. Usually, it manifests as a rapidly progressing dementia with ataxia and myoclonus, leading to death within one year [569,570].

Jeong et al. investigated the prevalence of different HERV families in CSF samples from healthy controls, patients with sporadic CJD, and patients with other (not specified in the paper) neurological diseases. RT-PCR showed that levels of HERV-W, HERV-L, FRD and ERV-9 are increased in CSF of individuals with sporadic CJD compared to controls. Concentrations of HERV-W and HERV-L were higher in CJD samples than in samples from the “other diseases” group. Importantly, the HERV-L family has not previously been shown to be associated with any other disease. The exact role of HERV expression in the pathogenesis of sporadic CJD is not clear, but the authors suggested it may be involved in neuroinflammatory processes [571].

#### 3.2.18. Neurofibromatosis Type I

Neurofibromatosis type I (NF1) is the most common autosomal dominant disorder affecting the nervous system [572], as well as the most common neurocutaneous disease [573]. Birth incidence of NF1 is estimated to be between 1:2000 and 1:3500 [364,572,574,575,576]. NF1 is a multisystem genetic disorder; it has a highly variable phenotype; and its symptoms include dermatological, ocular, neurological, musculoskeletal, vascular and cardiac manifestations, and tumors [575,576]. While most patients with NF1 have normal intelligence, their motor development is frequently delayed. Some patients show learning disabilities, ID, ASDs or epileptic seizures [573,574,576]. Learning or attention deficits or behavioral problems are evident in 50–80% of patients [576]. NF1 patients are prone to the development of benign and malignant tumors [572]; they have a five-fold risk for cancer in general and more than 2000-fold risk for neurogenic malignancies compared to the general population [575].

The only gene responsible for this disease is *NF1* [576] encoding neurofibromin 1, a protein participating in Ras and cAMP signaling pathways; in neurons it is involved in synaptic signaling and neural development [364,575,577]. No mutational hotspots have been identified in *NF1* [575]. However, it seems that this gene is particularly vulnerable for TE insertions. Wimmer et al. described 18 novel retrotransposon insertions within *NF1* in patients with neurofibromatosis type I. Strikingly, three of the identified TE integration sites were each used twice in non-related patients. Moreover, 6 out of 18 insertions were located close to each other, within a specific 1.5 kb region that constitutes less than 1% of this long gene. All this suggests that L1-mediated retrotransposition is not as random as previously thought, and some genomic sites are particularly vulnerable to it. Retrotransposon insertions account for ~0.4% of all *NF1* mutations, which is 2–4 times higher than expected considering the average insertion rate. Splicing defects were shown to be the prevalent effect of L1-mediated insertions identified in this study; there were 13 exonic Alu, L1 and poly(T) insertions that led to exon skipping or use of a cryptic splice site [363].

#### 3.2.19. Age-Related Macular Degeneration

Age-related macular degeneration (AMD) results from retinal pigmented epithelium (RPE) degeneration and is a leading cause of blindness. One of molecular traits specific for AMD is downregulation of *DICER1* expression; other miRNA-processing enzymes are not affected. Kaneko et al. sequenced dsRNA extracted from human RPE with AMD and identified increased level of Alu RNAs compared to control RPE samples. Experiments with pol III inhibitor applied to cultured RPE cells confirmed that Alu RNA molecules accumulating as a result of DICER1 deficiency are primary Alu transcripts and not sequences embedded in pol II-transcribed RNAs. Transfection of human or mouse RPE cells with a plasmid encoding Alu reduced cell viability. In vivo, subretinal transfection of plasmids encoding two different Alu RNAs induced RPE degeneration in WT mice; similar results were obtained with plasmids encoding B1 or B2 RNAs. Alu RNA accumulation in AMD and its experimental models correlated with increased caspase-3 cleavage, indicating a role for excess Alu RNA in apoptosis [578]. A following study by the same research group uncovered that the mechanism of Alu RNA cytotoxicity involves inflammatory response. Despite the immune privilege of the retina, multiple innate immune receptors are expressed in the RPE. Tarallo et al. tested a number of dsRNA sensor proteins for their interaction with Alu RNA in RPE cells using genetically modified mice strains. MyD88, a protein previously considered to be specific for immune cells, was identified as a dsRNA sensor responsible for Alu RNA cytotoxicity in RPE monoculture. Conditional KO of *MyD88* in the RPE protected against Alu RNA-induced RPE degeneration in vivo. Further experiments identified other participants of Alu RNA-mediated inflammatory response in RPE cells: IL-18, NLRP3 inflammasome and caspase-1 [471].

#### 3.2.20. X-Linked Dystonia-Parkinsonism

X-linked dystonia-parkinsonism (XDP) is a Mendelian neurodegenerative disease endemic to one island in Philippines. The clinical phenotype combines features of dystonia and parkinsonism, beginning with hyperkinetic symptoms that shift to hypokinetic movements over time; the mean disease onset age is 39.7 years. Aneichyk et al. integrated multiple genome and transcriptome assembly technologies to identify the mutation causing this disease. They found out that this is a SVA insertion within an intron of the *TAF1* gene [579].

#### 3.2.21. Ravine Encephalopathy

A very rare inherited condition was identified in a small polulation in the Ravine region of Reunion Island. This disease is characterized by extreme infantile anorexia and lethal encephalopathy with brainstem atrophy. It was associated with a point mutation in a degenerated TE (L1PA8 embedded in AluSz) in an intron of the *SLC7A2* gene. Two lncRNAs with unknown function, SLC7A2-IT1A and SLC7A2-IT1B, originate from this intron. The mutation greatly increases the probability for the mutant transcripts to be targeted by piRNA-mediated silencing. More than eight-fold decrease in SLC7A2-IT1A/B RNA level in patient brain tissue was registered. KD of SLC7A2-IT1A/B in human neuroblastoma cells caused their apoptose, suggesting the causal role of SLC7A2-IT1A/B silencing in Ravine encephalopathy [429].

## 4. Conclusions

TEs have fascinatingly diverse and complex interactions with host genomes. There are multiple examples of TE exaptation in different tissues and organs, and the introduction of whole-genome approaches in recent years allowed us to realize that the extent of TE involvement in multiple gene-regulation processes is actually much higher than had been previously considered.

Some germline TE insertions caused formation of new genes. Such TE domestication events played a role in the evolution of different cell types and tissues. It seems like domestications of a number of genes encoding neuron-specific proteins and non-coding RNAs happened by random chance rather than by some predictable pattern; the resulting genes are diverse in their function and derived from different TE families.

Apparently, the most prominent role of TE-related germline mutations in the human brain evolution was providing new regulatory elements. Of particular note is the impact of rapidly expanding Alu insertions in human evolution: it seems like these elements disproportionally contributed to the evolution of neuron-specific genes and their transcriptional regulation.

There is extensive evidence that in the brain, and specifically in the hippocampus, somatic TE insertions are much more common compared to non-neuronal somatic tissues. Most of the studies demonstrating increased TE mobilization in the brain did not discriminate between TE insertions that happened during and after neuronal differentiation; it should be noted that hippocampus includes an adult neurogenesis niche that contains NPCs. A number of in-vitro studies focused on NPCs showed that notable TE derepression (and activation of L1 elements in particular) happens during neuronal differentiation. This process is not random; some specific TE types were shown to be silenced in neurogenesis. It was suggested that at this stage L1-derived regulatory sequences are used to promote expression of specific genes. Another proposed role of TE mobilization in the context of neurogenesis is increasing heterogeneity in newly generated neurons. However, somatic retrotransposition was also confirmed to happen in adult neurons, but its function remains obscure.

TE activity was linked to multiple diseases with diverse mechanisms. TE mobilization or transcription was registered in various neuropathologies. In many cases, it is not obvious if there is a causal link between TE activity and pathological changes in neurons. HERV and L1 elements are associated with most of the diseases described in this review, since activation of these elements causes mutations and triggers innate immunity cascades. It was suggested that, at least in some cases, TE activation in neuropathology may simply reflect genome destabilization caused by some other factors.

TE involvement in myriad of normal and pathological processes emphasizes the importance of further studies in this field. However, TE abundance and repetitive nature make pinpointing specific transposition events particularly hard, especially in somatic cells. It is also important to highlight that functional assays are necessary to establish the causal role of transposition events in specific processes.

## Figures and Tables

**Figure 2 ijms-23-05847-f002:**
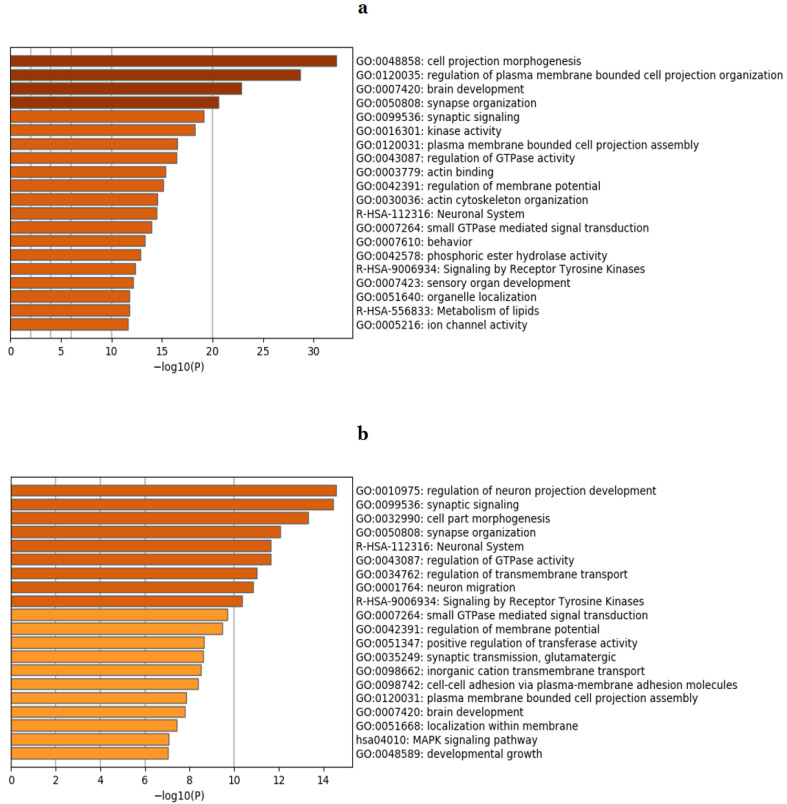
Analysis of datasets from [434]. (**a**). Gene Ontology enrichment analysis for the list of human genes targeted by recently mobilized transposons. (**b**). Gene Ontology enrichment analysis for the list of chimpanzee genes targeted by recently mobilized transposons.

**Table 1 ijms-23-05847-t001:** Mechanisms of interaction between TEs and host organisms. The right column shows section numbers in the present review.

**TE repression**	Epigenetic		DNA methylation	1.7.1
			Histone methylation	1.7.2
			KRAB-ZFP transcriptional repressors	1.7.3
	Epigenetic and post-transcriptional		RNA interference	1.7.4
	Other mechanisms			1.7.5
**TE exaptation**	TE derepression in cell stress conditions			1.7.7
	Regulation of host gene expression by TE sequences		Gene expression rate tuning	1.8.1
			Promoters, enhancers, insulators	1.8.2
			TF binding sites	1.8.3
	TE insertions affecting protein sequence	Insertions within exons, coding sequence mobilizations		1.8.4
		TE insertions affecting the fate of RNA	Exonization (new splice isoforms)	1.8.5
			Alternative polyadenylation sites	1.8.6
			RNA A-I editing	1.8.7
			Other mechanisms	1.8.8
	TE-encoded proteins and non-coding RNAs used by the host cell			1.8.9
	TEs and DNA repair			1.8.10

**Table 2 ijms-23-05847-t002:** Functions of exapted TEs in neuronal tissue.

Function	Cell Type	Type of TE Insertions	Section Number
A source of mutagenesis contributing to neuronal mosaicism	Neurons and NPCs	Somatic	2.2
RNA A-I editing of neuronal transcripts	Neurons, possibly NPCs	Germline	2.3
Facilitating neuronal differentiation via host gene transcription regulation	NPCs	Germline	2.4
Possible TE participation in neuronal plasticity	Neurons, possibly NPCs	Somatic	2.5
Possible TE participation in DNA repair	Neurons	Somatic	2.6
TE-derived *cis*-regulatory sequences in neuronal gene expression	Neurons and NPCs	Germline	2.7.1
TE-encoded neuronal proteins	Neurons, possibly NPCs	Germline	2.7.2
TE-encoded neuronal ncRNAs	Neurons and NPCs	Germline	2.7.3

**Table 3 ijms-23-05847-t003:** Neuropathologies associated with TE dysregulation.

Disease	Type of Disease	Associated TE Activity	Suggested Mechanisms of TE Participating in Pathogenesis	Section Number
Alzheimer’s disease	Neurodegenerative	Transcriptional activation of different TE types	Neuroinflammation; generation of defective *APP* retrocopies; speculated Alu-mediated alterations of *MAPT* transcripts	3.2.1
Parkinson’s disease	Neurodegenerative	Increased L1 transcription; germline TE insertions	Aggravation of oxidative stress; germline *PRKN* mutations	3.2.2
Huntington’s disease	Neurodegenerative	Increased L1 mobilization and transcription	Dysregulation of cell survival signaling pathways	3.2.3
Ataxia telangiectasia	Neurodegenerative	Increased L1 mobilization; L1 insertions with increased average length	No specific mechanisms proposed	3.2.4
Spinal muscular atrophy	Neurodegenerative, neuromuscular	Alu-mediated recombinations	Germline *SMN1* mutations	3.2.5
Amyotrophic lateral sclerosis	Neurodegenerative	Increased HERV-K transcription; increased L1 mobilization	Env protein toxicity; ORF1p aggregation with mutant host proteins	3.2.6
Fronto-temporal lobar degeneration	Neurodegenerative	Decreased binding between TDP-43 and its multiple TE RNA targets; increased L1 mobilization and transcription	No specific mechanisms proposed	3.2.6
Fragile X-associated tremor/ataxia syndrome	Neurodegenerative	Increased LTR transcription	The causal role of TEs in pathogenesis was confirmed, but the exact mechanisms of damage are unknown	3.2.7
Multiple sclerosis	Autoimmune, inflammatory	Increased HERV-W transcription	Env protein toxicity; Env-induced autoimmunity to myelin proteins	3.2.8
Aicardi–Goutières syndrome	Inflammatory, neurodelopmental	Increased L1 mobilization and transcription	Accumulation of L1 cDNA in the cytoplasm leading to IFN-I response and neuroinflammation	3.2.9
Glioblastoma	Malignant tumor	Increased HERV transcription; selective packaging of HERV RNA in microvesicles	No specific mechanisms proposed	3.2.10
Autism spectrum disorders	Neurodevelopmental	Increased L1 transcription; germline L1 and Alu insertions in autism-associated genes; increased HERV-H transcription	Increased L1 RNA level correlated with oxidative stress; germline TE insertions likely affect gene expression rate	3.2.11
Rett syndrome	Neurodevelopmental	Increased L1 mobilization; less frequent L1 insertions in exons	No specific mechanisms proposed	3.2.12
Schizophrenia	Mental (psychotic)	Increased L1 mobilization; germline and somatic L1 insertions in genes associated with SZ and similar pathologies; increased HERV-K10 and MSRV *env* transcription	L1 insertions affect gene expression rate (confirmed in vitro for somatic insertions). A link between ERV expression and myelin inflammation was proposed	3.2.13
Bipolar disorder	Mental (mood disorder)	Increased HERV-K10 and MSRV *env* transcription	A link between ERV expression and myelin inflammation was proposed	3.2.13
Major depressive disorder	Mental (mood disorder)	Increased L1 mobilization in blood cells; decreased L1 mobilization in prefrontal cortex	No specific mechanisms proposed	3.2.14
Post-traumatic stress disorder	Mental (anxiety)	Altered L1 and Alu methylation levels in blood cells; increased L1 transcription in the brain after stress exposure	No specific mechanisms proposed	3.2.15
Drug addiction	Substance use disorder	Increased L1 mobilization and transcription caused by metamphetamine, cocaine, morphine administration; L1 insertions in genes associated with cocaine addiction in addict brain samples; increased LTR and L1 transcription in alcoholic brain samples	Germline or early developmental stage somatic insertions in genes influencing predisposition to cocaine addiction	3.2.16
Creutzfeldt–Jakob disease	Neurodegenerative, prion	Increased HERV-W and HERV-L transcription	Neuroinflammation	3.2.17
Neurofibromatosis type I	Neurocutaneous	Germline TE insertions within *NF1* with unexpectedly high rate	Germline *NF1* mutations	3.2.18
Age-related macular degeneration	Neurodegenerative	Increased Alu transcription	Alu RNA accumulation in the cytoplasm causing innate immune response and apoptosis	3.2.19
X-linked dystonia-parkinsonism	Neurodegenerative	SVA insertion	Germline *TAF1* mutation	3.2.20
Ravine encephalopathy	Neurodegenerative	Point mutation within a TE-derived region	Germline *SLC7A2* mutation	3.2.21

## Data Availability

Publicly available datasets provided by third parties were analyzed in this study. These data can be found at https://www.cell.com/ajhg/fulltext/S0002-9297(07)63704-5#supplementaryMaterial (accessed on 20 May 2020). This publication is also included in the References list.

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
