# Peer review of "The Role of Transposable Elements of the Human Genome in Neuronal Function and Pathology"

_ijms, 2022, doi:10.3390/ijms23105847_

Round 1

Reviewer 1 Report

In their study titled “The role of transposable elements of the human genome in neuronal function and pathology,” Chesnokova, Beletskiy and Kolosov present an incredibly comprehensive review of transposable elements (TEs) and their various functions in the nervous system. The review introduces TEs, the various classes, structure and origins. It explains how TEs transpose, neofunctionalization of TEs, domestication of TEs, and TE contributions to evolution. The authors then review all of the neuronal diseases with reports of altered TE activity. In general, the review is well organized and well written. It is a huge undertaking to present the literature in such detail. The manuscript will have appeal to the TE community, and more broadly as an introduction to TEs and their functions in neuronal development, evolution and disease.

The biggest challenge with such a work is presenting the information without losing the reader, as the manuscript is more a book chapter than a review. To this end, it would help to have an “outline” at the beginning where the major sections and subheadings are listed, to facilitate navigation to specific topics. Along these lines, I would find it helpful if the authors could add a few summary tables or figures, one listing the various roles of TEs in neuronal tissue (generating genetic mosaicism, promoting RNA editing, response to stress, etc.), another summarizing all of the various diseases associated with TEs and a third for evolutionary functions/examples of exaptation, and another listing all of the examples of genes derived from TEs. This would help the reader get an overview of each section of the review, and enable them to access targeted information.

Another general suggestion would be to add introductory sentences and conclusion sentences to all paragraphs. There are multiple places where paragraphs either start or end directly with evidence from the literature, and the reader has to infer the conclusion the authors are trying to illustrate. Examples include Lines 624, 636, 799, 1126, 1235, 1353, 1756, 1961, 2025, etc. In some of these examples, the conclusion could help transition to the following section.

In some places, the language should be toned down. For example Line286 – “so exon disruption by Alus inserted within genes is definitely harmful” based on Ref83 showing Alus are rarely found in exons. This could reflect a bias in targeting or selective pressure against exon insertion, as suggested, but doesn’t necessarily mean all exon insertions are problematic. Another example is Line 387 – “must be invariably lethal”. In line 1159 – The authors state that large genes are neuronal specific. Some of the largest genes are also found in muscle tissue, perhaps the authors mean that very long genes often have neuron-specific isoforms? Line 1162 & 1874 – remove the “must”

There are a couple of places where I do not quite understand the relevance of entire paragraphs to human TE function in neurons. For example, the paragraph lines 1067-1074, lines 1127-1144, Lines 1237-1258 (here for example the causal link between TE and A-I editing is not really discussed, but this comes in a later section, it might make more sense to put this paragraph elsewhere). Another section of a paragraph is Lines 1376-1388. Given the length of the review, cutting out extraneous information, even if it is interesting in a different respect, could keep the review more focused.

The definition in line 837 is not correct for a cassette exon. I think the authors mean that the novel TE-derived exon serves as an alternative cassette exon, although it tends to only be included in minor splice isoforms. Also, in line 840, briefly list examples of “a few genetic diseases”.

Line 116 was unclear – how does this recent analysis in Ref38 indicate that “hot” L1s are more abundant, when the preceding line defines the number of copies as less than 100? Are more of them active, or more in different genetic backgrounds?

Line 125 – This sentence needs to be reworded – I am not sure if the elements that function in L1 mobilization or if the process of L1 mobilization are poorly understood.

Line141 – Is this accurate? I wasn’t aware that hnRNPA1 has an affinity to poly(A) sequences, as based on SELEX it binds UAGGGA/U and CLIP produced a motif of UUAGGGAG (see Bruun et al., BMC Biol, 2016).

Lines 324-326 – Could be moved up to Line299, to serve as an introduction to SVA elements.

Line 572 – Although the authors for Ref 168 didn’t propose a mechanism for this observation, maybe the authors have a mechanism in mind? Could chromatin state of genes and promoters or sequence bias in these regions actually explain the insertion bias?

Line 587 – “one of its reasons” might be clearer to understand as “or even a driver of genomic instability”

Line 599 – “hypertermia” I think should maybe be “hypothermia”?

Line 647 – Is this the concluding point for the preceding paragraph? It is an orphaned sentence.

Line 865 is also an orphaned sentence. I’m not sure where it fits with the bordering paragraphs. Same for Lines 931 to 935.

Line 1112 – I didn’t understand the basis for this sentence, is this reflecting endoreduplication? Why do neurons in different regions have different amounts of DNA?

Line 1199 – “lewels” should be “levels”

Line 1469 – “Reparation” should be “Repair”, reparation has a different meaning than what is meant here. There are several instances of “reparation” in the text that should be changes to “repair”.

Line 1500 – section title is 2.7 and subheading jumps to 2.8.1 on line 1501

Line 1605-1610 – Maybe move this section to the start of this paragraph (Line 1597). Arc needs to be introduced first, so that the paragraph makes more sense.

Line 2373 – I did not completely understand the evidence here of how mutations in SMN that lead to SMA are caused by Alu-mediated recombinations? This should be discussed more clearly. Is there actually direct evidence suggesting the Alu-like sequences in SMN cause the mutations, or is this conclusion just based on association?

Author Response

We thank the reviewer for taking their time to provide such a detailed and thorough review that helped to improve our paper. We corrected some specified places in the text; in this response we use line numbers corresponding to the review and to the previous version of the manuscript. In the revised version, line numbers are different. Some reference numbers are also different now because fragments of the text were rearranged.

The biggest challenge with such a work is presenting the information without losing the reader, as the manuscript is more a book chapter than a review. To this end, it would help to have an “outline” at the beginning where the major sections and subheadings are listed, to facilitate navigation to specific topics. Along these lines, I would find it helpful if the authors could add a few summary tables or figures, one listing the various roles of TEs in neuronal tissue (generating genetic mosaicism, promoting RNA editing, response to stress, etc.), another summarizing all of the various diseases associated with TEs and a third for evolutionary functions/examples of exaptation, and another listing all of the examples of genes derived from TEs. This would help the reader get an overview of each section of the review, and enable them to access targeted information.

We added three tables summarizing mechanisms described in three chapters of the review. These tables also include numbers of most subsections. We suppose that a table of contents would also be useful, but we are not able to add it now because page numbers are not specified yet.

Another general suggestion would be to add introductory sentences and conclusion sentences to all paragraphs. There are multiple places where paragraphs either start or end directly with evidence from the literature, and the reader has to infer the conclusion the authors are trying to illustrate. Examples include Lines 624, 636, 799, 1126, 1235, 1353, 1756, 1961, 2025, etc. In some of these examples, the conclusion could help transition to the following section.

We were aiming to cut most of the non-essential text because the review is already very long. Where possible, we summarized the data from different experimental papers in section beginnings and repeating it at section ends seems redundant. We added segues as requested, or rearranged text fragments near lines 636, 1126, 1235, 1756, 1961.

In some places, the language should be toned down. For example Line286 – “so exon disruption by Alus inserted within genes is definitely harmful” based on Ref83 showing Alus are rarely found in exons. This could reflect a bias in targeting or selective pressure against exon insertion, as suggested, but doesn’t necessarily mean all exon insertions are problematic. Another example is Line 387 – “must be invariably lethal”. In line 1159 – The authors state that large genes are neuronal specific. Some of the largest genes are also found in muscle tissue, perhaps the authors mean that very long genes often have neuron-specific isoforms? Line 1162 & 1874 – remove the “must”

We corrected lines 286, 387, 1162 and 1874 as specified.

We changed the phrasing in line 1159 to “many long genes are expressed in the brain or involved in neuronal-specific processes” that does not imply that long genes are not expressed in other tissues.

There are a couple of places where I do not quite understand the relevance of entire paragraphs to human TE function in neurons. For example, the paragraph lines 1067-1074, lines 1127-1144, Lines 1237-1258 (here for example the causal link between TE and A-I editing is not really discussed, but this comes in a later section, it might make more sense to put this paragraph elsewhere). Another section of a paragraph is Lines 1376-1388. Given the length of the review, cutting out extraneous information, even if it is interesting in a different respect, could keep the review more focused.

Lines 1067-1074 describe a paper about RNA-dependent DNA repair that very likely involves one of TE-encoded reverse transcriptases. This research group first described this mechanism in U2OS cells, but then they continued studying it in neurons (their paper about similar repair in neurons is described in the section 2.6). We shortened the description in lines 1067-1074 and added the mention of reverse transcriptases.

Lines 1127-1144 describe different mechanisms of mutations in neurons and NPCs that are not usually caused by transposons. We consider it necessary to mention that TEs are not the only  source of neuronal mutagenesis. This part also emphasizes an important distinction between NPCs and mature neurons: NPCs are much more prone to mutagenesis. Later we discuss that most TE mobilizations in neuronal tissue were in fact registered in NPCs.

We decided not to change lines 1237-1258. A-I editing is discussed in 3 different sections of our review, and while there is some overlap between them, they deal with different aspects of this phenomenon. In the section 1.8.7. we describe the mechanism in general and why it primarily targets TEs without focusing on its role in neurons; in the section 2.3 we describe its role in human and murine neurons; and in the the section 2.9.4 (2.8.4 in the revised version) we discuss the importance of Alu elements being widely distributed A-I editing sites specifically in primate genomes. In the revised version, we added a few changes to sections 1.8.7. and 2.3 to make it more clear that that A-I editing also happens in rodents that do not have Alus but have other TEs serving as A-I editing targets. We removed mentions of Alus from the headings and listed murine TEs that undergo editing.

We removed the detailed description of an old paper by Salganik et al. (lines 1376-1388) but we kept the citation.

We tried to shorten the text by cutting repeats and excessive details in a few more places, but since we added 3 tables and some requested elaborations, it is still very long.

The definition in line 837 is not correct for a cassette exon. I think the authors mean that the novel TE-derived exon serves as an alternative cassette exon, although it tends to only be included in minor splice isoforms. Also, in line 840, briefly list examples of “a few genetic diseases”.

We reworded this part so it does not imply that all cassette exons belong to minor splice isoforms: “Usually, the novel TE-derived exon is a so-called “cassette exon” that may be skipped as a result of splicing. Most TE-derived exons are included only in minor splice isoforms.”

We listed 3 hereditary diseases associated with Alu exons from the sources referenced in the cited review (https://pubmed.ncbi.nlm.nih.gov/17514354/). The review says “Pathologies such as the Alport  and the Sly syndromes are known to be caused by mutations that result in constitutive inclusion of an Alu exon... it was even discovered that alternative inclusion of an Alu exon might lead to a genetic disease... CCFDN syndrome”, but we do not see why only the last disease is called “genetic” here, since all 3 pathologies are inherited.

Line 116 was unclear – how does this recent analysis in Ref38 indicate that “hot” L1s are more abundant, when the preceding line defines the number of copies as less than 100? Are more of them active, or more in different genetic backgrounds?

We added the line “Beck et al. identified dozens of polymorphic active L1s with low allele frequencies”. The cited paper (https://pubmed.ncbi.nlm.nih.gov/20602998/) says “...68 full-length L1s that are differentially present among individuals but are absent from the human genome reference sequence... two L1s are only found in Africa... 37/68 of the newly identified L1s were ‘hot’ for retrotransposition when examined in a cultured cell assay... on average, the 68 L1Hs elements identified here are present at lower allele frequencies... than those in previous studies”.

Line 125 – This sentence needs to be reworded – I am not sure if the elements that function in L1 mobilization or if the process of L1 mobilization are poorly understood.

We changed the phrasing to “L1 sequence also contains some elements that seem to be functional, but their role in L1 mobilization is not obvious”. We hope it is clear that “elements” here refer to two extra promoters and one additional ORF described immediately after this sentence.

Line141 – Is this accurate? I wasn’t aware that hnRNPA1 has an affinity to poly(A) sequences, as based on SELEX it binds UAGGGA/U and CLIP produced a motif of UUAGGGAG (see Bruun et al., BMC Biol, 2016).

We changed the phrasing here to “hnRNPA1 that associates with poly(A)+ RNAs”. The paper cited in the manuscript (https://pubmed.ncbi.nlm.nih.gov/31010097/) says “hnRNPA1 is bound to poly (A) sequences of RNA in both the cytoplasm and nucleus” but it may be a misinterpretation of the cited source. We were able to find mentions of hnRNPA1 being accociated with poly(A)+ RNA in old and more recent papers: https://pubmed.ncbi.nlm.nih.gov/1371331/ , https://pubmed.ncbi.nlm.nih.gov/7510636/, https://pubmed.ncbi.nlm.nih.gov/11585913/, https://pubmed.ncbi.nlm.nih.gov/24065100/ , but these papers do not claim direct binding between hnRNPA1 and poly(A) sequence. Apparently this protein has a preferred binding motive, but may also bind other RNA sequences with lower affinity (https://pubmed.ncbi.nlm.nih.gov/8918813/). (We did not add these references to the revised version).

Lines 324-326 – Could be moved up to Line299, to serve as an introduction to SVA elements.

We aimed to describe three main types of human TEs using roughly the same plan: 1) the number of repeats in our genome; 2) the structure of the element and some details about its mobilization; 3) the most active subfamilies in humans. We did not change lines 324-326, but we moved the information about the L1HS Ta subfamily to the end of the corresponding section to make the descriptions more uniform (even though the description of L1 is much longer than two others).

Line 572 – Although the authors for Ref 168 didn’t propose a mechanism for this observation, maybe the authors have a mechanism in mind? Could chromatin state of genes and promoters or sequence bias in these regions actually explain the insertion bias?

Open chromatin being an easy target for TE integration would be the first guess, but L1 elements seem to be an exception among other transposons, and we discuss it in the section about L1 mobilization mechanism. As was shown by Sultana et al. (https://pubmed.ncbi.nlm.nih.gov/30956044/), L1 has a broad capacity for integration into all chromatin states. Sultana’s data show a link between L1 integration and host DNA replication. However, Kurnosov et al. (Ref 168) noted the insertion bias in both neurogenic and non-neurogenic brain regions, as well as in the myocardium, suggesting that DNA replication during cell proliferation is not a factor there.

The only explanation proposed by Kurnosov et al. is “...the uneven distribution of the discovered insertions in the genome can also result from the mapping bias: the sequencing reads better map to the unique genomic regions which comprise the actively transcribed chromatin than to the highly repetitive non-transcribed sequences.” We consider this too inconclusive and too long to cite.

We believe that some specific molecular mechanism favoring L1 insertions into genes and promoters exists, but has not been not discovered yet. We added this to the revised version, but it is pure speculation.

Line 587 – “one of its reasons” might be clearer to understand as “or even a driver of genomic instability”

We changed the wording as requested.

Line 599 – “hypertermia” I think should maybe be “hypothermia”?

“Hypertermia” is correct. The cited paper (https://pubmed.ncbi.nlm.nih.gov/10548739/) says “To cause hyperthermia, mice were... partially immersed in a 42.5°C bath for 25 min... rectal temperature in a mouse subjected to this procedure rose to 42°C after 12 min of heating and to a maximum of 43°C after 24 min.”

Line 647 – Is this the concluding point for the preceding paragraph? It is an orphaned sentence.

Line 865 is also an orphaned sentence. I’m not sure where it fits with the bordering paragraphs. Same for Lines 931 to 935.

Line 647 describes a mechanism of TEs influencing gene expression rate that is different from the mechanism in the previous paragraph. We rephrased this sentence and added the link to two next sections.

Lines 931-935 belong to the section listing “other mechanisms” that looks like bullet points by its nature, since it lists unique mechanisms that are not fitting into categories designated in previous sections.

The section with line 865 lists examples of exonized TEs. Line 865 is a short description of L1 exonization which has been observed but is not common. This line is between longer descriptions of Alu and LF-SINE exonizations. These 3 fragments differ in length but describe similar processes.

Line 1112 – I didn’t understand the basis for this sentence, is this reflecting endoreduplication? Why do neurons in different regions have different amounts of DNA?

We changed the phrase “neurons in different regions have different amounts of DNA” to more specific “neurons in frontal cortex have more DNA compared to the cerebellum of the same brain”.

The cited paper (https://pubmed.ncbi.nlm.nih.gov/20737596/) says “DCV [DNA content variation] within individual human brains showed regional variation, with increased prevalence in the frontal cortex and less variation in the cerebellum... There have been previous... reports of cell cycle reactivation with DNA synthesis in neurons of adult mammals... however our data do not support classical cell cycle mediated events... Aneuploidy... can be produced during neurogenesis... however, hypoploidy... is the most common form of demonstrated neural aneuploidy... which contrasts with the observed average increase in DNA content in the frontal cortex... Possible DCV functions include contributions to neural diversity and brain function by altering gene availability for transcriptional events or microRNAs.” The authors suggest two possible mechanisms explaining the observed phenomenon – aneuploidy and cell cycle reactivation – but it seems like both explanations are not exhaustive. We do not cite their speculations because in the next sentences we cite other papers describing known mutation mechanisms in neurons in general.

Line 1199 – “lewels” should be “levels”

The typo was fixed.

Line 1469 – “Reparation” should be “Repair”, reparation has a different meaning than what is meant here. There are several instances of “reparation” in the text that should be changes to “repair”.

We changed the term “reparation” to “repair” in all instances.

Line 1500 – section title is 2.7 and subheading jumps to 2.8.1 on line 1501

The numeration error was fixed.

Line 1605-1610 – Maybe move this section to the start of this paragraph (Line 1597). Arc needs to be introduced first, so that the paragraph makes more sense.

We moved the description of the ARC gene so now it precedes the description of the Arc protein.

Line 2373 – I did not completely understand the evidence here of how mutations in SMN that lead to SMA are caused by Alu-mediated recombinations? This should be discussed more clearly. Is there actually direct evidence suggesting the Alu-like sequences in SMN cause the mutations, or is this conclusion just based on association?

It seems like there is both direct evidence and speculation. The cited paper (https://pubmed.ncbi.nlm.nih.gov/29187847/) says “Transposable elements, including Alu elements, occupy more than 65% of the human SMN... A vast majority of SMA cases arise from deletion of a short genomic sequence encompassing exons 7 and 8 of SMN1... Although the information of the exact breakpoints of these deletions is not publicly available, they appear to include the Alu-rich intron 6 and the Alu-rich intergenic region downstream of exon 8... Other SMA cases involve Alu/Alu-mediated deletion of sequences from intron 4 through intron 6”. The part about exons 7-8 is a speculation based on abundance of Alu repeats in this region. The part about introns 4-6 has a link to another paper (https://pubmed.ncbi.nlm.nih.gov/10205265/) in which Alu-mediated mutations in this region were identified in SMA patients.

We changed the text to “Some of SMN1 mutations in SMA patients were found to be caused by Alu-mediated recombinations, and it was suggested that more yet unidentified Alu-related mutations in this gene may lead to SMA”.

Reviewer 2 Report

This review article presents the current state of the most recent publications on transposable elements (TEs) for human genome and its activity in neural tissue, and specifically in the hippocampus.
The role of mobile elements in genome evolution and functionality is poorly understood, and one might say that this topic is only at the beginning. Since, until now, the most common view of mobile elements is their parasitic and pathogenic nature. However, the situation with the evolution of the eukaryote genome, the adaptation of species and their genetic diversity, in parallel with mobile elements, is given little attention. Although this is most important, since the evolution of the eukaryote genome, without the participation of mobile elements, is impossible. 
Mobile elements are firstly retrotransposons, virus-like and endogenous viruses. The activity of mobile elements occurs individually in each cell. And if these changes have a critical effect, this cell dies. But other changes are preserved in somatic cells for the whole period of their life. Modern sequencing methods are limited in the length of reads and methods of assembling genomes. Therefore, we miss most of the changes in the human genome even when individuals are sequenced by third generation sequencing methods. 
In addition, tenderization of repeats occurs at the level of individual genomes, the length of which can vary greatly:
https://www.ncbi.nlm.nih.gov/labs/pmc/articles/PMC7215508/

Such tandem repeats are no less important for genome evolution and adaptation than individual TE insertions. TE insertions are still rare, and only make sense in generative cells that last for generations. While tenderization of repeats occurs more often and plays a significant role in the organization of chromosomes and throughout the entire cell cycle.
I think the author should mention the role of tandem repeats and the existing programs for their detection, if any.

Research in recent years has shown rather that non-coding sequences generally have higher control functions and are derived from retroviral RNA. Consequently, much of the human genetic material dates back to the integration of viruses during evolution. The main component of the human genome, therefore, does not consist of more or less stable determinants, but of flexible, superior functions of regulation and ordering, which were acquired in the process of co-evolution with the environment.

It was interesting for me to read this work and in general this review will be of interest to a wide range of specialists.
However, in the list of references that the authors have provided, there are no articles by authors in this field. It is preferable if the authors have also worked in this field and published articles in this field.
The authors have gone through a huge number of articles, which is of course a big work. But this is far from all that has been researched in this area. 
The human genome can produce hundreds of thousands of different protein sequences with its relatively modest number of coding genes. Since not the number of genes, but the diversity of alternatively spliced exons and in combination with mobile genetic elements, contribute to genetic diversity and complexity. It is interesting to note that the male Y chromosome is particularly poor in coding and rich in non-coding DNA of viral origin. Unlike all other human chromosomes, it does not contain Neanderthal DNA. 

https://mobilednajournal.biomedcentral.com/articles/10.1186/s13100-021-00250-2

https://www.ncbi.nlm.nih.gov/pmc/articles/PMC3376660/

As a progressive element, it is free from paleogenetic influences. Thus, the presence in the genome of the so-called "Junk DNA" is the result of "natural genetic engineering" that occurs in the cells of our body billions of times every second.

With each important stage of evolution, for example, with the advent of vertebrates, mammals, and, above all, primates, the number of endogenous retroviruses strains has sharply increased, which are the result of associations of symbiotic genomes. 

We must realise that existing approaches for genome sequencing and assembly are not perfect for analysing the rare and cell-specific changes associated with mobile elements. Therefore, what we have obtained from the human genome in the most recent version:
https://pubmed.ncbi.nlm.nih.gov/35357919/

is very far from reality, given the tandem and insertion-deletion diversity of mobile elements.

Some interesting work:
https://www.pnas.org/doi/10.1073/pnas.2105968118

Author Response

We thank the reviewer for reading and commenting our manuscript and for attracting our attention to new sources. We added citations from 3 of the new sources proposed by the reviewer (Hancks & Kazazian, Zhang et al. and secondary references from the review by Wang et al.) to the revised version. We did not use all recommended sources because our paper is already very long and we aim to focus on human transposons. By extension, we also discuss non-human mammalian and vertebrate TEs where analogies with human TEs could be made, and we mention a few examples from yeast and Drosophila research because these are model organisms in molecular biology. We are aware that many interesting mechanisms of transposon exaptation exist in plants, but we deliberately do not mention these in our paper because they have no relation to TE-dependent mechanisms in the human brain.

In addition, tenderization of repeats occurs at the level of individual genomes, the length of which can vary greatly:

https://www.ncbi.nlm.nih.gov/labs/pmc/articles/PMC7215508/

Such tandem repeats are no less important for genome evolution and adaptation than individual TE insertions. TE insertions are still rare, and only make sense in generative cells that last for generations. While tenderization of repeats occurs more often and plays a significant role in the organization of chromosomes and throughout the entire cell cycle.

I think the author should mention the role of tandem repeats and the existing programs for their detection, if any.

The term “tandem repeats” usually means simple repeats that are unable to mobilize, so we do not describe these elements in our review (besides the fact that “variable number of tandem repeats” is a part of SVA transposon sequence). The paper about LTR tandemization cited above refers to plant transposons that are also outside of the scope of our paper. We could not find any papers describing similar mechanisms in mammalian genomes.

However, in the list of references that the authors have provided, there are no articles by authors in this field. It is preferable if the authors have also worked in this field and published articles in this field.

Our lab primarily studies molecular mechanisms of transcription and translation in neurons, and we only started working with TEs 2 years ago, so we have no publications on this topic yet. We are currently obtaining results for a research paper about supposed L1-dependent DNA repair in primary neuronal cultures. We have one self-citation in the reference list (Chesnokova et al., a review about local translation in neurons).

Reviewer 3 Report

The manuscript is too long. It's poor of tables e diagrams to help to organize the data derived by this revision of literature. I consider the topic very important and I suggest the authors better organizse the collected data or publish a book chapter with the manuscript produced.

Author Response

We thank the reviewer for reading and commenting our manuscript.

The manuscript is too long. It's poor of tables e diagrams to help to organize the data derived by this revision of literature. I consider the topic very important and I suggest the authors better organizse the collected data or publish a book chapter with the manuscript produced.

We agree that the text is very long, and we removed some unimportant details during revision. However, other reviewers advised to add some more data, so we were not able to shorten the manuscript significantly. We added three tables summarizing the mechanisms described in each part to better organize the information.

Round 2

Reviewer 3 Report

I would thanks the authors for adding the tables that contribute to re-organize the work that they did.

I suggest to cite in the 1.8.2 section the review (doi:10.1080/07388551.2021.1888067) about the biotechnologial derived application from transposable elements.

In sections 1.8.3 or 1.8.4, I suggest citing the work of Palazzo et al. (doi:10.3390/cells11030583) on the role of Bari transposons in gene regulation and the role of their promoters capacity in trans-kingdom expression. From this point of view, interactions between promoter sequences and transcription factors appear to be less stringent and with a more plasticity. 

Author Response

We thank the reviewer for their advice.

I suggest to cite in the 1.8.2 section the review (doi:10.1080/07388551.2021.1888067) about the biotechnologial derived application from transposable elements.

We added the citation as suggested.

In sections 1.8.3 or 1.8.4, I suggest citing the work of Palazzo et al. (doi:10.3390/cells11030583) on the role of Bari transposons in gene regulation and the role of their promoters capacity in trans-kingdom expression. From this point of view, interactions between promoter sequences and transcription factors appear to be less stringent and with a more plasticity.

We added this citation to the introduction as a source of information about horizontal transposon transfer that was not mentioned in the previous manuscript version.